# How to Find the Exact Pareto Front for Multi-Objective MDPs?

**Yining Li** [*], **Peizhong Ju**[†]**,& Ness Shroff**[*,‡]

[*]Department of Electrical and Computer Engineering, The Ohio State University
[†]Department of Computer Science, University of Kentucky
[‡]Department of Computer Science and Engineering, The Ohio State University
`li.12312@osu.edu, peizhong.ju@uky.edu, shroff.11@osu.edu`

## Abstract

Multi-Objective Markov Decision Processes (MO-MDPs) are receiving increasing attention, as real-world decision-making problems often involve conflicting objectives that cannot be addressed by a single-objective MDP. The Pareto front identifies the set of policies that cannot be dominated, providing a foundation for finding Pareto optimal solutions that can efficiently adapt to various preferences. However, finding the Pareto front is a highly challenging problem. Most existing methods either (i) rely on traversing the *continuous preference space*, which is impractical and results in approximations that are difficult to evaluate against the true Pareto front, or (ii) focus solely on deterministic Pareto optimal policies, from which there are no known techniques to characterize the full Pareto front. Moreover, finding the structure of the Pareto front itself remains unclear even in the context of dynamic programming, where the MDP is fully known in advance. In this work, we address the challenge of efficiently discovering the Pareto front, involving both deterministic and stochastic Pareto optimal policies. By investigating the geometric structure of the Pareto front in MO-MDPs, we uncover a key property: the Pareto front is on the boundary of a convex polytope whose vertices all correspond to deterministic policies, and neighboring vertices of the Pareto front differ by only one state-action pair of the deterministic policy, almost surely. This insight transforms the global comparison across all policies into a localized search, drastically reducing the complexity of searching for the exact Pareto front. We develop an efficient algorithm that identifies the vertices of the Pareto front by solving a single-objective MDP only once and then traversing the edges of the Pareto front, making it more efficient than existing methods. Our empirical studies demonstrate the effectiveness of our theoretical strategy in discovering the Pareto front efficiently.

## 1 Introduction

In recent years, there has been growing interest in multi-objective Markov Decision Processes (MO-MDPs), where the reward involves multiple implicitly conflicting objectives (Xu et al., 2020b; Rame et al., 2024). Consequently, attention has shifted toward developing approaches for finding Pareto optimal policies, policies whose returns cannot be dominated by any other policies, in MO-MDPs and, more broadly, in multi-objective Reinforcement Learning (RL) (Abdolmaleki et al., 2020; Yang et al., 2019b; Lu et al., 2022; Zhou et al., 2024).

Finding Pareto optimal policies based on specific preferences is often sensitive to the scales of the objectives and struggles to adapt to changing preferences (Zhou et al., 2024; Qiu et al., 2024). Accurately capturing true preferences across multiple objectives is challenging, as the balance between objectives can be distorted by differences in scale (Abdolmaleki et al., 2020; Kim et al., 2024). When one reward objective has a much larger scale than others, even assigning a small weight to it cannot prevent it from dominating the scalarized reward, leading to an undesired Pareto optimal policy. In such cases, multiple adjustments to the preferences may be necessary to achieve the desired balance across objectives. Additionally, preferences may shift over time, requiring quick adaption to new preferences (Mossalam et al., 2016; Yang et al., 2019a; Jang et al., 2023). In such cases, solving for a Pareto optimal policy from scratch using a scalarization-based approach does not meet the need for fast adaptability.

On the other hand, the Pareto front which consists of all Pareto optimal policies directly represents the trade-offs between multiple objectives, thus avoiding sensitivity to differences in objective scales. Once the Pareto front is obtained, the Pareto optimal policies corresponding to various preference vectors can be selected from the Pareto front without recalculating from scratch.

The Pareto front approximation methods can be broadly divided into two categories. The first straightforward approach derives the Pareto front by solving single-objective problems obtained from scalarizing the multi-objective MDP for each preference and combining the resulting Pareto optimal policies (Qiu et al., 2024). Although state-of-the-art methods in the field of MO-RL leverage the similarity between Q-functions or policy representations of Pareto optimal policies for nearby preferences to avoid solving the optimal policy from scratch for each preference, thus reducing complexity (Parisi et al., 2014; Yang et al., 2019a; Lu et al., 2022), these methods still require sampling the continuous preference space to approximate the Pareto front. In cases where the Pareto front contains low-dimensional faces, sampling preference vectors on those faces makes it nearly impossible for approximation methods to identify such edge cases and accurately reconstruct the exact Pareto front. Another Pareto front finding method iteratively finds the preference that improves the current Pareto front the most to avoid blind traversing the preference space (Roijers, 2016; Mossalam et al., 2016). However, this method only identifies deterministic policies on the Pareto front, which does not capture the entire Pareto front. In 2D (2-objectives) cases, constructing the whole Pareto front from deterministic Pareto optimal policies simply connects adjacent deterministic points by a straight line. However, in general cases, it is not that simple. For example, in a 3D (3-objectives) case shown in Fig. 1, deriving the entire Pareto front (as in Fig. 1a) from the vertices corresponding to deterministic Pareto optimal policies (as in Fig. 1b) alone still requires significant effort. Therefore, even in scenarios where the MDP is fully known, deriving the entire Pareto front through dynamic programming remains complex. This raises a fundamental question:

*How can we efficiently obtain the full exact Pareto front in MO-MDPs?*

In this work, we address the problem of efficiently finding the Pareto front in MO-MDPs. We explore the geometric properties of the Pareto front in MO-MDP and reveal a surprising property of deterministic policies on the Pareto front: the Pareto front lies on the boundary of the convex polytope, and the neighboring deterministic policies, corresponding to neighboring vertices of the Pareto front, differ by one state-action pair almost surely. Building on this key insight, we propose an efficient algorithm for finding the Pareto front.

Specifically, our contributions are as follows:

1. **Geometric Properties of the Pareto Front**: We reveal several key geometric properties of the Pareto front, which form the foundation of our efficient Pareto front searching algorithm. First, we demonstrate that the Pareto front lies on the boundary of a convex polytope, with its vertices corresponding to deterministic policies. Notably, any neighboring policies on this boundary differ by only one state-action pair, almost surely[1]. Furthermore, we prove that deterministic Pareto optimal policies are connected and all Pareto-optimal faces incident to a deterministic policy can be efficiently extracted from the convex hull formed by that policy and its neighbors.

2. **Efficient Pareto Front Searching Algorithm**: We propose an efficient Pareto front searching algorithm that can discover the exact Pareto front, while also identifying all deterministic Pareto-optimal policies as a byproduct. The proposed algorithm only requires solving the single-objective optimal policy once for initialization, and the total iteration number is the same as the number of vertices on the Pareto front.

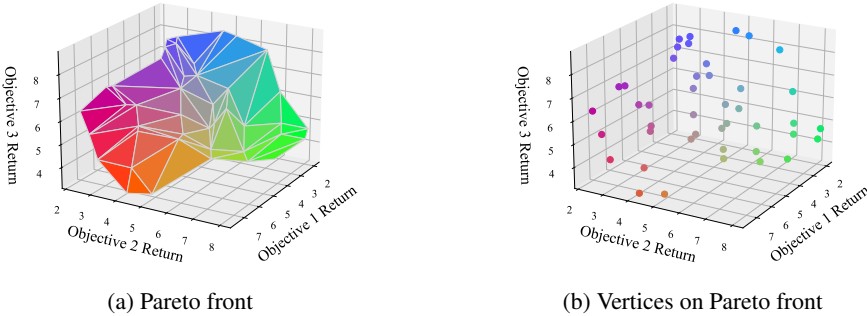

(a) Pareto front        (b) Vertices on Pareto front

Figure 1: Finding Pareto front and Pareto front vertices in MO-MDP

## 2 RELATED WORK

In the literature on both multi-objective optimization and reinforcement learning, two categories of problems are commonly considered. The first assumes that the (sequential) decision maker has preset or deduced preferences over

---

[1] For almost all $\mathbf{P}$ and $\boldsymbol{r}$ in the space of valid transition kernels and rewards (with Lebesgue measure 1), the neighboring deterministic policies on the convex polytope differs by at most one state-action pair.

multiple objectives in advance, and the goal is to find a Pareto optimal solution that performs well under these given preferences (Fernando et al., 2023; Chen et al., 2024; Mahapatra & Rajan, 2020; Xiao et al., 2024). The methods addressing this problem in multi-objective MDPs are specifically referred to as single-policy methods (Siddique et al., 2020; Peschl et al., 2021; Hwang et al., 2023; Zhou et al., 2024; Qiu et al., 2024). The second category focuses on finding a set of (approximate) Pareto optimal solutions that are diverse enough to represent the full range of the Pareto front (Liu et al., 2021; Zhang et al., 2023; Ye et al., 2024). This allows the (sequential) decision maker to choose the most preferable solution afterward. Due to its flexibility in adapting to changing preferences, this approach has received significant attention. However, finding the Pareto optimal front in the multi-objective optimization field does not take advantage of the specific structure of MO-MDPs, where multiple objectives share the same state and action spaces, as well as identical transition dynamics. Exploiting this structure in MO-MDPs could potentially lead to significant reductions in computational complexity. This section overviews the methods to get the Pareto optimal front estimation in MO-MDPs and MO-RL.

As early as White (1982), the dynamic programming-based method was proposed to find the optimal Pareto front for MO-MDPs, though the size of the candidate non-stationary policy set grows exponentially with the time horizon. To efficiently find the vertices on the Pareto front, Mossalam et al. (2016) introduced the Optimistic Linear Support (OLS) method, which solves a single-objective problem along the direction that improves the current set of candidate policies the most. However, OLS can only find vertices on the Pareto front and does not provide a method for constructing the entire Pareto front from these vertices, which remains non-trivial.

Van Moffaert & Nowé (2014) proposed Pareto Q-learning, which learns a set of Pareto optimal policies in the MO-RL setting. Building on this, subsequent work has investigated training a universal Pareto optimal policy set, where the policy network takes preference embedding as input and outputs the corresponding Pareto optimal policy (Yang et al., 2019a; Abels et al., 2019; Zhou et al., 2020; Lu et al., 2022). These Q-learning variants rely on the maximization of Q-function estimations over the action space, making them suitable only for discrete action spaces. Recent works have extended Q-learning with policy networks to handle continuous action spaces (Xu et al., 2020a; Basaklar et al., 2022). While these Q-learning methods update the Q-function for each preference vector by leveraging the Q-functions of similar visited preference vectors, thus avoiding the need to retrain from scratch, they still require updating the Q-function for each visited preference vector. Consequently, they still need to explore the preference space to achieve a near-accurate Pareto front. This process becomes computationally inefficient as the reward dimension increases.

There are also policy-based MORL algorithms designed to find Pareto optimal policies (Chen et al., 2021; Cai et al., 2023; Zhou et al., 2024; Parisi et al., 2014). A concept similar to our approach, which directly searches over the Pareto front, is the Pareto-following algorithm proposed by Parisi et al. (2014). This algorithm employs a modified policy-gradient method that adjusts the gradient direction to follow the Pareto front. While it can reduce the complexity of finding the Pareto front by avoiding the need to converge to the optimal policy from scratch, it requires multiple policy-gradient steps to identify even a nearby Pareto optimal policy. Furthermore, it cannot guarantee comprehensive coverage of the estimated policies, nor can it ensure discovering the true Pareto front.

In addition to value-based and policy-based methods that extend from single-objective RL, there are also heuristic approaches for combining policies that have shown promising performance in specific settings. For instance, rewarded soup (Rame et al., 2024; Jang et al., 2023) proposes learning optimal policies for individual preference criteria and linearly interpolating these policies to combine their capabilities. While Rame et al. (2024) demonstrated that rewarded soup is optimal when individual rewards are quadratic with respect to policy parameters, this quadratic reward condition is highly restrictive and does not apply to general MDPs.

# 3 PARETO FRONT SEARCHING ON MO-MDP

We begin with introducing to the basic concepts underlying MO-MDPs. Next, we provide an overview of the proposed efficient Pareto front searching algorithm to offer intuition into its workings. Finally, we present a detailed explanation of the algorithm. A proof sketch of the proposed algorithm will be provided in Section 4.

## 3.1 PRELIMINARIES ON MO-MDP

We consider a discounted finite MO-MDP $(\mathcal{S}, \mathcal{A}, \mathbf{P}, \boldsymbol{r}, \gamma)$, where $\mathcal{S}$ and $\mathcal{A}$ are the sets of states and actions, with cardinalities $S$ and $A$, respectively. The transition kernel $\mathbf{P}$ defines the probability $\mathbf{P}(s'|s, a)$ of transitioning from $s$ to $s'$ given action $a$. The reward tensor $\boldsymbol{r} \in \mathbb{R}^{S \times A \times D}$ specifies the $D$-dimensional reward $\boldsymbol{r}(s, a)$ obtained by taking action $a$ in state $s$. $\gamma < 1$ is the discount factor.

**Assumption 1.** *(Sufficient Coverage of the Initial State Distribution) We assume that the initial state distribution $\mu > 0$, meaning that the probability of starting from any state is larger than zero.*

The sufficient coverage of the initial state distribution assumption is commonly used in RL (Agarwal et al., 2020). Define $\Pi$ as the set of stationary policies, where for any $\pi \in \Pi$, $\pi(a|s)$ represents the probability of selecting $a$ at state $s$. The value function with policy $\pi$ is denoted as $\mathbf{V}^\pi(s) \in \mathbb{R}^d$, which is written as

$$\mathbf{V}^\pi(s) = \mathbb{E}\left[\sum_t \gamma^t \boldsymbol{r}(s_t, a_t)|s_0 = s, a_t \sim \pi(\cdot|s_t), s_{t+1} \sim \mathbf{P}(\cdot|s_t, a_t)\right].$$

We define the long-term return of policy $\pi$ as $J^\pi(\mu) = \mathbb{E}_{s\sim\mu(\cdot)}\left[\mathbf{V}^\pi(s)\right]$. Let $\mathbb{J}(\mu)$ denote the set of expected long-term returns for all stationary policies under the initial state distribution $\mu$, i.e., $\mathbb{J}(\mu) = \{J^\pi(\mu)|\pi \in \Pi\}$. For simplicity of notation, we will omit $\mu$ in the following sections.

$\mathbb{J}$ is convex as proven in Lu et al. (2022). We further emphasize that $\mathbb{J}$ is not only convex but also a convex polytope, with each vertex corresponding to a deterministic policy, as established in Lemma 1[2].

**Lemma 1.** *In discounted finite MO-MDP, under Assumption 1, $\mathbb{J}$ is a closed convex polytope, and its vertices are achieved by deterministic policies.*

As we investigate the boundary of $\mathbb{J}$, which is a convex polytope, we introduce relevant definitions concerning convex polytopes. Let $\mathbb{C} \subseteq \mathbb{R}^d$ be a convex polytope. A *face* of $\mathbb{C}$ is any set of the form $\mathbb{F} = \mathbb{C} \cap \{\boldsymbol{x} \in \mathbb{R}^d : \boldsymbol{w}^\top \boldsymbol{x} = c\}$, where $\boldsymbol{w}^\top \boldsymbol{y} \leq c$ for any $\boldsymbol{y} \in \mathbb{C}$. The faces of dimension 0, 1, and $\dim(\mathbb{C}) - 1$ are called vertices, edges, and facets, respectively.

To address the trade-offs between conflicting objectives, we introduce the well-known notions of dominance and Pareto optimality. For any two vectors $\boldsymbol{u}$ and $\boldsymbol{v}$, $\boldsymbol{u}$ *strictly dominates* $\boldsymbol{v}$, denoted as $\boldsymbol{u} \succ \boldsymbol{v}$, if each component of $\boldsymbol{u}$ is at least as large as the corresponding component of $\boldsymbol{v}$, and at least one component is strictly larger. A vector $\boldsymbol{u}$ is *Pareto optimal* in a set $\mathbb{S}$ if any other vector $\boldsymbol{v} \in \mathbb{S}$ does not strictly dominate $\boldsymbol{u}$. The *Pareto front* of a set $\mathbb{S}$, denoted as $\mathbb{P}(\mathbb{S})$, is the set of Pareto optimal vectors in $\mathbb{S}$.

In the context of MO-MDP, we say that policy $\pi_1$ dominates policy $\pi_2$, denoted as $\pi_1 \succ \pi_2$ if $J^{\pi_1} \succ J^{\pi_2}$. A policy $\pi$ is Pareto optimal if $J^\pi$ is Pareto optimal in $\mathbb{J}$. We similarly define the Pareto front of $\Pi$ as $\mathbb{P}(\Pi) = \{\pi \in \Pi \mid J^\pi \in \mathbb{P}(\mathbb{J})\}$.

### 3.2 ALGORITHM OVERVIEW

This section provides an overview of our proposed algorithm for efficiently identifying the entire Pareto front. By using a single-objective solver *only once* to get an initial Pareto optimal point and then searching along the surface of the Pareto front, our method ensures complete coverage of the Pareto front.

The algorithm maintains a queue $Q$ containing Pareto optimal policies whose neighboring policies have yet to be explored. The queue is initialized with a deterministic policy $\pi_0$, derived by solving a single-objective MDP, which is formulated from the original multi-objective MDP using an arbitrary positive preference vector, ensuring that $\pi_0$ lies on the Pareto front. The obtained policy $\pi_0$ serves as the starting point for systematically traversing the entire Pareto front.

In each loop, we pop out one element $\pi$ in the queue $Q$. The following steps are applied to explore the Pareto front in the neighborhood of $\pi$. We also use Fig. 2 to give an illustration of those steps in each loop.

(a) **(Neighbor Search)** First, we identify and evaluate all deterministic policies that differ from $\pi$ by only one state-action pair. From these, we discard any policies whose returns are dominated by returns of others[3].

As shown in Fig. 2a, the gray planes and edges represent previously discovered parts of the Pareto front, the black point corresponds to the returns of $\pi$ (the deterministic Pareto-optimal policy being explored), and the red points represent the policies that are not dominated among those differing from $\pi$ by only one state-action pair.

(b) **(Incident Faces Calculation)** Next, we compute the incident faces of the return of $\pi$ on the convex hull formed by the returns of $\pi$ and the policies identified in the first step, i.e., the non-dominated policies among deterministic policies differing from $\pi$ by only one state-action pair.

As depicted in Fig. 2b, the incident faces of $\pi$ on the constructed convex hull are illustrated with purple planes, and the black lines represent the edges of these incident faces.

(c) **(Pareto Face Extraction)**. In this final step, we select the faces from the incident faces identified in the second step. We add a face from the incident faces to the Pareto front if a non-negative combination of the

---

[2]Proof of this lemma is in Appendix D.

[3]We say that $\boldsymbol{u}$ is dominated by $\boldsymbol{v}$ if $\boldsymbol{u}$ is less than or equal to $\boldsymbol{v}$ in all objectives. A more formal definition of dominance is provided in Section 3.1.

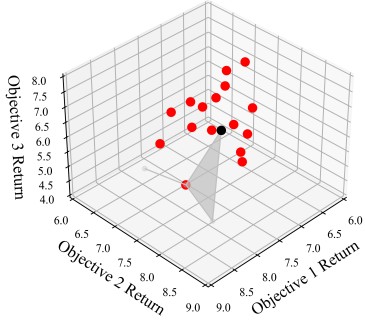
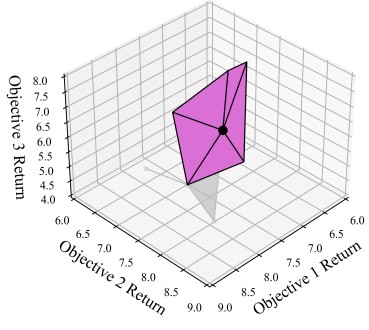
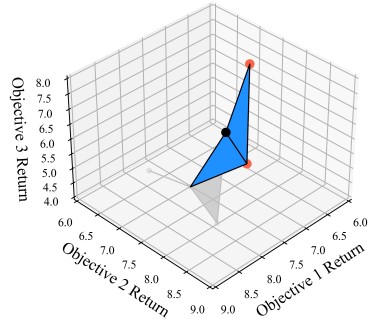

(a) Step 1: Selecting non-dominated distance-1 policies

(b) Step 2: Generate convex hull of current policy and non-dominated policies

(c) Step 3: Extract incident Pareto optimal faces

Figure 2: Illustrations of the steps for finding Pareto-optimal front at each iteration. $S = 5$, $A = 5$, and $D = 3$.

normal vectors of all intersecting facets on the face yields a strictly positive vector. Once a face is selected, all neighboring vertices on this face are added to the set of deterministic Pareto optimal policies. These vertices are also added to the queue $Q$ to explore their neighboring faces in the next iterations.

As illustrated in Fig. 2c, the selected faces are added to the set of Pareto front (marked in blue), and all neighboring vertices on the incident faces are shown as orange dots.

Applying the above steps to all policies in the queue $Q$ until the queue is empty ensures that the neighboring policies of all possible deterministic policies on the Pareto front have been visited. Since all vertices and their incident faces are traversed by the proposed algorithm, it guarantees full exploration of all faces of the Pareto front.

The intuition behind the proposed algorithm is that any deterministic Pareto optimal policy can identify all neighboring policies on the Pareto front by considering deterministic policies that differ by only one state-action pair. This ensures that all Pareto optimal policies are discovered during the search process. Additionally, the Pareto optimality justification excludes any policies that are not Pareto optimal, guaranteeing that the search trajectory remains on the Pareto front. The detailed theoretical results will be shown in Section 4. As the algorithm is guaranteed to consistently search along the Pareto front, the total number of iterations is the same as the number of deterministic Pareto optimal policies.

### 3.3 PARETO FRONT SEARCHING

This section presents the pseudocode and details of the proposed Pareto front searching algorithm. Before diving into details, we introduce some necessary notations as preparation for presenting our algorithms[4]. We denote the set of deterministic policies that differ from $\pi$ by exactly one state-action pair as $\Pi_1(\pi)$, and the set of non-dominated policies among $\Pi_1(\pi)$ as $\Pi_{1,\mathrm{ND}}(\pi)$. Let $\mathrm{Conv}(J^{\Pi_{1,\mathrm{ND}}} \cup \{J^\pi\})$ represent the convex hull constructed from the long-term returns of $\Pi_{1,\mathrm{ND}}(\pi)$ and $\pi$. The set of incident faces of $J^\pi$ on the convex hull $\mathrm{Conv}(J^{\Pi_{1,\mathrm{ND}}} \cup \{J^\pi\})$ is denoted by $\mathbb{F}(J^\pi, \mathrm{Conv}(J^{\Pi_{1,\mathrm{ND}}} \cup \{J^\pi\}))$.

The pseudocode for the Pareto front searching algorithm is presented in Algorithm 1. While the primary objective of Algorithm 1 is to identify the entire Pareto front, a byproduct of the algorithm is the identification of all vertices on the Pareto front that correspond to all deterministic Pareto optimal policies. Therefore, the output of Algorithm 1 includes both the Pareto front, denoted as $\mathbb{P}(\Pi)$, and the set of deterministic Pareto optimal policies, denoted as $\mathbb{V}(\Pi)$.

We begin by finding a Pareto optimal policy through scalarization and adding it to the queue $Q$ as part of the initialization, as detailed in Lines 4 and 5 of Algorithm 1.

**Neighbour Search** As detailed in Lines 9 and 10 of Algorithm 1, for any deterministic Pareto optimal policy $\pi$ popped from $Q$, we first evaluate all policies in $\Pi_1(\pi) \setminus \mathbb{Z}$, which differ from $\pi$ by only one state-action pair and have not yet been visited (with the visited policies stored in $\mathbb{Z}$). Once the long-term returns of $\Pi_1(\pi)$ are known, we select the set of non-dominated policies, $\Pi_{1,\mathrm{ND}}(\pi)$, the set of policies in $\Pi_1(\pi)$ that cannot be dominated by any other policies in $\Pi_1(\pi)$. Specifically, the PPrune algorithm from Roijers (2016) efficiently identifies the set of non-dominated elements from a given set, whose details are shown in Algorithm 3 in Appendix I.

---

[4]Notation table is provided in Table 1 of Appendix B.

---

**Algorithm 1** Pareto Front Searching Algorithm

---

**Input:** MDP settings: $(\mathcal{S}, \mathcal{A}, \mathbf{P}, \boldsymbol{r}, \gamma)$
**Output:** the set of Pareto optimal faces $\mathbb{P}(\Pi)$ and the set of deterministic Pareto optimal policies $\mathbb{V}(\Pi)$

1: Initialize $\mathbb{P}(\Pi) \leftarrow \varnothing, \mathbb{V}(\Pi) \leftarrow \varnothing$.
2: Initialize queue $Q \leftarrow \varnothing$.          *// Queue of Pareto optimal policies to explore*
3: Initialize set $\mathbb{Z} \leftarrow \varnothing$.          *// Set of visited deterministic policies*
4: **Initialization:** Sample a weight vector $\boldsymbol{w} \succ \mathbf{0}$, solve the optimal deterministic policy $\pi_0$ for $(\mathcal{S}, \mathcal{A}, \mathbf{P}, \boldsymbol{w}^\top \boldsymbol{r}, \gamma)$.
5: $Q.\text{push}(\pi_0), \mathbb{V}(\Pi).\text{add}(\pi_0)$.
6: **while** $Q$ is not empty **do**
7:     $\pi \leftarrow Q.\text{pop}()$.
8:     Select $\Pi_1(\pi)$.          *//Select all deterministic policies that differ from $\pi$ by one state-action pair*
9:     Evaluate $\Pi_1(\pi) \setminus \mathbb{Z}$ and update $\mathbb{Z}$ by $\mathbb{Z}.\text{add}(\Pi_1(\pi))$.          *//Evaluate policies not yet visited*
10:     Select non-dominated policies $\Pi_{1,\text{ND}}(\pi)$ from $\Pi_1(\pi)$.
11:     Calculate the convex hull $\text{Conv}(J^{\Pi_{1,\text{ND}}} \cup \{J^\pi\})$ formed by $\{J^{\tilde{\pi}} \mid \tilde{\pi} \in \Pi_{1,\text{ND}}(\pi) \cup \{\pi\}\}$.
12:     Extract $\mathbb{F}(J^\pi, \text{Conv}(J^{\Pi_{1,\text{ND}}} \cup \{J^\pi\}))$.          *//the set of incident facets of $J^\pi$ on $\text{Conv}(J^{\Pi_{1,\text{ND}}} \cup \{J^\pi\})$*
13:     Extract Pareto optimal faces $\tilde{\mathcal{F}}$ and vertices $\tilde{\mathcal{V}}$ from $\mathbb{F}(J^\pi, \text{Conv}(J^{\Pi_{1,\text{ND}}} \cup \{J^\pi\}))$ using Algorithm 2.
14:     $Q.\text{push}(\tilde{\mathcal{V}} \setminus \mathbb{V}(\Pi))$.
15:     $\mathbb{P}(\Pi).\text{add}(\tilde{\mathcal{F}}), \mathbb{V}(\Pi).\text{add}(\tilde{\mathcal{V}})$.
16: **end while**

---

Although the number of policies differing by one state-action pair is proportional to the state and action space, i.e., $|\Pi_1(\pi)| = S \times (A - 1)$, the number of non-dominated policies $\Pi_{1,\text{ND}}(\pi)$ is usually small. This helps reduce the complexity of calculating the convex hull and extracting Pareto front faces in the subsequent steps.

**Incident Facets Calculation** In Lines 11 and 12, we aim to compute $\mathbb{F}(J^\pi, \text{Conv}(J^{\Pi_{1,\text{ND}}} \cup \{J^\pi\}))$, the set of $(D-1)$-dimensional facets of $\text{Conv}(J^{\Pi_{1,\text{ND}}} \cup \{J^\pi\})$ that intersect at $J^\pi$. A straightforward approach is to calculate the convex hull of the set of vertices $J^{\Pi_{1,\text{ND}}} \cup \{J^\pi\}$ (as shown in Line 11) and select all facets that intersect at $J^\pi$. However, since we are only interested in the facets of $\text{Conv}(J^{\Pi_{1,\text{ND}}} \cup \{J^\pi\})$ that involve $J^\pi$, computing the entire convex hull is unnecessary. Specifically, facets that do not involve $J^\pi$ do not need to be considered. An efficient solution is to adapt the Quickhull algorithm (Barber et al., 1996), a well-established method for computing the convex hull of a set of points, to our algorithm. This adaptation focuses on updating only the facets related to $J^\pi$, thereby avoiding unnecessary calculations of irrelevant facets.

**Pareto Face Extraction** We apply Algorithm 2 to extract all faces where the Pareto front intersects at $J^\pi$ from $\mathbb{F}(J^\pi, \text{Conv}(J^{\Pi_{1,\text{ND}}} \cup \{J^\pi\}))$.

Consider a $k$-dimensional face of a $D$-dimensional polytope. Let the normal vectors to the $D-1$-dimensional facets that intersect at this face be denoted as $\{\boldsymbol{w}_i\}_{i=1}^n$. Specifically, when $k = D - 1$, then the facet that intersects on the face is itself, and $n = 1$. The face is Pareto optimal if for any $j = 1, \cdots, D$,

$$\sum_{i=1}^n \alpha_i^* [\mathbf{w}_i]_j > 0, \tag{1}$$

where $\boldsymbol{\alpha}^*$ is the solution to the following linear programming problem:

$$\text{maximize}_{\boldsymbol{\alpha}} \quad \min_{j \in \{1, \cdots, D\}} \sum_{i=1}^n \alpha_i [\mathbf{w}_i]_j, \tag{2}$$

$$\text{subject to} \quad \alpha_i \geq 0, \quad \forall i \in \{1, 2, \ldots, n\}.$$

Here, $\mathbf{w}_i = [[\mathbf{w}_i]_1, \cdots, [\mathbf{w}_i]_D]^\top$ represents the normal vectors, and $[\mathbf{w}_i]_j$ refers to the $j$-th component of the $i$-th normal vector. The objective function aims to maximize the smallest component of the weighted sum across all dimensions $j = 1, \cdots, D$, ensuring that the face is Pareto optimal. A detailed lemma showing the Pareto front criterion and its proof is presented in Appendix G.

Figure 3: Convex polytope and its Pareto front (red edge and plane)

It is possible that a high-dimensional face is not on the Pareto front, while some lower-dimensional components of that face still belong to the Pareto front. Figure 3 illustrates this scenario. The cyan and red planes form a 3D polytope, where the red line and red planes represent the Pareto front of the polytope. Notably, the Pareto front includes not only 2D planes but also 1D edges, such as the line segment AB. This observation implies that even if a high-dimensional

face (e.g., the plane containing AB) is not Pareto optimal, it is necessary to examine its lower-dimensional faces (e.g., edge AB) to determine if they are Pareto optimal.

In Algorithm 2, We traverse from $(D-1)$-dimensional faces down to 1-dimensional faces (edges) to verify if each face is Pareto optimal. Faces that have not yet been verified are pushed into the queue $Q_{\text{face}}$. If the face $H$ popped from $Q_{\text{face}}$ is verified to be Pareto optimal via Eq. (1), it is added to the Pareto face set, and its vertices are added to the set of Pareto optimal vertices. Otherwise, we proceed to examine the subfaces of $F$ that are one dimension lower.

Note that if a Pareto optimal face has vertices corresponding to deterministic policies $\{\pi_i\}_{i=1}^k$, then the policies constructing the face are convex combinations of $\{\pi_i\}_{i=1}^k$, expressed as $\{\pi|\pi(a|s) = \sum_{i=1}^k \alpha_i \pi_i(a|s), \forall s, a, \alpha_i \geq 0, \sum_{i=1}^k \alpha_i = 1\}$. The lemma demonstrating this result, along with its detailed proof, can be found in Lemma 24 of Appendix G.

---

**Algorithm 2** Pareto Optimal Face Selection

---

**Input:** Incident faces $\mathbb{F}(J^\pi, \text{Conv}(J^{\Pi_{1,\text{ND}}} \cup \{J^\pi\}))$

1: Initialize the set of Pareto optimal faces $\mathcal{F} \leftarrow \varnothing$, the set of Pareto optimal vertices $\mathcal{V} \leftarrow \varnothing$, the queue of faces for checking Pareto optimality $Q_{\text{face}} \leftarrow \varnothing$.
2: For all $G \in \mathbb{F}(J^\pi, \text{Conv}(J^{\Pi_{1,\text{ND}}} \cup \{J^\pi\}))$, $Q_{\text{face}}.\text{push}(G)$ and $w(G) \leftarrow$ normal vector of $G$.
         *// $w(G)$ represents the set of normal vectors of facets that intersect $G$*
3: **while** $Q_{\text{face}}$ is not empty **do**
4:     $H \leftarrow Q_{\text{face}}.\text{pop}()$.
5:     **if** there exist vertices of $H$ not in $\mathcal{F}$ **then**          *// $H \notin \mathcal{F}$*
6:        **if** $w(H)$ has a feasible solution to Eq. (1) **then**
7:           $\mathcal{F}.\text{add}(H)$ and $\mathcal{V}.\text{add}(\text{vertices of } H)$.
8:        **else if** $\dim(H) > 1$ **then**
9:           Let $\mathcal{S}$ be the set of $(\dim(H) - 1)$-dimensional faces of $H$ that intersect $J^\pi$.
10:          **for** each $\tilde{H} \in \mathcal{S}$ **do**
11:             $Q_{\text{face}}.\text{push}(\tilde{H})$ and $w(\tilde{H}).\text{add}(w(H))$.
12:          **end for**
13:        **end if**
14:     **end if**
15: **end while**
16: Return $\mathcal{F}$ and $\mathcal{V}$.

---

## 4 WHY CAN THE PARETO-FRONT SEARCHING ALGORITHM WORK?

In this section, we present the theoretical foundations behind Algorithm 1. This section is divided into three parts, and we give an overview of each part as follows.

1. **Distance-one Property on Boundary of $\mathbb{J}(\mu)$:** We prove that neighboring deterministic policies on the boundary of $\mathbb{J}$ differ by only one state-action pair. This ensures that we can efficiently find all neighboring policies of $\pi$ on both the boundary of $\mathbb{J}$ and the Pareto front by searching within the $\Pi_1(\pi)$.

2. **Sufficiency of Traversing Over Edges:** We show that for any vertex on the Pareto front, at least one edge on the Pareto front connects it to another Pareto-optimal vertex. This property guarantees that we can traverse the entire Pareto front by exploring neighboring deterministic policies.

3. **Locality Property of the Pareto front:** We establish that the faces of the Pareto front intersecting at a deterministic policy can be found by computing the convex hull of the returns of this deterministic policy and non-dominated deterministic policies that differ by one state-action pair. This ensures efficient discovery of Pareto front faces through local convex hull computation, reducing the computational overhead.

### 4.1 DISTANCE-ONE PROPERTY ON BOUNDARY OF $\mathbb{J}$

We present a theorem stating that the endpoints of any edge on the boundary of $\mathbb{J}$ correspond to deterministic policies that differ by only one state-action pair, almost surely. Specifically, for a finite MDP $(\mathcal{S}, \mathcal{A}, \mathbf{P}, \mathbf{r}, \gamma)$, where $\mathbf{P}$ is the transition probability kernel and $\mathbf{r} \geq 0$ is the reward function, the set of $\mathbf{P}$ and $\mathbf{r}$ for which deterministic policies on the edge of the Pareto front differ by more than one state-action pair has Lebesgue measure zero. The proof of this theorem is provided in Appendix D.

**Theorem 1.** *In a discounted finite MO-MDP $(\mathcal{S}, \mathcal{A}, \mathbf{P}, \mathbf{r}, \gamma)$ and under Assumption 1, any edge on the boundary of $\mathbb{J}$ connecting two deterministic policies corresponds to policies that differ by only one state-action pair, almost surely.*

Since the Pareto front lies on the boundary of $\mathbb{J}$, this theorem guarantees that for any policy $\pi$ on the Pareto front, the neighboring policies on the Pareto front can be found by searching among $\Pi_1(\pi)$, a set of policies that differ from $\pi$ by only one state-action pair. This significantly reduces the search space, making it more efficient to traverse the Pareto front.

**Proof Sketch** The proof follows two key steps: (1) First, we show that any face on the boundary of $\mathbb{J}$ is formed by the long-term returns of policies that are the convex combinations of the deterministic policies corresponding to the vertices of the face. Specifically, any point on an edge between two deterministic policies is the long-term return of a convex combination of these policies. (2) Next, we establish that if two deterministic policies differ by only one state-action pair, the long-term returns of their convex combinations form a straight line. In contrast, if two deterministic policies differ by more than one state-action pair, the long-term returns of their convex combinations do not form a straight line between them for almost all MDPs.

By combining these two observations, we conclude that the endpoints of any edge on the boundary of $\mathbb{J}$ correspond to deterministic policies differing by exactly one state-action pair almost surely. Since the Pareto front lies on the boundary of $\mathbb{J}$, it follows that neighboring policies on the Pareto front differ by only one state-action pair. This completes the proof.

## 4.2 Sufficiency of Traversing Over Edges

From the previous section, we have the properties of the neighboring policies on the Pareto front. Hence, a natural approach would be to traverse all vertices of the Pareto front by moving along the edges connecting them. However, this approach relies on the assumption that the vertices on the Pareto front are connected. If the vertices are not connected, we risk being stuck at a single vertex on the Pareto front without a way to reach other unconnected vertices. To address this, we present Lemma 2 showing that the Pareto front of $\mathbb{J}$ is connected (proof is provided in Appendix E).

**Lemma 2.** *(Existence of Neighboring Edges on the Pareto Front of $\mathbb{J}$) Suppose $\mathbb{J}$ contains multiple Pareto optimal vertices. Let $J^{\pi_1}$ be a Pareto optimal vertex on $\mathbb{J}$. Let $J_N(\pi_1, \mathbb{J}) = \{J^\pi \mid \pi \in \Pi_N(\pi_1, \mathbb{J})\}$ represent all neighboring vertices of $J^{\pi_1}$ on $\mathbb{J}$. Then, there exists at least one neighboring vertex $J \in J_N(\pi_1, \mathbb{J})$ such that the edge connecting $J$ and $J^{\pi_1}$ lies on the Pareto front.*

**Proof Sketch** We prove Lemma 2 by contradiction. Assume there exists a vertex on the convex polytope $\mathbb{J}$ that is not connected by an edge lying on the Pareto front. In this case, we show that this vertex would dominate all other points on the polytope, implying that it is the only Pareto optimal point, which leads to a contradiction. Therefore, if the Pareto front contains more than one deterministic Pareto optimal policy, the vertex must be connected to at least one edge on the Pareto front. This completes the proof.

Lemma 2 only shows that a Pareto optimal vertex is connected to an edge (1-dimensional) on the Pareto front. However, it does not guarantee that one of the neighboring facets of a Pareto optimal vertex must also lie on the Pareto front. A facet of $\mathbb{J}$ lies on the Pareto front if and only if its normal vector is strictly positive (which will be shown later in Lemma 3), meaning that all points on the facet are not dominated by any other points in $\mathbb{J}$. However, the normal vectors of all facets intersecting at a Pareto optimal vertex are not necessarily strictly positive. This observation is consistent with Fig. 3, where point A is a Pareto optimal vertex, but it only lies on a Pareto optimal edge AB, and all its incident faces are not Pareto optimal. This also justifies that constructing the Pareto front in a 2D case (with two reward objectives) is relatively simple, as we only need to connect the neighbor vertices. However, in higher dimensions, the Pareto front becomes more complex, and it is impossible to build the Pareto front purely from vertices.

## 4.3 Locality Property of the Pareto front

Previous theoretical results show that it is possible to traverse the Pareto front through edges constructed by deterministic policies that differ by only one state-action pair. However, each policy has $S \times (A - 1)$ neighboring policies, and the challenge lies in efficiently selecting the Pareto optimal policies among them and constructing the Pareto front based on these neighbors.

This section provides the theoretical foundations for retrieving the boundary of $\mathbb{J}$ from the convex hull constructed by a vertex and the neighboring policies that differ by only one state-action pair. Furthermore, we show that the Pareto front can be retrieved by (1) constructing the convex hull using the vertex and locally non-dominated policies, and (2) applying Lemma 3 to extract all Pareto optimal faces from the convex hull.

Let deterministic policy $\pi$ correspond to a vertex on $\mathbb{J}$. We denote the set of neighbors on the boundary of $\mathbb{J}$ to $J^\pi$ as $\mathrm{N}(J^\pi, \mathbb{J})$, and the set of deterministic policies whose long-term returns make up $\mathrm{N}(J^\pi, \mathbb{J})$ as $\mathrm{N}_\pi(J^\pi, \mathbb{J})$. Let $\mathbb{F}(J^\pi, \mathbb{J})$ and $\mathbb{F}(J^\pi, \mathrm{Conv}(J^{\Pi_1} \cup \{J^\pi\})$ be the set of incident faces of $J^\pi$ on $\mathbb{J}$ and $\mathrm{Conv}(J^{\Pi_1} \cup \{J^\pi\})$, respectively.

**Theorem 2.** *The neighboring vertices and faces of $J^\pi$ on the convex hull $\mathrm{Conv}(J^{\Pi_1} \cup \{J^\pi\})$ are the same as those on $\mathbb{J}$. Formally, $\mathrm{N}(J^\pi, \mathrm{Conv}(J^{\Pi_1} \cup \{J^\pi\})) = \mathrm{N}(J^\pi, \mathbb{J})$, $\mathrm{N}_\pi(J^\pi, \mathrm{Conv}(J^{\Pi_1} \cup \{J^\pi\})) = \mathrm{N}_\pi(J^\pi, \mathbb{J})$, and $\mathbb{F}(J^\pi, \mathrm{Conv}(J^{\Pi_1} \cup \{\{J^\pi\})) = \mathbb{F}(J^\pi, \mathbb{J})$.*

The proof is provided in Theorem 6 and Proposition 3 of Appendix F. This theorem guarantees that we can find all neighboring vertices and faces on the boundary of $\mathbb{J}$ from the local convex hull $\mathrm{Conv}(J^{\Pi_1} \cup \{J^\pi\})$. By iterating through the process of constructing the convex hull of all distance-one policies and retaining the neighboring faces, we can eventually traverse the entire boundary of $\mathbb{J}$.

We also establish key properties of the Pareto front on a convex polytope (proof is provided in Appendix G).

**Lemma 3.** *Given a convex polytope $\mathbb{C}$, let $F$ be face of $\mathbb{C}$ defined as the intersection of $n$ facets, i.e., $F = \cap_{i=1}^n F_i$, where $n \geq 1$ and each facet is of the form $F_i = \mathbb{C} \cap \{\boldsymbol{x} \in \mathbb{R}^d : \boldsymbol{w}_i^\top \boldsymbol{x} = c_i\}$. Then the face $F$ lies on the Pareto front of $\mathbb{C}$ if and only if there exists a linear combination of the normal vectors, weighted by $\boldsymbol{\alpha}_i \geq 0$ for any $i \in \{1, \cdots, n\}$, such that $\sum_i \boldsymbol{\alpha}_i [\boldsymbol{w}_i]_j > 0$ where $[\boldsymbol{w}_i]_j$ is the $j$-th component of $\boldsymbol{w}_i$.*

Based on Lemma 3, we can also extract the Pareto optimal faces from the local convex hull. Let $\mathbb{F}(J^\pi, \mathbb{P}(\mathbb{J}))$ be the set of incident faces of $J^\pi$ on the Pareto front $\mathbb{P}(\mathbb{J})$. The proof is shown in Proposition 6 of Appendix F.

**Proposition 1.** *Let $\mathbb{F}_P(J^\pi, \mathrm{Conv}(J^{\Pi_{1,ND}} \cup \{J^\pi\}))$ denote the incident faces of $J^\pi$ on the convex hull $\mathrm{Conv}(J^{\Pi_{1,ND}} \cup \{J^\pi\})$ that satisfy the conditions of Lemma 3. Then,*

$$\mathbb{F}(J^\pi, \mathbb{P}(\mathbb{J})) = \mathbb{F}_P(J^\pi, \mathrm{Conv}(J^{\Pi_{1,ND}} \cup \{J^\pi\})).$$

This proposition demonstrates that the faces of the Pareto front that intersect with a vertex are equivalent to the Pareto optimal faces of the local convex hull constructed by the vertex and its neighboring vertices. Therefore, we conclude that all faces of the Pareto front that intersect with a vertex can be identified by computing the convex hull of neighboring non-dominated policies and then applying the Pareto front criterion in Lemma 3, which corresponds to Eq. (1), to extract the Pareto optimal faces from the local convex hull.

## 5 ALGORITHM EVALUATION

In this section, we empirically evaluate the performance of our algorithm in discovering the full Pareto front, including the vertices that correspond to deterministic Pareto optimal policies in MO-MDPs. To the best of our knowledge, this is the first method capable of precisely identifying the complete Pareto front without traversing the preference vector space. We start with giving an example where the shape of the Pareto front necessitates identifying the entire front, rather than focusing solely on deterministic policies. Then we compare our algorithm with existing methods in terms of efficiency in finding both the full Pareto front and the set of deterministic Pareto optimal policies.

We demonstrate a Pareto front structure that is not immediately evident from the vertices alone even in a simple setting. We consider a basic MDP setting, where we only have 4 states and 3 actions, and the reward dimension is 3. The transition kernel is uniformly distributed between 0 and 1 and normalized to sum to 1, while the reward functions are uniformly distributed between 0 and 1. The discount factor is 0.9. As illustrated in Fig. 4, the Pareto front is more complex than simply being the convex hull of its vertices. For instance, A is a point on the Pareto front, but none of its incident facets are on the Pareto front. To find an edge connecting A to another policy on the Pareto front, we must further examine which edge that connects point A and other points corresponding to deterministic Pareto optimal policies is also on the Pareto front by Lemma 3. This edge is difficult to derive using only the knowledge of deterministic policies on the Pareto front.

### 5.1 ABILITY AND EFFICIENCY IN FINDING THE PARETO FRONT

As the Pareto front searching algorithm is the first to recover the entire precise Pareto front, we provide a benchmark algorithm for comparison to demonstrate both the ability and efficiency of our method in identifying all faces of the true Pareto front. The benchmark algorithm retrieves the Pareto front in two steps: (1) computing the convex hull of all non-dominated deterministic policies and (2) applying Algorithm 2 to identify the Pareto front. Details are shown in Algorithm 4.

Since the number of deterministic policies increases exponentially with the number of states, and the complexity of calculating the convex hull is proportional to the number of input points (Barber et al., 1996), the benchmark algorithm's

complexity grows exponentially with the number of actions. Therefore, we compare the performance of our algorithm against the benchmark in small MDP settings. Specifically, we consider state spaces of size $5$ and $8$, action spaces ranging from 5 to 7, and a reward dimension of 3. Our algorithm successfully identifies all faces of the true Pareto front in all cases. As shown in Fig. 5, the runtime of the benchmark algorithm increases exponentially with the number of actions, making it impractical even in moderately sized action spaces. In contrast, our algorithm maintains a controllable runtime by efficiently traversing edges on the Pareto front to identify the Pareto faces.

## 5.2 EFFICIENCY IN FINDING THE PARETO FRONT VERTICES

We compare the efficiency of finding all vertices on the Pareto front (i.e., deterministic Pareto optimal policies) with OLS (Roijers, 2016). For each iteration, OLS solves the single-objective MDP for a given preference vector and calculates new preference vectors by constructing a convex hull of all candidate preference vectors. In contrast, our algorithm requires only a single single-objective planning step during the initialization phase to obtain the initial deterministic Pareto-optimal policy. Rather than constructing a global convex hull encompassing all potential Pareto-optimal policies, our algorithm builds the convex hull locally based on the vertex and its neighboring policies. As a result, our algorithm achieves significantly lower computational complexity compared to OLS. A detailed complexity comparison is provided in Appendix H.

We compare the running time of our algorithm against OLS in terms of efficiency for finding all vertices on the Pareto front. Specifically, we consider scenarios where the state space size is $5$ and $8$, and the reward dimension is $4$. As shown in Fig. 6, while both OLS and the proposed algorithm efficiently find all vertices when the action and state space are small, the running time for OLS increases significantly with even slight expansions in the state and action space. In contrast, our proposed algorithm maintains a more manageable running time.

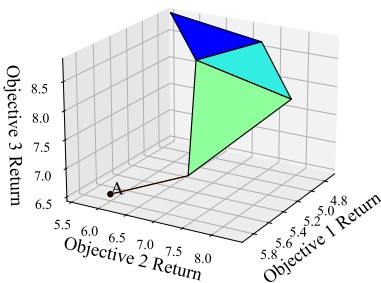

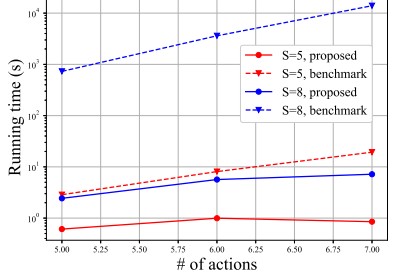

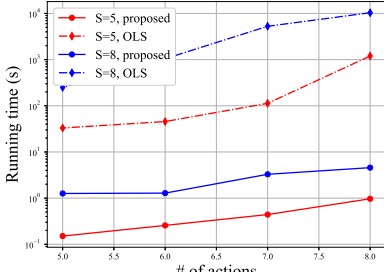

Figure 4: Pareto front of a simple MDP with $S = 4$, $A = 3$, and $D = 3$.

Figure 5: Comparison between the proposed Pareto front searching algorithm and the benchmark algorithm when $D = 3$.

Figure 6: Comparison between the proposed deterministic Pareto optimal policies searching algorithm and the OLS algorithm when $D = 4$.

## 6 CONCLUSION

This paper investigated the geometric properties of the Pareto front in MO-MDPs. We proved that any neighboring deterministic policies on the Pareto front differ in exactly one state-action pair almost surely. We also demonstrated that the Pareto front is continuous in MO-MDPs, enabling traversing the Pareto front by moving along edges. Building on these theoretical insights, we proposed an efficient Pareto front searching algorithm, which was validated through experiments. The algorithm effectively explores the Pareto front by leveraging the structure of the deterministic policies.

ACKNOWLEDGMENTS

This work has been supported in part by the U.S. National Science Foundation under the grants: NSF AI Institute (AI-EDGE) 2112471, CNS-2312836, and was sponsored by the Army Research Laboratory under Cooperative Agreement Number W911NF-23-2-0225. The views and conclusions contained in this document are those of the authors and should not be interpreted as representing the official policies, either expressed or implied, of the Army Research Laboratory or the U.S. Government. The U.S. Government is authorized to reproduce and distribute reprints for Government purposes notwithstanding any copyright notation herein.

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

## A    DISCUSSION ON PARETO OPTIMALITY

We define the Pareto optimal policy set based on an initial state distribution $\mu$. Unlike single-objective reinforcement learning, where the optimal policy maximizes the long-term return for all states under any initial state distribution, Pareto optimality with respect to a specific initial state distribution $\mu$ does not ensure Pareto optimality across all states (Lu et al., 2022). Lu et al. (2022) introduces the concept of aggregate Pareto efficiency, where a policy $\pi$ is considered aggregate Pareto optimal if there exists no policy $\pi'$ such that $\mathbf{V}^{\pi'} \succ \mathbf{V}^{\pi}$. The Pareto front $\mathbb{P}(\Pi)$ defined in Section 3.1 only forms a subset of the aggregate Pareto optimal set.

We focus on the definition of Pareto optimality over a given state distribution for two key reasons:

1. **Relevance to Practical Applications**: In real-world applications, the starting state often follows a specific distribution. For example, in RLHF applications in Large Language Models (LLM), input texts typically follow a certain distribution. Thus, it is unnecessary to optimize for Pareto efficiency over every individual state and objective, as this would not align with the natural distribution of inputs in such settings.

2. **Convergence to Aggregate Pareto Optimality**: As the discount factor $\gamma \to 1$, $\mathbb{P}(\Pi)$ approaches the aggregate Pareto optimal set (Lu et al., 2022). Therefore, in settings with a high discount factor, focusing on $\mathbb{P}(\Pi)$ provides a close approximation to the aggregate Pareto optimal set.

## B    NOTATIONS

Let $\mathbf{I}_M$ denote the identity matrix of size $M$, $\mathbf{1}_M$ denote the $M$-dimensional vector where all elements are 1, and $\mathbf{0}_{M \times N}$ denote a zero matrix with dimensions $M \times N$. Let $\mathbf{A}^\top$ denote the transpose of the matrix $\mathbf{A}$. Let $\mathbf{A} > (\geq)\mathbf{B}$ denote that each element of $\mathbf{A}$ is strictly greater than (greater than or equal to) the corresponding element of $\mathbf{B}$. Given an MDP $(\mathcal{S}, \mathcal{A}, \mathbf{P}, \boldsymbol{r}, \gamma)$, for any policy $\pi$, define the state transition probability matrix $\mathbf{P}^\pi$ and the expected reward vector $\boldsymbol{r}^\pi$ as:

$$\mathbf{P}^\pi(s, s') \coloneqq \sum_{a \in \mathcal{A}} \mathbf{P}(s'|s, a)\pi(a|s), \quad \boldsymbol{r}^\pi(s) \coloneqq \sum_{a \in \mathcal{A}} \boldsymbol{r}(s, a)\pi(a|s).$$

We provide the notation table as follows.

Table 1: Notation table

| Notation | Description |
|---|---|
| $\Pi$ | Set of stationary policies |
| $\Pi_D$ | Set of deterministic stationary policies |
| $J^\pi(\mu)$ | Long-term return of policy $\pi$ given initial state distribution $\mu$ |
| $J^U(\mu)$ | Set of long-term returns of all stationary policies in $U$ with initial distribution $\mu$ |
| $\mathbb{J}(\mu)$ | Set of long-term returns of all stationary policies with initial distribution $\mu$ |
| $\mathbb{P}(\mathbb{S})$ | Pareto front of the set $\mathbb{S}$ |
| $\mathbb{V}(\mathbb{C})$ | Set of vertices of the polytope $\mathbb{C}$ |
| $\mathbb{B}(\mathbb{C})$ | Boundary of the polytope $\mathbb{C}$ |
| $\mathbb{P}(\Pi)$ | Set of Pareto optimal policies $\{\pi \in \Pi \mid J^\pi \in \mathbb{P}(\mathbb{J}(\mu))\}$ |
| $\mathrm{Conv}(\mathbb{S})$ | Convex hull of the set $\mathbb{S}$ |
| $\mathrm{N}(\boldsymbol{x}, \mathbb{C})$ | Set of neighboring vertices of vertex $\boldsymbol{x}$ on $\mathbb{C}$ |
| $\mathbb{F}(\boldsymbol{x}, \mathbb{C})$ | Set of incident faces of vertex $\boldsymbol{x}$ on $\mathbb{C}$ |
| $\Pi_1(\pi)$ | Set of deterministic policies differing from $\pi$ by exactly one state-action pair |
| $\Pi_{1,\mathrm{ND}}(\pi)$ | Set of non-dominated policies among $\Pi_1(\pi)$ |
| $\Pi_N(\pi, \mathcal{M})$ | Set of deterministic policies that differ from $\pi$ at all states in $\mathcal{M}$ and matching $\pi$ at all other states |

## C  Supporting Lemmas and Theorems

In this section, for the sake of completeness in the paper, we present some well-known lemmas and theorems, along with a few easily proven lemmas, which will be used in the subsequent proofs.

We begin by presenting some well-known theorems and lemmas in the field of linear algebra.

**Theorem 3.** *(Rank-nullity theorem) Let* $\mathbf{A}$ *be an* $m \times n$ *matrix. Then, the following relationship holds:* $\operatorname{rank}(\mathbf{A}) + \dim(\operatorname{null}(\mathbf{A})) = n$, *where* $\operatorname{rank}(\mathbf{A})$ *is the dimension of the column space of* $\mathbf{A}$, *and* $\operatorname{null}(\mathbf{A})$ *is the null space of* $\mathbf{A}$.

**Lemma 4.** *Let* $\mathbf{A}$ *be an* $m \times n$ *matrix and* $\boldsymbol{b} \neq \mathbf{0}$ *be an* $m$-*dimensional vector. If* $\mathbf{A}\boldsymbol{x} = \boldsymbol{b}$ *has at least one solution, then the dimension of the solution space of* $\mathbf{A}\boldsymbol{x} = \boldsymbol{b}$ *is equal to* $\dim(\operatorname{null}(\mathbf{A}))$, *where* $\operatorname{null}(\mathbf{A})$ *is the null space of* $\mathbf{A}$.

**Lemma 5.** *Let* $\mathbf{A}$ *be an* $m \times n$ *matrix with full column rank. If* $\mathbf{AB} = k\mathbf{AC}$, *then* $\mathbf{B} = k\mathbf{C}$.

*Proof.* From $\mathbf{AB} = k\mathbf{AC}$, we can subtract $k\mathbf{AC}$ from both sides to get $\mathbf{A}(\mathbf{B} - k\mathbf{C}) = \mathbf{0}_m$. Since $\mathbf{A}$ has full column rank, it follows from Theorem 3 that the dimension of its null space must be zero, and the only solution is $\mathbf{A}\boldsymbol{x} = \mathbf{0}_m$ is $\boldsymbol{x} = \mathbf{0}_n$. Therefore, the only solution to $\mathbf{A}(\mathbf{B} - k\mathbf{C}) = \mathbf{0}_m$ is $\mathbf{B} - k\mathbf{C} = \mathbf{0}_n$. Hence, $\mathbf{B} = k\mathbf{C}$. $\qquad\square$

**Lemma 6.** *([Horn & Johnson, 2012](#)) Let* $\mathbf{A} \in \mathbb{R}^{n \times n}$ *be a square matrix. The maximum absolute value of the eigenvalues of* $\mathbf{A}$, *denoted as* $\rho(\mathbf{A})$, *satisfies the following inequality:*

$$\rho(\mathbf{A}) \leq \|\mathbf{A}\|_\infty,$$

*where* $\|\mathbf{A}\|_\infty = \max_{1 \leq i \leq n} \sum_{j=1}^n |A_{ij}|$. *is the infinity norm of the matrix* $\mathbf{A}$.

Then we present some lemmas related to the multi-objective optimization.

**Lemma 7.** *([Boyd & Vandenberghe, 2004](#)) Let* $\mathbb{S}$ *denote the set of achievable objective values and let* $\mathbb{P}(\mathbb{S})$ *denote the set of Pareto optimal values in* $\mathbb{C}$. *Then* $\mathbb{P}(\mathbb{S}) \subseteq \mathbb{B}(\mathbb{S})$, *i.e., every Pareto optimal value lies in the boundary of the set of achievable objective values.*

**Lemma 8.** *Let* $\boldsymbol{x}$ *be a point in a set* $\mathbb{S}$. $\boldsymbol{x}$ *lies on the Pareto front of* $\mathbb{S}$ *if and only if there exists a vector* $\boldsymbol{w} > 0$ *such that* $\boldsymbol{w}^\top(\boldsymbol{x} - \boldsymbol{y}) \geq 0$ *for all* $\boldsymbol{y} \in \mathbb{S}$.

*Proof.* ($\Longleftarrow$) Assume, for the sake of contradiction, that $\boldsymbol{x}$ is not on the Pareto front of $\mathbb{S}$. This means there exists a point $\boldsymbol{y} \in \mathbb{S}$ that strictly dominates $\boldsymbol{x}$, i.e., $\boldsymbol{y} \succ \boldsymbol{x}$. This implies that $\boldsymbol{y}$ is at least as large as the corresponding objective of $\boldsymbol{x}$ and $\boldsymbol{y}$ is strictly better than $\boldsymbol{x}$ in at least one objective. Therefore, given $\boldsymbol{w} > 0$, we have $\boldsymbol{w}^\top(\boldsymbol{x} - \boldsymbol{y}) < 0$. However, this contradicts the assumption that $\boldsymbol{w}^\top(\boldsymbol{x} - \boldsymbol{y}) \geq 0$ for all $\boldsymbol{y} \in \mathbb{S}$. Thus, no point in $\mathbb{S}$ strictly dominates $\boldsymbol{x}$, which implies that $\boldsymbol{x}$ is on the Pareto front.

($\Longrightarrow$) If $\boldsymbol{x}$ is Pareto optimal, then for any $\boldsymbol{y} \in \mathbb{S}$, $\boldsymbol{x}$ is at least as large as the corresponding objective of $\boldsymbol{y}$ and $\boldsymbol{x}$ is strictly better than $\boldsymbol{y}$ in at least one objective. Given $\boldsymbol{w} > 0$, $\boldsymbol{w}^\top(\boldsymbol{x} - \boldsymbol{y}) \geq 0$. $\qquad\square$

Next, we restate some definitions in the main body of the paper and present several lemmas related to polytopes.

**Definition 1.** *([Ziegler, 2012](#)) Let* $\mathbb{C} \subseteq \mathbb{R}^d$ *be a convex polytope. A linear inequality* $\boldsymbol{w}^\top x \leq c_0$ *is valid for* $\mathbb{C}$ *if it is satisfied for all points* $x \in \mathbb{C}$. *A face of* $\mathbb{C}$ *is any set of the form* $\mathbb{F} = \mathbb{C} \cap \{x \in \mathbb{R}^d : \boldsymbol{w}^\top x = c_0\}$ *where* $\boldsymbol{w}^\top \boldsymbol{x} \leq c_0$ *is a valid inequality for* $\mathbb{C}$. *The dimension of a face is the dimension of its affine hull, i.e.,* $\dim(F) := \dim(\operatorname{aff}(F))$. *The faces of dimension smaller than* $\dim(\mathbb{C})$ *are called proper faces. The faces of* $0$, $1$, $\dim(\mathbb{C}) - 2$, $\dim(\mathbb{C}) - 1$ *are called vertices, edges, ridges, and facets, respectively.*

**Lemma 9.** *([Grünbaum, 2013](#)) Every nonempty proper face of a convex polytope* $\mathbb{C}$ *is an intersection of facets of* $\mathbb{C}$.

**Lemma 10.** *Given a* $d$-*dimensional convex polytope, if the optimization objectives are the individual coordinate values, then the Pareto front of the convex polytope is a continuous set.*

Finally, we give some lemmas related to MDP.

**Lemma 11.** *([Altman, 2021](#)) The occupancy measure of any policy* $\pi$ *and initial distribution* $\mu$ *is defined as* $d_\mu^\pi(s) := (1-\gamma)\sum_{t=0}^\infty \gamma^t P(s_t = s|s_0 \sim \mu(\cdot))$. *Similarly, define* $\bar{d}_\mu^\pi(s,a) := (1-\gamma)\sum_{t=0}^\infty \gamma^t P(s_t = s, a_t = a|s_0 \sim \mu(\cdot)) = d_\mu^\pi(s)\pi(a|s)$. *Define for any set of policies* $U$, $L^U(\mu) := \cup_{\pi \in U}\{\bar{d}_\mu^\pi\}$ *and define* $L(\mu) := \cup_{\pi \in \Pi}\{\bar{d}_\mu^\pi\}$. *Then,* $L(\mu)$ *is closed convex polytope and* $L(\mu) = \operatorname{Conv}(L^{\Pi_D}(\mu))$.

**Lemma 12.** *([Sutton & Barto, 2018](#)) Given MDP* $(\mathcal{S}, \mathcal{A}, \mathbf{P}, \boldsymbol{r}, \gamma)$, *there exists at least one policy that is always better than or equal to all other policies in all states, denoted as optimal policy. The optimal policy* $\pi^*$ *satisfies* $\pi^*(s) = \arg\max_{a \in \mathcal{A}} Q^*(s,a)$, *where* $Q^*(s,a)$ *is the solution to the Bellman optimality equation* $Q^*(s,a) = \mathbb{E}\left[\boldsymbol{r}(s,a) + \gamma \max_{a'} Q^*(s',a') \mid s,a\right]$.

**Lemma 13.** *If there exists $\mu$ where $\mu(s) > 0$ for any $s \in \mathcal{S}$ such that a deterministic policy $\pi^*$ maximizes for $J^{\pi^*}(\mu) = \mu^\top \mathbf{V}^{\pi^*}$, then $\pi^*$ is the optimal policy, i.e., $\mathbf{V}^{\pi^*}(s) = \max_\pi \mathbf{V}^\pi(s)$ for any $s$.*

*Proof.* We prove this by contradiction. If $\pi^*$ is not the optimal policy, and there exists optimal policy $u \neq \pi^*$ such that $V^{\pi^*}(s) \leq V^u(s)$ for any $s \in \mathcal{S}$, and on at least one state $V^{\pi^*}(s) < V^u(s)$. Then $J^u(\mu) > J^{\pi^*}(\mu)$ since $\mu(s) > 0$ for any $s$. This contradicts the condition that $\pi^*$ maximizes for $J^\pi(\mu)$. So $\pi^*$ is also an optimal policy. $\qquad\square$

**Lemma 14.** *Given a single-objective MDP $(\mathcal{S}, \mathcal{A}, \mathbf{P}, r, \gamma)$. If $\Pi^*$ is the set of the optimal deterministic policies, then the set of all optimal policies (including stochastic and deterministic policies) is the convex hull constructed by $\Pi^*$.*

*Proof.* We first show that **all policies in the convex hull constructed by the set of optimal deterministic policies $\Pi^*$ are optimal policies**. By Lemma 12, the optimal policies share the same state-action value function, that is, $Q^{\pi_i}(s,a) = Q^{\pi_j}(s,a) = Q^*(s,a)$ for any $\pi_i, \pi_j \in \Pi^*$. Since $\pi_i \in \Pi^*$ are deterministic policies, with a little abuse notation of $\pi$, we have $\pi_i(s) = \arg_{a \in \mathcal{A}} \max Q^*(s,a)$ for any $i$. Let $\boldsymbol{\alpha}$ be a $(|\Pi| - 1)$-dimension standard simplex. Let $\pi$ be the convex combination of $\Pi^*$, that is, $\pi(a|s) = \sum_i \boldsymbol{\alpha}_i \pi_i(a|s)$ for any $s$. As each possible choice of $\pi$ at state $s$ maximizes $Q^*(s,a)$, i.e., $\pi_i(\arg_{a \in \mathcal{A}} \max Q^*(s,a)|s) = 1$, $\pi$ at state $s$ maximizes $Q^*(s,a)$ and $\pi$ is an optimal policy. Hence any convex combination $\pi$ optimal deterministic policies are also optimal.

Then we want to show **for any policies not on the convex hull the set of optimal deterministic policies $\Pi$, it cannot be the optimal policy**. We prove this by contradiction. Suppose an optimal policy $\pi$ is not on the convex hull by all deterministic policies. In that case, it is written as $\pi = \sum_{i, \pi_i \in \Pi^*} \alpha_i \pi_i + \sum_{j, \pi_j \in \bar{\Pi}} \beta_j \pi_j$, where $\sum_i \alpha_i + \sum_i \beta_j = 1$, $\alpha_i \geq 0$, $\beta_j \geq 0$, and $\bar{\Pi}$ is a set of non-optimal deterministic policies. Suppose on state $s$ such that $V^{\pi_j}(s) \leq V^*(s)$, then $\pi(s) \neq \arg\max^\pi Q^*(s,a)$, which indicates $\pi$ is no longer an optimal policy. This contradicts the assumption that $\pi$ is an optimal policy. $\qquad\square$

**Lemma 15.** *Given an MDP $(\mathcal{S}, \mathcal{A}, \boldsymbol{P}, \boldsymbol{r}, \gamma)$ and a deterministic policy $\pi : \mathcal{S} \to \mathcal{A}$, let $\pi \in \Pi_N(\pi_1, \mathcal{M})$. The transition matrix $\mathbf{P}^\pi$ and reward vector $\boldsymbol{r}^\pi$ can be decomposed into block matrices as follows:*

$$\mathbf{P}^\pi = \begin{bmatrix} \mathbf{A} & \mathbf{B} \\ \mathbf{C}^\pi & \mathbf{D}^\pi \end{bmatrix}, \quad \boldsymbol{r}^\pi = \begin{bmatrix} \boldsymbol{X} \\ \boldsymbol{Y}^\pi \end{bmatrix}.$$

*Following the definitions of $\Gamma$ and $\mathbf{V}$ in Eq. (9) we have*

$$\Gamma^\pi = \begin{bmatrix} \gamma(\mathbf{I}_{S-M} - \gamma\mathbf{A})^{-1}\mathbf{B} \\ \mathbf{I}_M \end{bmatrix}, \quad \tilde{\mathbf{V}}^\pi = \begin{bmatrix} (\mathbf{I}_{S-M} - \gamma\mathbf{A})^{-1}\boldsymbol{X} \\ \mathbf{0}_M \end{bmatrix}.$$

*Proof.* Since the transition probabilities and rewards starting from $\mathcal{S} \setminus \mathcal{M}$ are the same for any $\pi \in \Pi_N(\pi_1, \mathcal{M})$, we conclude that $\mathbf{A}$, $\mathbf{B}$, and $\boldsymbol{X}$ (the relevant blocks of $\mathbf{P}^\pi$ and $\boldsymbol{r}^\pi$) are independent of $\pi$. However, $\mathbf{C}^\pi$, $\mathbf{D}^\pi$, and $\boldsymbol{Y}^\pi$ vary with $\pi$, as indicated by the subscripts.

**Analysis of $\Gamma$**  We decompose $\Gamma^\pi$ into block matrix form, i.e., $\Gamma^\pi = \left[ (\Gamma_1^\pi)^\top, (\Gamma_2^\pi)^\top \right]^\top$, where $\Gamma_1^\pi \in \mathbb{R}^{(S-M) \times M}$ and $\Gamma_2^\pi \in \mathbb{R}^{M \times M}$.

The definition of $\mathbf{H}(\bar{s}_i, \mathcal{M}, \bar{s}_j)$ that for any $\bar{s}_i, \bar{s}_j \in \mathcal{M}$ with $i \neq j$ shows that $\mathbf{H}(\bar{s}_i, \bar{s}_i) = 0$ with probability 1 and $\mathbf{H}(\bar{s}_i, \bar{s}_j) = \infty$ with probability 1. Thus,

$$[\Gamma_2^\pi]_{i,j} = \mathbb{E}\left[ \gamma^{\mathbf{H}(\bar{s}_i, \bar{s}_j)} \right] = 0 \quad \text{for } i \neq j, \quad \text{and} \quad [\Gamma_2]_{i,i} = \mathbb{E}\left[ \gamma^{\mathbf{H}(\bar{s}_i, \bar{s}_i)} \right] = 1.$$

For $\Gamma_1^\pi$, starting from $s_i \in \mathcal{S} \setminus \mathcal{M}$, the probability of reaching $\bar{s}_j \in \mathcal{M}$ in one step is $\mathbf{B}_{i,j}$. The probability of reaching $\bar{s}_j$ in two steps is $\mathbf{A}_{i,:}\mathbf{1}_{S-M}$. Repeating this reasoning for subsequent steps, we obtain:

$$\begin{aligned} [\Gamma_1^\pi]_{i,j} &= P(S^\pi(s_i, \mathcal{M}) = \bar{s}_j)\mathbb{E}\left[ \mathbb{E}\left[ \gamma^{\mathbf{H}(s_i, \bar{s}_j)} \right] \middle| S^\pi(s_i, \mathcal{M}) = \bar{s}_j \right] \\ &= \gamma\mathbf{B}_{i,j} + \gamma^2\mathbf{A}_{i,:}\mathbf{B}_{:,j} + \gamma^3\mathbf{A}_{i,:}\mathbf{A}\mathbf{B}_{:,j} + \cdots \end{aligned}$$

Since $\|\gamma\mathbf{A}\|_\infty < 1$, it follows from Lemma 6 that the eigenvalues of $\gamma\mathbf{A}$ are smaller than 1. Therefore, $\mathbf{I}_{S-M} - \gamma\mathbf{A}$ does not have any zero eigenvalues, implying that it is invertible. Summing over all steps, we have:

$$\Gamma_1^\pi = \gamma\sum_{n=0}^\infty (\gamma\mathbf{A})^n\mathbf{B} = \gamma(\mathbf{I}_{S-M} - \gamma\mathbf{A})^{-1}\mathbf{B}.$$

Thus, we conclude:

$$\Gamma^\pi = \begin{bmatrix} \Gamma_1^\pi \\ \Gamma_2^\pi \end{bmatrix} = \begin{bmatrix} \gamma(\mathbf{I}_{S-M} - \gamma\mathbf{A})^{-1}\mathbf{B} \\ \mathbf{I}_M \end{bmatrix}.$$

**Analysis of V** Similarly, we decompose $\mathbf{V}$ into block matrix form: $\tilde{\mathbf{V}}^\pi = \left[ \left(\tilde{\mathbf{V}}_1^\pi\right)^\top, \left(\tilde{\mathbf{V}}_2^\pi\right)^\top \right]^\top$, where $\tilde{\mathbf{V}}_1^\pi \in \mathbb{R}^{(S-M)\times D}$ and $\tilde{\mathbf{V}}_2^\pi \in \mathbb{R}^{M\times D}$.

For $\tilde{\mathbf{V}}_2^\pi$, starting from any $\bar{s} \in \mathcal{M}$, we reach $\mathcal{M}$ with probability 1 in zero steps, so the total reward is 0. Thus, $\tilde{\mathbf{V}}_2^\pi = \mathbf{0}_{M\times M}$.

Starting from state $s_i \in \mathcal{S} \setminus \mathcal{M}$, it has immediate reward $[\mathbf{X}]_{(i,\cdot)}$. For the next state, it has probability $[\mathbf{A}]_{i,j}$ to move to state $s_j \in S \setminus \mathcal{M}$, and it has probability $\sum_j [\mathbf{B}]_{i,j}$ to move to $\bar{s}_j \in \mathcal{M}$ which terminates the process with future return 0. Therefore, we can write the Bellman equation-like form:

$$\begin{aligned} \tilde{\mathbf{V}}_1^\pi =& \mathbf{X} + \gamma\mathbf{A}\tilde{\mathbf{V}}_1^\pi + \gamma\mathbf{B}\mathbf{0}_{(S-M)\times D} \\ =& \mathbf{X} + \gamma\mathbf{A}\tilde{\mathbf{V}}_1^\pi. \end{aligned}$$

Solving this equation gives:

$$\tilde{\mathbf{V}}_1^\pi = (\mathbf{I}_{S-M} - \gamma\mathbf{A})^{-1}\mathbf{X}.$$

Hence, we conclude:

$$\tilde{\mathbf{V}}^\pi = \begin{bmatrix} \tilde{\mathbf{V}}_1^\pi \\ \tilde{\mathbf{V}}_2^\pi \end{bmatrix} = \begin{bmatrix} (\mathbf{I}_{S-M} - \gamma\mathbf{A})^{-1}\mathbf{X} \\ \mathbf{0}_M \end{bmatrix}.$$

$\square$

**Lemma 16.** *(Properties of $\mathbf{F}^\pi$ and $\mathbf{E}^\pi$) The functions $\mathbf{F}^\pi$ and $\mathbf{E}^\pi$, defined in Eq. (12), can be expressed as follows:*

$$\mathbf{F}^\pi := \gamma\mathbf{P}^\pi(\mathcal{M})\Gamma,$$
$$\mathbf{E}^\pi := \mathbf{r}^\pi(\mathcal{M}) + \gamma\mathbf{P}^\pi(\mathcal{M})\tilde{\mathbf{V}},$$

*where $\Gamma$ and $\tilde{\mathbf{V}}$ are predefined matrices given in Lemma 15, $\mathcal{M}$ is a subset of states, and $\mathbf{P}^\pi(\mathcal{M})$ is the transition probability matrix under policy $\pi$ restricted to the subset $\mathcal{M}$.*

*We can derive the following properties:*

1. *Gradients of $\mathbf{F}^\pi$ and $\mathbf{E}^\pi$ with respect to $\pi$. For any $\bar{s} \in \mathcal{M}$ and $a \in \mathcal{A}$, the partial derivatives of $\mathbf{F}^\pi$ and $\mathbf{E}^\pi$ with respect to the policy $\pi$ are given by:*

$$\frac{\partial \mathbf{F}^\pi}{\partial \pi(\bar{s}, a)} = \gamma\mathbf{P}(\delta(\bar{s}), a, \cdot)\Gamma, \quad \frac{\partial \mathbf{E}^\pi}{\partial \pi(\bar{s}, a)} = \mathbf{r}(\delta(\bar{s}), a, \cdot) + \gamma\mathbf{P}(\delta(\bar{s}), a, \cdot)\tilde{\mathbf{V}},$$

   *where $\delta(\bar{s})$ is a unit vector indicating the state $\bar{s}$, $Z = \mathbf{P}(\delta(\bar{s}), a, \cdot) \in \mathbb{R}^{M\times S}$, with $Z(\bar{s}, \cdot) = \mathbf{P}(\bar{s}, a, \cdot)$, and $Z(\bar{s}', \cdot) = 0$ for any $\bar{s}' \neq \bar{s}$.*

2. *Simplification when $\pi$ is a deterministic policy. If $\pi$ is a deterministic policy, then $\mathbf{F}^\pi$ and $\mathbf{E}^\pi$ can be simplified as follows:*

$$\mathbf{F}^\pi = \sum_{\bar{s}\in\mathcal{M}} \gamma\mathbf{P}(\delta(\bar{s}), \pi(\bar{s}), \cdot)\Gamma,$$
$$\mathbf{E}^\pi = \sum_{\bar{s}\in\mathcal{M}} \left( \mathbf{r}(\delta(\bar{s}), \pi(\bar{s}), \cdot) + \gamma\mathbf{P}(\delta(\bar{s}), \pi(\bar{s}), \cdot)\tilde{\mathbf{V}} \right).$$

3. *Given $\pi = \alpha\pi_1 + (1-\alpha)\pi_2$, where $0 \le \alpha \le 1$ to ensure $\pi$ is a valid policy, we have*

$$\begin{aligned} \mathbf{F}^\pi =& \alpha\mathbf{F}^{\pi_1} + (1-\alpha)\mathbf{F}^{\pi_2}, \\ \mathbf{E}^\pi =& \alpha\mathbf{E}^{\pi_1} + (1-\alpha)\mathbf{E}^{\pi_2}. \end{aligned}$$

*Proof.* **(1)** Recall that the state transition matrix and reward function under policy $\pi$ are defined as follows:

$$\mathbf{P}^\pi(s, s') := \sum_{a\in\mathcal{A}} \mathbf{P}(s, a, s')\pi(s, a), \quad \mathbf{r}^\pi(s) = \sum_{a\in\mathcal{A}} \mathbf{r}(s, a)\pi(s, a),$$

where $\mathbf{P}(s, a, s')$ is the probability of transitioning from state $s$ to state $s'$ given action $a$, and $\mathbf{r}(s, a)$ is the reward for taking action $a$ in state $s$.

For any $\bar{s} \in \mathcal{M}$ and $a \in \mathcal{A}$, we compute the derivatives of $\mathbf{P}^\pi(s, s')$ and $\mathbf{r}^\pi(s)$ with respect to the policy $\pi(\bar{s}, a)$:

$$\frac{\partial \mathbf{P}^\pi(s, s')}{\partial \pi(\bar{s}, a)} = \mathbf{P}(s, a, s')\delta(s = \bar{s}), \quad \frac{\partial \mathbf{r}^\pi(s)}{\partial \pi(\bar{s}, a)} = \mathbf{r}(s, a)\delta(s = \bar{s}),$$

where $\delta(s = \bar{s})$ is the Kronecker delta function, which is 1 if $s = \bar{s}$ and 0 otherwise.

Next, substituting these expressions into the derivatives of $\mathbf{F}^\pi$ and $\mathbf{E}^\pi$ with respect to $\pi(\bar{s}, a)$, we obtain:

$$\frac{\partial \mathbf{F}^\pi}{\partial \pi(\bar{s}, a)} = \frac{\gamma \partial \mathbf{P}^\pi(s, s')}{\partial \pi(\bar{s}, a)}\Gamma = \gamma \mathbf{P}(\delta(\bar{s}), a, \cdot)\Gamma,$$

$$\frac{\partial \mathbf{E}^\pi}{\partial \pi(\bar{s}, a)} = \frac{\partial \mathbf{r}^\pi(s)}{\partial \pi(\bar{s}, a)} + \frac{\partial \gamma \mathbf{P}^\pi(s, s')}{\partial \pi(\bar{s}, a)}\tilde{V}$$

$$= \mathbf{r}(\delta(\bar{s}), a) + \gamma \mathbf{P}(\delta(\bar{s}), a, \cdot)\tilde{V}.$$

**(2)** When $\pi$ is a deterministic policy, we recall that $\pi$ maps each state $s \in \mathcal{S}$ to a specific action $\pi(s) \in \mathcal{A}$. In this case, with a slight abuse of notation $\pi$, the transition matrix and reward function are simplified as follows:

$$\mathbf{P}^\pi(s, s') = \sum_{a \in \mathcal{A}} \mathbf{P}(s, a, s')\pi(s, a) = \mathbf{P}(s, \pi(s), s'),$$

$$\mathbf{r}^\pi(s) = \sum_{a \in \mathcal{A}} \mathbf{r}(s, a)\pi(s, a) = \mathbf{r}(s, \pi(s)),$$

where the policy $\pi$ deterministically selects action $\pi(s)$ in each state $s$.

Substituting these expressions into the definitions of $\mathbf{F}^\pi$ and $\mathbf{E}^\pi$, we obtain the desired results:

$$\mathbf{F}^\pi = \sum_{\bar{s} \in \mathcal{M}} \gamma \mathbf{P}(\delta(\bar{s}), \pi(\bar{s}), \cdot)\Gamma,$$

$$\mathbf{E}^\pi = \sum_{\bar{s} \in \mathcal{M}} \left( \mathbf{r}(\delta(\bar{s}), \pi(\bar{s})) + \gamma \mathbf{P}(\delta(\bar{s}), \pi(\bar{s}), \cdot)\tilde{V} \right).$$

**(3)** Since the transition probability matrix and reward function under a mixed policy $\alpha\pi_1 + (1 - \alpha)\pi_2$ can be expressed as linear combinations of the respective components under policies $\pi_1$ and $\pi_2$, we have:

$$\mathbf{P}^{\alpha\pi_1 + (1-\alpha)\pi_2}(\mathcal{M}) = \alpha\mathbf{P}^{\pi_1}(\mathcal{M}) + (1 - \alpha)\mathbf{P}^{\pi_2}(\mathcal{M}),$$

and

$$\mathbf{r}^{\alpha\pi_1 + (1-\alpha)\pi_2}(\mathcal{M}) = \alpha\mathbf{r}^{\pi_1}(\mathcal{M}) + (1 - \alpha)\mathbf{r}^{\pi_2}(\mathcal{M}).$$

By applying these expressions in conjunction with the definitions of $\mathbf{F}^\pi$ and $\mathbf{E}^\pi$, the desired result follows. $\qquad \square$

## D DISTANCE-1 PROPERTY

Let $\Pi_{\text{all}}$ be the set of all policies, including both stationary and non-stationary policies, and let $\mathbb{J}_{\text{all}}(\mu) = \{J^\pi(\mu) | \pi \in \Pi_{\text{all}}\}$ denote the set of achievable long-term returns for all policies in $\Pi_{\text{all}}$. Recall that $\mathbb{J}(\mu) = \{J^\pi(\mu) | \pi \in \Pi\}$ represents the set of achievable long-term returns for all policies in the set of stationary policies $\Pi$. By Theorem 3.1 of Altman (2021), it is sufficient to represent the $\mathbb{J}(\mu)$ using only stationary policies, i.e., $\mathbb{J}_{\text{all}}(\mu) = \mathbb{J}(\mu)$. Therefore, we focus on the set of stationary policies to construct the Pareto front in our paper.

The Pareto front of $\mathbb{J}(\mu)$ is denoted as $\mathbb{P}(\mathbb{J}(\mu))$. Lemma 1 shows that the $\mathbb{J}(\mu)$ construct a convex polytope. The boundary and vertices of the convex polytope $\mathbb{J}(\mu)$ are denoted as $\mathbb{B}(\mathbb{J}(\mu))$ and $\mathbb{V}(\mathbb{J}(\mu))$, respectively. Lemma 1 also implies the vertices of $\mathbb{J}(\mu)$ are deterministic policies, that is, $\mathbb{V}(\mathbb{J}(\mu)) \subseteq \Pi_D$.

**Lemma 17.** *(Restatement of Lemma 1) $\mathbb{J}(\mu)$ is a closed convex polytope, and the vertices of $\mathbb{J}(\mu)$ can be achieved by deterministic policies.*

*Proof.* We observe that $\mathbb{J}(\mu)$ can be obtained by applying a linear transformation to $L(\mu)$, that is, $\mathbb{J}(\mu) = \{(\bar{d}_\mu^\pi)^\top r | \bar{d}_\mu^\pi \in L(\mu)\}$.

Then we prove the following lemma, which shows that linear transformations preserve the convex polytope structure and a subset of its vertices.

**Lemma 18.** *Suppose $\mathbb{C}$ is a convex polytope in $n$-dimension space whose vertices set is $\mathbb{V}$, and $\mathbf{A} \in \mathbb{R}^{m \times n}$. Let $\bar{\mathbb{C}} = \{\mathbf{A}x | x \in \mathbb{C}\}$, then $\bar{\mathbb{C}}$ is also a convex polytope with vertices $\bar{\mathbb{V}}$, and $\bar{\mathbb{V}} \subseteq \{\mathbf{A}v | v \in \mathbb{V}\}$.*

*Proof.* Since $\mathbb{C}$ is a convex polytope, it is the convex hull of its finite vertex set $\mathbb{V} = \{v_1, v_2, \ldots, v_k\}$. That is, $\mathbb{C} = \text{Conv}(\mathbb{V}) = \left\{ \sum_{i=1}^{k} \alpha_i v_i \mid \alpha_i \geq 0, \sum_{i=1}^{k} \alpha_i = 1 \right\}$. Applying the linear map $\mathbf{A}$ to $\mathbb{C}$, we get:

$$\bar{\mathbb{C}} = \mathbf{A}\mathbb{C} = \left\{ \mathbf{A}\left(\sum_{i=1}^{k} \alpha_i v_i\right) \mid \alpha_i \geq 0, \sum_{i=1}^{k} \alpha_i = 1 \right\}$$
$$= \left\{ \sum_{i=1}^{k} \alpha_i (\mathbf{A}v_i) \mid \alpha_i \geq 0, \sum_{i=1}^{k} \alpha_i = 1 \right\}.$$

This shows that $\bar{\mathbb{C}}$ is the convex hull of the finite set $\{\mathbf{A}v_1, \mathbf{A}v_2, \ldots, \mathbf{A}v_k\}$, which implies that $\bar{\mathbb{C}}$ is a convex polytope, and the set of vertices $\bar{\mathbb{V}}$ is a subset of $\{\mathbf{A}v | v \in \mathbb{V}\} = \{\mathbf{A}v_1, \mathbf{A}v_2, \ldots, \mathbf{A}v_k\}$. $\square$

The conclusion can be obtained by combining Lemma 11 and Lemma 18. $\square$

By the definition of the Pareto front, the Pareto front is the set of non-dominated policies on $\mathbb{J}(\mu)$. By Lemma 7, the Pareto front is on the boundary of the convex polytope.

We define two policies, $\pi_1$ and $\pi_2$, as *neighboring policies* if an edge on the boundary of the convex polytope $\mathbb{B}(\mathbb{J}(\mu))$ connects the long-term returns of $\pi_1$ and $\pi_2$.

**Lemma 19.** *Under Assumption 1, for any edge on $\mathbb{B}(\mathbb{J}(\mu))$ with endpoints denoted as $J^{\pi_1}(\mu)$ and $J^{\pi_2}(\mu)$, any point on this edge can only be achieved by the long-term return of a convex combination of policies $\pi_1$ and $\pi_2$. Specifically, for any $\alpha \in [0,1]$, there exists a $\beta \in [0,1]$ such that*

$$\alpha J^{\pi_1}(\mu) + (1-\alpha)J^{\pi_1}(\mu) = J^{\beta\pi_1 + (1-\beta)\pi_2}(\mu).$$

*Proof.* Let $\pi_1$ and $\pi_2$ are neighboring points on $\mathbb{B}(\mathbb{J}(\mu))$, and $E(\pi_1, \pi_2)$ is an edge connecting two vertices. Since $E(\pi_1, \pi_2)$ is a face of $\mathbb{J}(\mu)$, by Definition 1, there exists weight vector $w$ and scalar $c_0$ such that $E(\pi_1, \pi_2) = \mathbb{C} \cap \{x \in \mathbb{R}^d : w^\top x = c_0\}$ and $w^\top x \leq c_0$ for any $x \in \mathbb{J}(\mu)$. Therefore, we have

$$\{\pi_1, \pi_2\} = \arg \max_{\pi \in \mathbb{V}(\mathbb{J}(\mu))} w^\top J^\pi(\mu).$$

Moreover, since any points within the convex polytope can be written as the convex combination of extreme points, and the solutions of extreme points to $\arg \max_{\pi \in \Pi_D} w^\top J^\pi(\mu)$ only contains $\pi_1$ and $\pi_2$, therefore

$$\{\alpha J^{\pi_1}(\mu) + (1-\alpha)J^{\pi_1}(\mu) | \alpha \in [0,1]\} = \arg \max_{\pi \in \Pi} w^\top J^\pi(\mu). \tag{3}$$

We construct a single-objective MDP $(\mathcal{S}, \mathcal{A}, P, r^\top w, \gamma)$, where we use the preference vector $w$ to transform $M$-objective reward tensor $r \in \mathbb{R}^{S \times A \times D}$ to single-objective reward $\in \mathbb{R}^{S \times A}$. Under Assumption 1, $\pi_1$ and $\pi_2$ are optimal deterministic policies on $(\mathcal{S}, \mathcal{A}, P, r^\top w, \gamma)$ by Lemma 13. By Lemma 14, the optimal policies on $(\mathcal{S}, \mathcal{A}, P, r^\top w, \gamma)$ lies in the convex hull constructed by $\pi_1$ and $\pi_2$, i.e.,

$$\{J^{\beta\pi_1 + (1-\beta)\pi_2}(\mu) | \beta \in [0,1]\} = \arg \max_{\pi \in \Pi} w^\top J^\pi(\mu). \tag{4}$$

Combining Eq. (3) and Eq. (4), we obtain

$$\{\alpha J^{\pi_1}(\mu) + (1-\alpha)J^{\pi_1}(\mu) \mid \alpha \in [0,1]\} = \{J^{\beta\pi_1 + (1-\beta)\pi_2}(\mu) \mid \beta \in [0,1]\}.$$

This result indicates that, for an initial distribution $\mu$, the edge between neighboring policies in $\mathbb{J}(\mu)$ can only be achieved through the long-term return of the convex combination of these neighboring policies. This concludes our proof. $\square$

**Definition 2.** *We define the distance between two deterministic policies, $\pi_1$ and $\pi_2$, as the number of states where the policies differ. This distance is denoted by $d(\pi_1, \pi_2)$.*

**Lemma 20.** *Given a discounted finite MDP $(\mathcal{S}, \mathcal{A}, \mathbf{P}, \boldsymbol{r}, \gamma)$, where $\mathbf{P}$ is the transition probability kernel and $\boldsymbol{r} \geq 0$ is the reward function. Let $\pi_1$ and $\pi_2$ be two deterministic policies. Under Assumption 1, we have*

1. *$d(\pi_1, \pi_2) = 1$: If $\pi_1$ and $\pi_2$ differ by exactly one state-action pair, $\{J_{\alpha\pi_1 + (1-\alpha)\pi_2}(\mu) | \alpha \in [0,1]\}$ forms a straight line segment between $J^{\pi_1}(\mu)$ and $J^{\pi_1}(\mu)$.*

2. *$d(\pi_1, \pi_2) > 1$: If $\pi_1$ and $\pi_2$ differ by more than one state-action pair, for almost all $\mathbf{P}$ and $\boldsymbol{r}$ in the space of valid transition kernels and rewards (with Lebesgue measure 1), $\{J_{\alpha\pi_1 + (1-\alpha)\pi_2}(\mu) | \alpha \in [0,1]\}$ does not form the straight line segment between $J^{\pi_1}(\mu)$ and $J^{\pi_1}(\mu)$.*

*Proof.* Let $\mathcal{M}$ be the set of states that $\pi_1$ and $\pi_2$ differ on. By definition of distance between two deterministic policies, $M := |\mathcal{M}| = d(\pi_1, \pi_2)$.

Recall that $\Pi_N(\pi, \mathcal{M})$ denotes the set of deterministic policies that differ from $\pi$ only at states within $\mathcal{M}$, while matching $\pi$ at all other states. In other words, all policies in $\Pi_N(\pi, \mathcal{M})$ take the same actions for states outside the set $\mathcal{M}$. For states that cannot reach $\mathcal{M}$ within a finite number of steps, the value function is identical across all such policies. Conversely, for states that can reach $\mathcal{M}$ in finite steps, the value function depends on the policy within $\mathcal{M}$. With some slight abuse of notation, in the remainder of this proof, we focus only on the set of states that can reach $\mathcal{M}$ within a finite number of steps, denoted as $\mathcal{S}$.

Starting from state $s$ under policy $\pi$, we define **hitting time of the set** $\mathcal{M} \subseteq \mathcal{S}$, denoted as $\mathbf{H}^\pi(s, \mathcal{M})$, as the first time step at which the process reaches any state in $\mathcal{M}$. It is defined as:

$$\mathbf{H}^\pi(s, \mathcal{M}) := \inf\{t \geq 0 \mid s_0 = s, s_t \in \mathcal{M}\},$$

where $s_t$ denotes the state of the process at time $t$. For any $\bar{s} \in \mathcal{M}$, we define the random variable $\mathbf{H}^\pi(\bar{s}, \mathcal{M}) = 0$ as it takes 0 steps from $\bar{s}$ to $\mathcal{M}$. Moreover, we define **the total return of $d$-th objective before hitting** $\mathcal{M}$ from state $s$ under policy $\pi$ as the sum of discounted rewards until the process reaches any state in $\mathcal{M}$. It is given by:

$$\tilde{R}_d^\pi(s, \mathcal{M}) := \sum_{t=0}^{\mathbf{H}^\pi(s, \mathcal{M}) - 1} \gamma^t \boldsymbol{r}_d(s_t, a_t),$$

where $r(s_t, a_t)$ is the reward at time step $t$.

Similarly, Starting from state $s$ under policy $\pi$, we denote the **hitting time of a specific state** $\bar{s} \in \mathcal{M}$, **conditioned on avoiding all other states in** $\mathcal{M}$ **before reaching** $\bar{s}$ as $\mathbf{H}^\pi(s, \mathcal{M}, \bar{s})$ with slight abuse of notation. It is defined as:

$$\mathbf{H}^\pi(s, \mathcal{M}, \bar{s}) := \inf\{t \geq 0 \mid s_0 = s, s_t = \bar{s}, s_k \notin \mathcal{M}, \forall k < t\}. \tag{5}$$

For any $\bar{s}', \bar{s} \in \mathcal{M}$ with $\bar{s}' \neq \bar{s}$, the random variable $\mathbf{H}^\pi(\bar{s}', \mathcal{M}, \bar{s})$ are defined as:

- $\mathbf{H}^\pi(\bar{s}, \mathcal{M}, \bar{s}) = 0$ with probability 1, as it takes 0 steps to reach $\bar{s}$ from $\bar{s}$,

- $\mathbf{H}^\pi(\bar{s}', \mathcal{M}, \bar{s}) = \infty$ with probability 1, as it is impossible to reach $\bar{s}$ first from $\bar{s}'$ (since we are already at $\bar{s}' \in \mathcal{M}$).

The total return before hitting $\bar{s} \in \mathcal{M}$, conditioned on avoiding all other states in $\mathcal{M}$ before reaching $\bar{s}$, denoted as $\tilde{R}_d^\pi(s, \mathcal{M}, \bar{s})$:

$$\tilde{R}_d^\pi(s, \mathcal{M}, \bar{s}) := \sum_{t=0}^{\mathbf{H}^\pi(s, \mathcal{M}, \bar{s}) - 1} \gamma^t \boldsymbol{r}_d(s_t, a_t). \tag{6}$$

The **arrival state** is defined as the first state in $\mathcal{M}$ reached by the process, starting from $s$ under policy $\pi$. Denote the arrival state as:

$$S^\pi(s, \mathcal{M}) := s_{\mathbf{H}^\pi(s, \mathcal{M})}, \tag{7}$$

where $s_{\mathbf{H}^\pi(s, \mathcal{M})}$ is the state of the process at the hitting time $\mathbf{H}^\pi(s, \mathcal{M})$. Since $S^\pi(s, \mathcal{M})$ is the first state in $\mathcal{M}$ that is reached, the hitting time of $S^\pi(s, \mathcal{M})$ is indeed the same as the hitting time of $\mathcal{M}$. Similarly, the total return before

hitting $\mathcal{M}$ is the same as the return before hitting $S^\pi(s, \mathcal{M})$ conditioned on avoiding hitting other states in $\mathcal{M}$, that is:

$$
\begin{aligned}
\mathbf{H}^\pi(s, \mathcal{M}) =& \mathbf{H}^\pi(s, \mathcal{M}, S^\pi(s, \mathcal{M})), \\
\tilde{R}_d^\pi(s, \mathcal{M}) =& \sum_{t=0}^{\mathbf{H}^\pi(s,\mathcal{M})-1} \gamma^t \boldsymbol{r}_d(s_t, a_t) = \sum_{t=0}^{\mathbf{H}^\pi(s,\mathcal{M},S^\pi(s,\mathcal{M}))-1} \gamma^t \boldsymbol{r}_d(s_t, a_t) = \tilde{R}_d^\pi(s, \mathcal{M}, S^\pi(s, \mathcal{M})).
\end{aligned}
\tag{8}
$$

Denote the expectation of the total return and discount before hitting $\bar{s}$ conditioned on avoiding all other states in $\mathcal{M}$ before reaching $\bar{s}$ as $\tilde{\mathbf{V}}_d^\pi = [\tilde{V}_d^\pi(s_0, \mathcal{M}), \cdots, \tilde{V}_d^\pi(s_{|\mathcal{S}|}, \mathcal{M})]^T \in \mathbb{R}^{S\times 1}$, $\tilde{\mathbf{V}}^\pi \in \mathbb{R}^{S\times D}$, and $\tilde{\Gamma} \in \mathbb{R}^{S\times M}$, respectively. For $s \in \mathcal{S}$, $s' \in \mathcal{M}$, $d \in \{1, \cdots, D\}$, we define

$$
\begin{aligned}
\tilde{V}_d^\pi(s, \mathcal{M}) :=& \mathbb{E}\left[\tilde{R}_d^\pi(s, \mathcal{M})\right], \\
\tilde{\mathbf{V}}^\pi =& [\tilde{\mathbf{V}}_1^\pi, \ldots, \tilde{\mathbf{V}}_D^\pi], \\
\Gamma^\pi(s, \mathcal{M}, \bar{s}) =& P(S^\pi(s, \mathcal{M}) = \bar{s})\mathbb{E}\left[\gamma^{\mathbf{H}(s,\mathcal{M},\bar{s})}\right].
\end{aligned}
\tag{9}
$$

Recall $\mathbf{V}^\pi \in \mathbb{R}^{S\times D}$ is the value function with initial state distribution $\mu$, where $\mu$ is ignored for the notation simplicity. Moreover, we use $\mathbf{V}_d^\pi(s)$ and $\mathbf{V}_d^\pi(\mathcal{M})$ to denote the $d$-th objective value function at state $s$ and at state set $\mathcal{M}$, respectively. By the definition of value function, $\mathbf{V}_d^\pi(s)$ can be written as

$$
\begin{aligned}
\mathbf{V}_d^\pi(s) =& \mathbb{E}\left[\sum_{t=0}^\infty \gamma^t \boldsymbol{r}_d(s_t, a_t) \Big| s_0 = s\right] \\
=& \mathbb{E}\left[\mathbb{E}\left[\sum_{t=0}^\infty \gamma^t \boldsymbol{r}_{d,t} \Big| S^\pi(s, \mathcal{M})\right] \Big| s_0 = s\right].
\end{aligned}
$$

We next want to calculate the conditional expectation of long-term return given $S^\pi(s, \mathcal{M}) = \bar{s}$.

$$
\begin{aligned}
\mathbb{E}\left[\sum_{t=0}^\infty \gamma^t \boldsymbol{r}_{d,t} \Big| S^\pi(s, \mathcal{M}) = \bar{s}\right] =& \mathbb{E}\left[\sum_{t=0}^{\mathbf{H}(s,\mathcal{M},\bar{s})-1} \gamma^t \boldsymbol{r}_{d,t} + \gamma^{\mathbf{H}(s,\mathcal{M},\bar{s})}\mathbf{V}_d^\pi(\bar{s})\right] \\
=& \mathbb{E}\left[\sum_{t=0}^{\mathbf{H}(s,\mathcal{M},\bar{s})-1} \gamma^t \boldsymbol{r}_{d,t}\right] + \mathbb{E}\left[\gamma^{\mathbf{H}(s,\mathcal{M},\bar{s})}\right]\mathbf{V}_d^\pi(\bar{s}) \\
& \text{(By linearity of expectations and } \mathbf{V}_d^\pi(\bar{s}) \text{ is a constant)} \\
=& \mathbb{E}\left[\tilde{R}_d^\pi(s, \mathcal{M}, \bar{s})\right] + \mathbb{E}\left[\gamma^{\mathbf{H}(s,\mathcal{M},\bar{s})}\right]\mathbf{V}_d^\pi(\bar{s}) \quad \text{(by definition of } \tilde{R}_d^\pi(s, \mathcal{M}, \bar{s}) \text{ in Eq. (6))}
\end{aligned}
$$

Plugging the above equation into $\mathbf{V}_d^\pi(s)$, we have

$$
\begin{aligned}
\mathbf{V}_d^\pi(s) =& \sum_{\bar{s}\in\mathcal{M}} P(S^\pi(s, \mathcal{M}) = \bar{s})\mathbb{E}\left[\sum_{t=0}^\infty \gamma^t \boldsymbol{r}_{d,t} \Big| S^\pi(s, \mathcal{M}) = \bar{s}\right] \\
=& \sum_{\bar{s}\in\mathcal{M}} P(S^\pi(s, \mathcal{M}) = \bar{s})\mathbb{E}\left[\tilde{R}_d^\pi(s, \mathcal{M}, \bar{s})\right] + \sum_{\bar{s}\in\mathcal{M}} P(S^\pi(s, \mathcal{M}) = \bar{s})\mathbb{E}\left[\gamma^{\mathbf{H}(s,\mathcal{M},\bar{s})}\right]\mathbf{V}_d^\pi(\bar{s}) \\
\overset{①}{=}& \mathbb{E}\left[\tilde{R}_d^\pi(s, \mathcal{M})\right] + \sum_{\bar{s}\in\mathcal{M}} P(S^\pi(s, \mathcal{M}) = \bar{s})\mathbb{E}\left[\gamma^{\mathbf{H}(s,\mathcal{M},\bar{s})}\right]\mathbf{V}_d^\pi(\bar{s}) \\
=& \tilde{\mathbf{V}}_d^\pi(s, \mathcal{M}) + \sum_{\bar{s}\in\mathcal{M}} \Gamma^\pi(s, \mathcal{M}, \bar{s})\mathbf{V}_d^\pi(\bar{s}) \quad \text{(by Eq. (9))},
\end{aligned}
$$

where ① is because

$$
\begin{aligned}
\sum_{\bar{s} \in \mathcal{M}} P(S^{\pi}(s, \mathcal{M}) = \bar{s}) \mathbb{E}\left[\tilde{R}_d^{\pi}(s, \mathcal{M}, \bar{s})\right] =& \mathbb{E}\left[\mathbb{E}\left[\sum_{t=0}^{\mathbf{H}(s, \mathcal{M}, S^{\pi}(s, \mathcal{M}))-1} \gamma^t \boldsymbol{r}_d(s_t, a_t)\right] \Bigg| S^{\pi}(s, \mathcal{M}) \Bigg| s_0 = s\right] \\
=& \mathbb{E}\left[\mathbb{E}\left[\sum_{t=0}^{\mathbf{H}(s, \mathcal{M})-1} \gamma^t \boldsymbol{r}_d(s_t, a_t)\right] \Bigg| S^{\pi}(s, \mathcal{M}) \Bigg| s_0 = s\right] \quad \text{(by Eq. (8))} \\
=& \mathbb{E}\left[\sum_{t=0}^{\mathbf{H}(s, \mathcal{M})-1} \gamma^t \boldsymbol{r}_d(s_t, a_t) \Bigg| s_0 = s\right] \\
=& \tilde{R}_d^{\pi}(s, \mathcal{M}).
\end{aligned}
$$

Lemma 15 gives the value of $\Gamma^{\pi}$ and $\tilde{\mathbf{V}}^{\pi}$ as following

$$
\Gamma^{\pi} = \begin{bmatrix} \gamma(\mathbf{I}_{S-M} - \gamma\mathbf{A})^{-1}\mathbf{B} \\ \mathbf{I}_M \end{bmatrix}, \quad \tilde{\mathbf{V}}^{\pi} = \begin{bmatrix} (\mathbf{I}_{S-M} - \gamma\mathbf{A})^{-1}\boldsymbol{X} \\ \mathbf{0}_M \end{bmatrix}.
$$

Note that $\Gamma^{\pi}$ and $\tilde{\mathbf{V}}^{\pi}$ are the same for any policy in $\Pi_N(\pi_1, \mathcal{M})$. Since we only consider policies within $\Pi_N(\pi_1, \mathcal{M})$, for simplicity of notation, we omit the superscripts $\pi$ in $\Gamma^{\pi}$ and $\tilde{\mathbf{V}}^{\pi}$ in the following sections, as they remain constant for any policies within $\Pi_N(\pi_1, \mathcal{M})$.

Rearranging $\mathbf{V}^{\pi}$, we have

$$
\mathbf{V}^{\pi} = \tilde{\mathbf{V}} + \Gamma \mathbf{V}^{\pi}(\mathcal{M}). \tag{10}
$$

By the Bellman equation and Eq. (10), we can rewrite the $\mathbf{V}^{\pi}(\mathcal{M})$ as

$$
\begin{aligned}
\mathbf{V}^{\pi}(\mathcal{M}) =& \boldsymbol{r}^{\pi}(\mathcal{M}) + \gamma\mathbf{P}^{\pi}(\mathcal{M})\mathbf{V}^{\pi} \quad \text{(by Bellman equation)} \\
=& \boldsymbol{r}^{\pi}(\mathcal{M}) + \gamma\mathbf{P}^{\pi}(\mathcal{M})\left(\tilde{\mathbf{V}} + \Gamma\mathbf{V}^{\pi}(\mathcal{M})\right) \quad \text{(by Eq. (10))} \\
=& \left(\boldsymbol{r}^{\pi}(\mathcal{M}) + \gamma\mathbf{P}^{\pi}(\mathcal{M})\tilde{\mathbf{V}}\right) + \gamma\mathbf{P}^{\pi}(\mathcal{M})\Gamma\mathbf{V}^{\pi}(\mathcal{M}).
\end{aligned}
$$

Since $[\mathbf{A}, \mathbf{B}]$ is a stochastic matrix, we have $(\mathbf{I}_{S-M} - \gamma\mathbf{A})^{-1}\mathbf{B}\mathbf{1}_M < (\mathbf{I}_{S-M} - \mathbf{A})^{-1}\mathbf{B}\mathbf{1}_M = \mathbf{1}_{S-M}$, where the inequality means $(\mathbf{I}_{S-M} - \gamma\mathbf{A})^{-1}\mathbf{B}\mathbf{1}_M$ is smaller than $(\mathbf{I}_{S-M} - \mathbf{A})^{-1}\mathbf{B}\mathbf{1}_M$ component-wisely. Moreover, since each element of $\mathbf{A}$ and $\mathbf{B}$ are larger than zero, we also have $(\mathbf{I}_{S-M} - \gamma\mathbf{A})^{-1}\mathbf{B}\mathbf{1}_M > \mathbf{0}_{S-M}$. As each element of $\mathbf{C}^{\pi}$ and $\mathbf{D}^{\pi}$ are not smaller than zero, it follows that $\mathbf{0}_M \le \left(\mathbf{C}^{\pi}(\mathbf{I}_{S-M} - \gamma\mathbf{A})^{-1}\mathbf{B} + \mathbf{D}^{\pi}\right)\mathbf{1}_M < (\mathbf{C}^{\pi} + \mathbf{D}^{\pi})\mathbf{1}_M = \mathbf{1}_M$. Hence, $\|\mathbf{C}^{\pi}(\mathbf{I}_{S-M} - \gamma\mathbf{A})^{-1}\mathbf{B} + \mathbf{D}^{\pi}\|_{\infty} < 1$. By Lemma 6, $\mathbf{P}^{\pi}(\mathcal{M})\Gamma = \gamma\mathbf{C}^{\pi}(\mathbf{I}_{S-M} - \gamma\mathbf{A})^{-1}\mathbf{B} + \mathbf{D}^{\pi}$ has eigenvalues all smaller than 1. Hence the eigenvalue of $(\mathbf{I}_M - \gamma\mathbf{P}^{\pi}(\mathcal{M})\Gamma)$ are larger than zero, implying it is invertible. Rearranging the above equation, we have

$$
\mathbf{V}^{\pi}(\mathcal{M}) = (\mathbf{I}_M - \gamma\mathbf{P}^{\pi}(\mathcal{M})\Gamma)^{-1}\left(\boldsymbol{r}^{\pi}(\mathcal{M}) + \gamma\mathbf{P}^{\pi}(\mathcal{M})\tilde{\mathbf{V}}\right). \tag{11}
$$

Define

$$
\begin{aligned}
\mathbf{F}^{\pi} :=& \gamma\mathbf{P}^{\pi}(\mathcal{M})\Gamma, \\
\mathbf{E}^{\pi} :=& \boldsymbol{r}^{\pi}(\mathcal{M}) + \gamma\mathbf{P}^{\pi}(\mathcal{M})\tilde{\mathbf{V}}.
\end{aligned} \tag{12}
$$

Plugging Eq. (11) into Eq. (10), we obtain

$$
\begin{aligned}
\mathbf{V}^{\pi} =& \tilde{\mathbf{V}} + \Gamma \cdot (\mathbf{I}_M - \gamma\mathbf{P}^{\pi}(\mathcal{M})\Gamma)^{-1}\left(\boldsymbol{r}^{\pi}(\mathcal{M}) + \gamma\mathbf{P}^{\pi}(\mathcal{M})\tilde{\mathbf{V}}\right) \\
=& \tilde{\mathbf{V}} + \Gamma \cdot (\mathbf{I}_M - \mathbf{F}^{\pi})^{-1}\mathbf{E}^{\pi}. \quad \text{(by definition of } \mathbf{F}^{\pi} \text{ and } \mathbf{E}^{\pi})
\end{aligned} \tag{13}
$$

We move to investigate the derivative of $\mathbf{V}^\pi$ with respect to $\pi(\bar{s}, a)$.

$$\frac{\partial \mathbf{V}^\pi}{\partial \pi(\bar{s}, a)} = \Gamma \cdot \frac{\partial \left(\mathbf{I}_M - \mathbf{F}^\pi\right)^{-1}}{\partial \pi(\bar{s}, a)} \mathbf{E}^\pi + \Gamma \cdot \left(\mathbf{I}_M - \mathbf{F}^\pi\right)^{-1} \frac{\partial \mathbf{E}^\pi}{\partial \pi(\bar{s}, a)}$$

$$= \Gamma \cdot \left(\mathbf{I}_M - \mathbf{F}^\pi\right)^{-1} \frac{\partial \mathbf{F}^\pi}{\partial \pi(\bar{s}, a)} \left(\mathbf{I}_M - \mathbf{F}^\pi\right)^{-1} \mathbf{E}^\pi + \Gamma \cdot \left(\mathbf{I}_M - \mathbf{F}^\pi\right)^{-1} \frac{\partial \mathbf{E}^\pi}{\partial \pi(\bar{s}, a)}$$

$$\left(\text{by } \frac{\partial \mathbf{Y}^{-1}}{\partial x} = -\mathbf{Y}^{-1} \frac{\partial \mathbf{Y}}{\partial x} \mathbf{Y}^{-1} \text{ (Petersen et al., 2008)}\right)$$

$$= \Gamma \cdot \left(\mathbf{I}_M - \mathbf{F}^\pi\right)^{-1} \mathbf{P}(\delta(\bar{s}), a, \cdot) \Gamma \cdot \left(\mathbf{I}_M - \mathbf{F}^\pi\right)^{-1} \mathbf{E}^\pi$$

$$+ \Gamma \cdot \left(\mathbf{I}_M - \mathbf{F}^\pi\right)^{-1} \left(\mathbf{r}(\delta(\bar{s}), a, \cdot) + \mathbf{P}(\delta(\bar{s}), a, \cdot)\tilde{\mathbf{V}}\right). \quad \text{(by Lemma 16)}$$

Then we can calculate the directional derivative on the direction of $\pi_2 - \pi_1$, where $\pi_2$ and $\pi_1$ are deterministic policies. Moreover, $\pi_1(\bar{s}) \neq \pi_2(\bar{s})$ for any state $\bar{s}$ belonging to $\mathcal{M}$, and $\pi_1(\bar{s}) = \pi_2(\bar{s})$ for any state other than $\mathcal{M}$. The calculation of the directional derivative, denoted as $\left\langle \frac{\partial \mathbf{V}^\pi}{\partial \pi}, \pi_2 - \pi_1 \right\rangle$, is as follows:

$$\left\langle \frac{\partial \mathbf{V}^\pi}{\partial \pi}, \pi_2 - \pi_1 \right\rangle = \sum_{\bar{s} \in \mathcal{M}, a \in \mathcal{A}} \frac{\partial \mathbf{V}^\pi}{\partial \pi(\bar{s}, a)} \left(\pi_2(\bar{s}, a) - \pi_1(\bar{s}, a)\right)$$

$$= \Gamma \cdot \left(\mathbf{I}_M - \mathbf{F}^\pi\right)^{-1} \sum_{\bar{s} \in \mathcal{M}} \left(\mathbf{P}(\delta(\bar{s}), \pi_2(\bar{s}), \cdot) - \mathbf{P}(\delta(\bar{s}), \pi_1(\bar{s}), \cdot)\right) \Gamma \cdot \left(\mathbf{I}_M - \mathbf{F}^\pi\right)^{-1} \mathbf{E}^\pi$$

$$+ \Gamma \cdot \left(\mathbf{I}_M - \mathbf{F}^\pi\right)^{-1} \sum_{\bar{s} \in \mathcal{M}} \left(\left(\mathbf{r}(\delta(\bar{s}), \pi_2(\bar{s}), \cdot) + \mathbf{P}(\delta(\bar{s}), \pi_2(\bar{s}), \cdot)\tilde{\mathbf{V}}\right)\right.$$

$$\left. - \left(\mathbf{r}(\delta(\bar{s}), \pi_1(\bar{s}), \cdot) + \mathbf{P}(\delta(\bar{s}), \pi_1\bar{s}), \cdot)\tilde{\mathbf{V}}\right)\right)$$

$$\text{(by (1) of Lemma 16)}$$

$$= \Gamma \cdot \left(\mathbf{I}_M - \mathbf{F}^\pi\right)^{-1} \left(\mathbf{F}^{\pi_2} - \mathbf{F}^{\pi_1}\right) \left(\mathbf{I}_M - \mathbf{F}^\pi\right)^{-1} \mathbf{E}^\pi + \Gamma \cdot \left(\mathbf{I}_M - \mathbf{F}^\pi\right)^{-1} \left(\mathbf{E}^{\pi_2} - \mathbf{E}^{\pi_1}\right)$$

$$\text{(by (2) of Lemma 16)}.$$

Consequently, the directional derivative along the direction of $\pi_2 - \pi_1$ at a convex combination of $\pi_1$ and $\pi_2$, namely $\alpha\pi_1 + (1-\alpha)\pi_2$ with $0 \leq \alpha \leq 1$, can be expressed as follows:

$$\left\langle \frac{\partial \mathbf{V}^\pi}{\partial \pi} | \pi = \alpha\pi_1 + (1-\alpha)\pi_2, \pi_2 - \pi_1 \right\rangle$$

$$= \Gamma \cdot \left(\mathbf{I}_M - \mathbf{F}^\pi\right)^{-1} \left(\mathbf{F}^{\pi_2} - \mathbf{F}^{\pi_1}\right) \left(\mathbf{I}_M - \mathbf{F}^\pi\right)^{-1} \mathbf{E}^\pi + \Gamma \cdot \left(\mathbf{I}_M - \mathbf{F}^\pi\right)^{-1} \left(\mathbf{E}^{\pi_2} - \mathbf{E}^{\pi_1}\right)$$

$$= \Gamma \cdot \left(\mathbf{I}_M - \mathbf{F}^\pi\right)^{-1} \left(\mathbf{F}^{\pi_2} - \mathbf{F}^{\pi_1}\right) \left(\mathbf{I}_M - \mathbf{F}^\pi\right)^{-1} \left(\alpha\mathbf{E}^{\pi_1} + (1-\alpha)\mathbf{E}^{\pi_2}\right)$$

$$+ \Gamma \cdot \left(\mathbf{I}_M - \mathbf{F}^\pi\right)^{-1} \left(\mathbf{E}^{\pi_2} - \mathbf{E}^{\pi_1}\right)$$

$$\text{(by } \mathbf{E}^\pi = \alpha\mathbf{E}^{\pi_1} + (1-\alpha)\mathbf{E}^{\pi_2} \text{ from (3) of Lemma 16)} \quad (14)$$

$$= \Gamma \cdot \left(\mathbf{I}_M - \mathbf{F}^\pi\right)^{-1} \left((1-\alpha)\left(\mathbf{F}^{\pi_2} - \mathbf{F}^{\pi_1}\right) + \left(\mathbf{I}_M - \mathbf{F}^\pi\right)\right) \left(\mathbf{I}_M - \mathbf{F}^\pi\right)^{-1} \mathbf{E}^{\pi_2}$$

$$+ \Gamma \cdot \left(\mathbf{I}_M - \mathbf{F}^\pi\right)^{-1} \left(\alpha\left(\mathbf{F}^{\pi_2} - \mathbf{F}^{\pi_1}\right) - \left(\mathbf{I}_M - \mathbf{F}^\pi\right)\right) \left(\mathbf{I}_M - \mathbf{F}^\pi\right)^{-1} \mathbf{E}^{\pi_1}$$

$$= \Gamma \cdot \left(\mathbf{I}_M - \mathbf{F}^\pi\right)^{-1} \left(\mathbf{I}_M - \mathbf{F}^{\pi_1}\right) \left(\mathbf{I}_M - \mathbf{F}^\pi\right)^{-1} \mathbf{E}^{\pi_2}$$

$$- \Gamma \cdot \left(\mathbf{I}_M - \mathbf{F}^\pi\right)^{-1} \left(\mathbf{I}_M - \mathbf{F}^{\pi_2}\right) \left(\mathbf{I}_M - \mathbf{F}^\pi\right)^{-1} \mathbf{E}^{\pi_1}$$

$$\text{(by } \mathbf{F}^\pi = \alpha\mathbf{F}^{\pi_1} + (1-\alpha)\mathbf{F}^{\pi_2} \text{ from (3) of Lemma 16)}.$$

Furthermore, the directional derivative along the direction $\pi_2 - \pi_1$ at the point $\pi_1$ can be written as:

$$\left\langle \frac{\partial \mathbf{V}^\pi}{\partial \pi} | \pi = \pi_1, \pi_2 - \pi_1 \right\rangle = \Gamma \cdot \left(\mathbf{I}_M - \mathbf{F}^{\pi_1}\right)^{-1} \left(\mathbf{I}_M - \mathbf{F}^{\pi_1}\right) \left(\mathbf{I}_M - \mathbf{F}^{\pi_1}\right)^{-1} \mathbf{E}^{\pi_2}$$

$$- \Gamma \cdot \left(\mathbf{I}_M - \mathbf{F}^{\pi_1}\right)^{-1} \left(\mathbf{I}_M - \mathbf{F}^{\pi_2}\right) \left(\mathbf{I}_M - \mathbf{F}^{\pi_1}\right)^{-1} \mathbf{E}^{\pi_1}$$

$$= \Gamma \cdot \left(\mathbf{I}_M - \mathbf{F}^{\pi_1}\right)^{-1} \mathbf{E}^{\pi_2} - \Gamma \cdot \left(\mathbf{I}_M - \mathbf{F}^{\pi_1}\right)^{-1} \left(\mathbf{I}_M - \mathbf{F}^{\pi_2}\right) \left(\mathbf{I}_M - \mathbf{F}^{\pi_1}\right)^{-1} \mathbf{E}^{\pi_1}$$

$$= \Gamma \cdot \left(\mathbf{I}_M - \mathbf{F}^{\pi_1}\right)^{-1} \left(\mathbf{I}_M - \mathbf{F}^{\pi_2}\right) \left(\left(\mathbf{I}_M - \mathbf{F}^{\pi_2}\right)^{-1} \mathbf{E}^{\pi_2} - \left(\mathbf{I}_M - \mathbf{F}^{\pi_1}\right)^{-1} \mathbf{E}^{\pi_1}\right).$$

$$(15)$$

The statement that $\mathbf{V}^{\alpha\pi_1+(1-\alpha)\pi_2}$ lies on the straight line between $\mathbf{V}^{\pi_1}$ and $\mathbf{V}^{\pi_2}$ is equivalent to that the directional derivative of $\mathbf{V}^\pi$ along the direction $\pi_2 - \pi_1$ at the point $\alpha\pi_1 + (1-\alpha)\pi_2$ aligns with the direction of $\mathbf{V}^{\pi_2} - \mathbf{V}^{\pi_1}$. That is, for any $0 \le \alpha \le 1$, there exists $k \in \mathbb{R}$ which is related to $\alpha$ such that the following equation holds

$$\left\langle \frac{\partial \mathbf{V}^\pi}{\partial \pi}\Big|_{\pi=\alpha\pi_1+(1-\alpha)\pi_2}, \pi_2 - \pi_1 \right\rangle = k\left(\mathbf{V}^{\pi_2} - \mathbf{V}^{\pi_1}\right). \tag{16}$$

Plugging Eq. (13) and Eq. (14) into Eq. (16), LHS and RHS can be written as

$$\begin{aligned}
\text{LHS} =& \Gamma \cdot (\mathbf{I}_M - \mathbf{F}^\pi)^{-1}(\mathbf{I}_M - \mathbf{F}^{\pi_1})(\mathbf{I}_M - \mathbf{F}^\pi)^{-1}\mathbf{E}^{\pi_2} \\
& - \Gamma \cdot (\mathbf{I}_M - \mathbf{F}^\pi)^{-1}(\mathbf{I}_M - \mathbf{F}^{\pi_2})(\mathbf{I}_M - \mathbf{F}^\pi)^{-1}\mathbf{E}^{\pi_1}, \\
\text{RHS} =& k\left(\mathbf{V}^{\pi_2} - \mathbf{V}^{\pi_1}\right) \\
=& k\left(\left(\tilde{\mathbf{V}} + \Gamma \cdot (\mathbf{I}_M - \mathbf{F}^{\pi_2})^{-1}\mathbf{E}^{\pi_2}\right) - \left(\tilde{\mathbf{V}} + \Gamma \cdot (\mathbf{I}_M - \mathbf{F}^{\pi_1})^{-1}\mathbf{E}^{\pi_1}\right)\right) \\
=& k\Gamma \cdot \left((\mathbf{I}_M - \mathbf{F}^{\pi_2})^{-1}\mathbf{E}^{\pi_2} - (\mathbf{I}_M - \mathbf{F}^{\pi_1})^{-1}\mathbf{E}^{\pi_1}\right).
\end{aligned}$$

Note that when $M = 1$, $\mathbf{I}_M - \mathbf{F}^\pi$ is a scalar and $\mathbf{E}^\pi \in \mathbb{R}^d$. In this case, for any $0 \le \alpha \le 1$, there exists $k$ such that LHS = RHS. To show this, define scalars $b \coloneqq (\mathbf{I}_M - \mathbf{F}^\pi)$, $b_1 \coloneqq (\mathbf{I}_M - \mathbf{F}^{\pi_1})$, $b_2 \coloneqq (\mathbf{I}_M - \mathbf{F}^{\pi_2})$. Then we can express LHS $= b_1 b_2 / b^2 \Gamma\left(\mathbf{E}^{\pi_2}/b_2 - \mathbf{E}^{\pi_1}/b_1\right)$ and RHS $= \Gamma\left(\mathbf{E}^{\pi_2}/b_2 - \mathbf{E}^{\pi_1}/b_1\right)$. Clearly letting $k = b_1 b_2 / b^2$ makes Eq. (16) hold. As a result, $\mathbf{V}^{\alpha\pi_1+(1-\alpha)\pi_2}$ *lies on the straight line between* $\mathbf{V}^{\pi_1}$ *and* $\mathbf{V}^{\pi_2}$ *when* $d(\pi_1, \pi_2) = 1$.

A necessary condition for the statement $\mathbf{V}^{\alpha\pi_1+(1-\alpha)\pi_2}$ lies on the straight line between $\mathbf{V}^{\pi_1}$ and $\mathbf{V}^{\pi_2}$ is that the directional derivative of $\mathbf{V}^\pi$ along the direction $\pi_2 - \pi_1$ at the point $\pi_1$ aligns with the direction of $\mathbf{V}^{\pi_2} - \mathbf{V}^{\pi_1}$, i.e., Eq. (16) when $\pi = \pi_1$. Formally, there exists a scalar $k$ such that

$$\left\langle \frac{\partial \mathbf{V}^\pi}{\partial \pi}\Big|_{\pi=\pi_1}, \pi_2 - \pi_1 \right\rangle = k\left(\mathbf{V}^{\pi_2} - \mathbf{V}^{\pi_1}\right). \tag{17}$$

Define $\Lambda \coloneqq (\mathbf{I}_M - \mathbf{F}^{\pi_2})^{-1}\mathbf{E}^{\pi_2} - (\mathbf{I}_M - \mathbf{F}^{\pi_1})^{-1}\mathbf{E}^{\pi_1}$. Plugging Eq. (15) into LHS, we have

$$\begin{aligned}
\text{LHS} =& \Gamma \cdot (\mathbf{I}_M - \mathbf{F}^{\pi_1})^{-1}(\mathbf{I}_M - \mathbf{F}^{\pi_2})\left((\mathbf{I}_M - \mathbf{F}^{\pi_2})^{-1}\mathbf{E}^{\pi_2} - (\mathbf{I}_M - \mathbf{F}^{\pi_1})^{-1}\mathbf{E}^{\pi_1}\right) \\
=& \Gamma \cdot (\mathbf{I}_M - \mathbf{F}^{\pi_1})^{-1}(\mathbf{I}_M - \mathbf{F}^{\pi_2})\Lambda,
\end{aligned}$$

RHS can be rewritten as

$$\text{RHS} = k\Gamma \cdot \Lambda.$$

By Lemma 15, $\Gamma = \left[\left(\gamma(\mathbf{I}_{S-M} - \gamma\mathbf{A})^{-1}\mathbf{B}\right)^\top, \mathbf{I}_M\right]^\top \in \mathbb{R}^{S\times M}$ and $\Gamma$ has rank $M \le S$. Therefore, $\Gamma$ has full column rank. By Lemma 5, Eq. (17) is equivalent to the following equation:

$$\left((\mathbf{I}_M - \mathbf{F}^{\pi_1})^{-1}(\mathbf{I}_M - \mathbf{F}^{\pi_2}) - k\mathbf{I}_M\right)\Lambda = \mathbf{0}. \tag{18}$$

Note that $\Lambda \in \mathbb{R}^{M\times D}$, and $\mathrm{rank}(\Lambda) \le \min(M, D)$.

**(1) When** $\mathrm{rank}(\Lambda) = M$, $\Lambda$ has full row rank. This case also implies $M \le D$. By Theorem 3, the only solution to $\mathbf{X}\Lambda = \mathbf{0}$ is $\mathbf{X} = \mathbf{0}$. Therefore, Eq. (18) is equal to

$$(\mathbf{I}_M - \mathbf{F}^{\pi_1})^{-1}(\mathbf{I}_M - \mathbf{F}^{\pi_2}) - k\mathbf{I}_M = \mathbf{0}.$$

Substituting the definition of $\mathbf{F}^\pi$ as $\mathbf{F}^\pi = \gamma\mathbf{P}^\pi(\mathcal{M})\Gamma$ in the above equation and rearrange the equation, we have

$$\mathbf{P}^{\pi_2}(\mathcal{M})\Gamma = \frac{1}{\gamma}\left(\mathbf{I}_M - k\left(\mathbf{I}_M - \gamma\mathbf{P}^{\pi_1}(\mathcal{M})\Gamma\right)\right).$$

Moreover, to make $\mathbf{P}^{\pi_2}$ a valid probability distribution vector, $\mathbf{P}^{\pi_2}$ should satisfy $\mathbf{P}^{\pi_2}\mathbf{1}_S = \mathbf{1}_M$. Therefore, $\mathbf{P}^{\pi_2}(\mathcal{M})$ should satisfy $\mathbf{P}^{\pi_2}(\mathcal{M})[\Gamma, \mathbf{1}_S] = [\mathbf{I}_M - k(\mathbf{I}_M - \gamma\mathbf{P}^{\pi_1}(\mathcal{M})\Gamma)/\gamma, \mathbf{1}_M]$.

When $M = S$, $[\Gamma, \mathbf{1}_S] \in \mathbb{R}^{S\times(S+1)}$ has rank $S$. When $M < S$, Since $[\Gamma, \mathbf{1}_S]$ has $S$ rows and rank at least $M$. Hence, $[\Gamma, \mathbf{1}_S]$ has $S$ rows and rank at least $M$ in both cases. Fix $k$. By the rank-nullity theorem Theorem 3 and Lemma 4, for $\bar{s} \in \mathcal{M}$, if the solution of $\mathbf{P}^{\pi_2}(\bar{s})$ exists, the dimension of the solution $\mathbf{P}^{\pi_2}(\bar{s})$ is at most $S - M$. On the other hand, if no solution to $\mathbf{P}^{\pi_2}(\bar{s})$ exists, the dimension of the solution space is 0. Therefore, the dimension of solution space of $\mathbf{P}^{\pi_2}(\mathcal{M})$ is at most $M \times (S - M)$.

By varying $k \in \mathbb{R}$, the dimension of solution space of $\mathbf{P}^{\pi_2}(\mathcal{M})$ is $M \times (S - M) + 1$.

Since valid probability transition matrix $\mathbf{P}^{\pi_2}(\mathcal{M})$ requires that the row sums equal one, the ambient space $\mathbf{P}^{\pi_2}(\mathcal{M})$ lies in has dimension $M \times (S - 1)$. Since the solution space of $\mathbf{P}^{\pi_2}(\mathcal{M})$ that satisfies Eq. (18) has dimension at most $M \times (S - M) + 1$ with $\mathrm{rank}(\Lambda) > 0$, while the ambient space where $\mathbf{P}^{\pi_2}(\mathcal{M})$ resides has dimension $M \times (S - 1)$. When $M > 1$, the dimension of the solution space is significantly lower than that of the full space. Therefore, the set of points where $\mathbf{P}^{\pi_2}(\mathcal{M})$ satisfies Eq. (18) forms a subset of Lebesgue measure 0 in the $M \times (S - 1)$-dimensional space.

**(2) When** $0 < \mathrm{rank}(\Lambda) < M$, the dimension of the solution set to $\mathbf{X}\Lambda = \mathbf{0}$ is $M \times (M - \mathrm{rank}(\Lambda))$ by Theorem 3. Let $\mathbf{H}$ be a possible solution to $\mathbf{X}\Lambda = \mathbf{0}$. Then we have

$$\left(\mathbf{I}_M - \mathbf{F}^{\pi_1}\right)^{-1} \left(\mathbf{I}_M - \mathbf{F}^{\pi_2}\right) - k\mathbf{I}_M = \mathbf{H}.$$

Rewrite the above equation, we have

$$\mathbf{P}^{\pi_2}(\mathcal{M})\Gamma = \frac{1}{\gamma}\left(\mathbf{I}_M - k\left(\mathbf{I}_M - \gamma\mathbf{P}^{\pi_1}(\mathcal{M})\Gamma \cdot (\mathbf{H} + k\mathbf{I}_M)\right)\right).$$

Define $\mathbb{G} = \{\mathbf{G} | \mathbf{G} = \frac{1}{\gamma}\left(\mathbf{I}_M - k\left(\mathbf{I}_M - \gamma\mathbf{P}^{\pi_1}(\mathcal{M})\Gamma\left(\mathbf{H} + k\mathbf{I}_M\right)\right)\right), \mathbf{H}\Lambda = \mathbf{0}, k \in \mathbb{R}\}$. Since the dimension of the solution set to $\mathbf{X}\Lambda = \mathbf{0}$ is $M \times (M - \mathrm{rank}(\Lambda))$ and $\mathbf{I}_M$ is not a solution to $\mathbf{X}\Lambda = \mathbf{0}$, we have $\dim(\mathbb{G}) = M \times (M - \mathrm{rank}(\Lambda)) + 1$.

Fix $G$ in $\mathbb{G}$. Similar to the analysis of (1), the dimension of solution space of $\mathbf{P}^{\pi_2}(\mathcal{M})$ is at most $M \times (S - M)$.

Combining with the dimension of $\mathbb{G}$, the dimension of solution space of $\mathbf{P}^{\pi_2}(\mathcal{M})$ that satisfies Eq. (18) is at most $M \times (S - M) + M \times (M - \mathrm{rank}(\Lambda)) + 1 = M \times (S - \mathrm{rank}(\Lambda)) + 1$. Since the solution space of $\mathbf{P}^{\pi_2}(\mathcal{M})$ that satisfies Eq. (18) has dimension at most $M \times (S - \mathrm{rank}(\Lambda)) + 1$ with $\mathrm{rank}(\Lambda) > 0$, while the ambient space where $\mathbf{P}^{\pi_2}(\mathcal{M})$ resides has dimension $M \times (S - 1)$. When $M > 1$, the dimension of the solution space is significantly lower than that of the full space. Therefore, according to results from measure theory, the set of points where $\mathbf{P}^{\pi_2}(\mathcal{M})$ satisfies Eq. (18) forms a subset of Lebesgue measure 0 in the $M \times (S - 1)$-dimensional space.

**(3) When** $\mathrm{rank}(\Lambda) = 0$, we have

$$\left(\mathbf{I}_M - \mathbf{F}^{\pi_2}\right)^{-1} \mathbf{E}^{\pi_2} - \left(\mathbf{I}_M - \mathbf{F}^{\pi_1}\right)^{-1} \mathbf{E}^{\pi_1} = \mathbf{0}.$$

Plugging $\mathbf{E}^{\pi} = \mathbf{r}^{\pi}(\mathcal{M}) + \gamma\mathbf{P}^{\pi}(\mathcal{M})\tilde{\mathbf{V}}$ into the above equation and rearrange it, we have

$$\mathbf{r}^{\pi_2}(\mathcal{M}) = \left(\mathbf{I}_M - \mathbf{F}^{\pi_2}\right)\left(\mathbf{I}_M - \mathbf{F}^{\pi_1}\right)^{-1} \left(\mathbf{r}^{\pi_1}(\mathcal{M}) + \gamma\mathbf{P}^{\pi_1}(\mathcal{M})\tilde{\mathbf{V}}\right) - \gamma\mathbf{P}^{\pi_2}(\mathcal{M})\tilde{\mathbf{V}}.$$

When $\mathbf{P}$ and $\mathbf{r}^{\pi_1}(\mathcal{M})$ are given, $\mathbf{r}^{\pi_2}(\mathcal{M})$ has only one solution. Hence, the solution space of $\mathbf{r}^{\pi_2}(\mathcal{M})$ that satisfies Eq. (18) has dimension 0, while the ambient space where $\mathbf{r}^{\pi_2}(\mathcal{M})$ resides has dimension $M \times D$. The dimension of the solution space is significantly lower than that of the full space. Therefore, according to results from measure theory, the set of points where $\mathbf{r}^{\pi_2}(\mathcal{M})$ satisfies Eq. (18) forms a subset of Lebesgue measure 0 in the $M \times D$-dimensional space.

$\square$

**Theorem 4.** *Given a finite MDP $(\mathcal{S}, \mathcal{A}, \mathbf{P}, \mathbf{r}, \gamma)$, where $\mathbf{P}$ is the transition probability kernel and $\mathbf{r} \geq 0$ is the reward function. The endpoints for any edges on $\mathbb{B}(\mathbb{J}(\mu))$ are deterministic policies and only differ in one state-action pair almost surely.*

*Proof.* Suppose $\pi_1$ and $\pi_2$ are two neighbouring vertices in $\mathbb{J}(\mu)$. By Lemma 1, $\pi_1$ and $\pi_2$ that are vertices of $\mathbb{J}(\mu)$ are deterministic policies.

Lemma 19 shows that the edge connecting $\pi_1$ and $\pi_2$ can only be formed by the long-term return of policies that are convex combination of $\pi_1$ and $\pi_2$, that is,

$$\{\alpha J^{\pi_1}(\mu) + (1 - \alpha)J^{\pi_1}(\mu) \mid \alpha \in [0, 1]\} = \{J^{\beta\pi_1 + (1-\beta)\pi_2}(\mu) \mid \beta \in [0, 1]\}.$$

However, if $d(\pi_1, \pi_2) > 1$, Lemma 20 shows that it is almost surely that

$$\{\alpha J^{\pi_1}(\mu) + (1 - \alpha)J^{\pi_1}(\mu) \mid \alpha \in [0, 1]\} \neq \{J^{\beta\pi_1 + (1-\beta)\pi_2}(\mu) \mid \beta \in [0, 1]\}.$$

This leads to a contradiction. Therefore, we must have $d(\pi_1, \pi_2) = 1$. This concludes the proof. $\square$

**Theorem 5.** *Let $\mathbb{C}$ be a convex polytope, and let $\mathbb{V}$ be the set of vertices of $\mathbb{C}$. Let $\mathbb{P}$ denote the Pareto front of $\mathbb{C}$, with $\mathbb{V}(\mathbb{P})$ and $\mathbb{F}(\mathbb{P})$ representing the set of vertices and faces on the Pareto front, respectively. Let $\mathbb{V}_P \subseteq \mathbb{V}(\mathbb{P})$ and $\mathbb{F}_P \subseteq \mathbb{F}(\mathbb{P})$ be subsets of the vertices and faces of the Pareto front, such that every face in $\mathbb{F}_P$ is composed of vertices in $\mathbb{V}_P$. Given a set of vertices $\mathbb{T} \subseteq \mathbb{V}$ such that $\mathbb{V}_P \subseteq \mathbb{T}$, then $\mathbb{F}_P \subseteq \mathbb{F}(\mathrm{Conv}(\mathbb{T}))$.*

*Proof.* Pick a face $F$ from $\mathbb{F}_P$. By Lemma 7, $F$ is also on the boundary of $\mathbb{C}$. Since the vertices of $F$ belong to $\mathbb{V}_P$, we can express $F$ as:

$$F = \mathbb{P} \cap \{\boldsymbol{x} \in \mathbb{R}^d : \boldsymbol{w}^\top \boldsymbol{x} = c\},$$

where and $\boldsymbol{w}^\top \boldsymbol{x} \leq c$ for all $\boldsymbol{x} \in \mathbb{C}$. Moreover, since $F$ is on the Pareto front, we have $\boldsymbol{w} > 0$.

Given that $\mathbb{V}_P \subseteq \mathbb{T}$, $F$ is a slice of the convex polytope $\mathrm{Conv}(\mathbb{T})$. $F$ is a valid face of $\mathrm{Conv}(\mathbb{T})$ only if $\boldsymbol{w}^\top \boldsymbol{x} \leq c$ for all $\boldsymbol{x} \in \mathrm{Conv}(\mathbb{T})$.

We proceed by contradiction. Suppose there exists $\boldsymbol{x} \in \mathrm{Conv}(\mathbb{T})$ such that $\boldsymbol{w}^\top \boldsymbol{x} > c$. Since $\mathbb{T} \subseteq \mathbb{V}$ and $\mathbb{C} = \mathrm{Conv}(\mathbb{V})$, it follows that $\mathrm{Conv}(\mathbb{T}) \subseteq \mathbb{C}$. This would imply that $\boldsymbol{w}^\top (\boldsymbol{x} - \boldsymbol{y}) > 0$ for some $\boldsymbol{y} \in F$ and $\boldsymbol{x} \in \mathbb{C}$. As $\boldsymbol{w} > 0$, by Lemma 8, this would imply that $\boldsymbol{x}$ dominates $\boldsymbol{y}$, which contradicts the assumption that $\boldsymbol{y}$ is a Pareto optimal point of $\mathbb{C}$. Therefore, no such $\boldsymbol{x}$ can exist, and $F$ is a valid face of $\mathrm{Conv}(\mathbb{T})$.

Thus, we conclude that $\mathbb{F}_P \subseteq \mathbb{F}(\mathrm{Conv}(\mathbb{T}))$. $\qquad\square$

Consider a special case of Theorem 5 when $\mathbb{V}_P = \mathbb{V}(\mathbb{P})$ and $\mathbb{F}_P = \mathbb{F}(\mathbb{P})$, we have the following proposition.

**Proposition 2.** *Let $\mathbb{C}$ be a convex polytope, and let $\mathbb{V}$ be the set of vertices of $\mathbb{C}$. Let $\mathbb{P}$ denote the Pareto front of $\mathbb{C}$, with $\mathbb{V}(\mathbb{P})$ and $\mathbb{F}(\mathbb{P})$ representing the set of vertices and faces on the Pareto front, respectively. Given a set of vertices $\mathbb{T} \subseteq \mathbb{V}$ such that $\mathbb{V}(\mathbb{P}) \subseteq \mathbb{T}$, then $\mathbb{F}(\mathbb{P}) \subseteq \mathbb{F}(\mathrm{Conv}(\mathbb{T}))$.*

# E    SUFFICIENCY OF TRAVERSING OVER EDGES

**Lemma 21.** *(Existence of Neighborhood Edges on Pareto front of $\mathbb{J}(\mu)$) Suppose $\mathbb{J}(\mu)$ contains multiple Pareto optimal vertices. Let $\pi$ be a Pareto optimal policy, and let $\mathrm{N}(J^\pi(\mu), \mathbb{J}(\mu))$ represent all neighboring points of $J^\pi(\mu)$ on the boundary of $\mathbb{J}(\mu)$. Then, there exists $J \in \mathrm{N}(J^\pi(\mu), \mathbb{J}(\mu))$ such that the edge connecting $J$ and $J^\pi(\mu)$ lies on the Pareto front.*

*Proof.* By Lemma 1, $\mathbb{J}(\mu)$ is a convex polytope. Given $J \in \mathrm{N}(J^\pi(\mu), \mathbb{J}(\mu))$, $J$ and $J^{\pi_1}(\mu)$ are neighboring vertices on $\mathbb{J}(\mu)$, and the edge between $J$ and $J^{\pi_1}(\mu)$ is a 1-dimensional face of $\mathbb{J}(\mu)$. By the definition of Definition 1, there exists a vector $\boldsymbol{w}_J$ such that $\boldsymbol{w}_J^\top J^{\pi_1}(\mu) = \boldsymbol{w}_J^\top J$, and for any $\boldsymbol{x} \in \mathbb{J}(\mu)$ that not on the edge between $J$ and $J^{\pi_1}(\mu)$, we have $\boldsymbol{w}_J^\top J > \boldsymbol{w}_J^\top \boldsymbol{x}$.

**We first prove that there exists $J \in \mathrm{N}(J^\pi(\mu), \mathbb{J}(\mu))$ such that $\boldsymbol{w}_J > 0$.**  We prove this by contradiction. Assume that for every $J \in \mathrm{N}(J^\pi(\mu), \mathbb{J}(\mu))$, there does not exist $\boldsymbol{w}_J > 0$.

By the definition of the Pareto front, if $J^\pi(\mu)$ is not strictly dominated by $J$, meaning that $J^\pi(\mu)$ is greater than or equal to $J$ in some dimensions and strictly smaller in others, then there must exist a positive vector $\boldsymbol{w}_J > 0$. Hence, if for every $J \in \mathrm{N}(J^\pi(\mu), \mathbb{J}(\mu))$, $\boldsymbol{w}_J > 0$ does not exist, it implies that either $J^\pi(\mu)$ is strictly dominated by $J$ or $J$ is strictly dominated by $J^\pi(\mu)$. Since $J^\pi(\mu)$ is on the Pareto front, no point in $\mathbb{J}(\mu)$ can strictly dominate $J^\pi(\mu)$. Therefore, it must be the case that $J$ is strictly dominated by $J^\pi(\mu)$ for all $J \in \mathrm{N}(J^\pi(\mu), \mathbb{J}(\mu))$. This implies that for all $J \in \mathrm{N}(J^\pi(\mu), \mathbb{J}(\mu))$ and any $\boldsymbol{w} > 0$, we have $\boldsymbol{w}^\top(J - J^{\pi_1}(\mu)) < 0$.

We then show that $J^\pi(\mu)$ dominates every point within $\mathbb{J}(\mu)$. For any point $\boldsymbol{x} \in \mathbb{J}(\mu)$, we can express $\boldsymbol{x}$ as a linear combination of $J^\pi(\mu)$ and the neighboring points in $\mathrm{N}(J^\pi(\mu), \mathbb{J}(\mu))$ with non-negative weight vector $\boldsymbol{\alpha}$:

$$\boldsymbol{x} = J^\pi(\mu) + \sum_{J_i \in \mathrm{N}(J^\pi(\mu), \mathbb{J}(\mu))} \boldsymbol{\alpha}_i(J_i - J^\pi(\mu)).$$

Consider any such point $\boldsymbol{x}$ and any vector $\boldsymbol{w} > 0$. We have:

$$
\begin{aligned}
\boldsymbol{w}^\top(\boldsymbol{x} - J^\pi(\mu)) &= \boldsymbol{w}^\top \left( J^\pi(\mu) + \sum_{J_i \in \mathrm{N}(J^\pi(\mu), \mathbb{J}(\mu))} \boldsymbol{\alpha}_i(J_i - J^\pi(\mu)) - J^\pi(\mu) \right) \\
&= \boldsymbol{w}^\top \left( \sum_{J_i \in \mathrm{N}(J^\pi(\mu), \mathbb{J}(\mu))} \boldsymbol{\alpha}_i(J_i - J^{\pi_1}(\mu)) \right) \\
&= \sum_{J_i \in \mathrm{N}(J^\pi(\mu), \mathbb{J}(\mu))} \boldsymbol{\alpha}_i \boldsymbol{w}^\top(J_i - J^{\pi_1}(\mu)) \\
&< 0.
\end{aligned}
$$

where the last inequality is because $\boldsymbol{w}^\top (J_i - J^{\pi_1}(\mu)) < 0$ for any $J_i \in \mathrm{N}(J^\pi(\mu), \mathbb{J}(\mu))$ from previous analysis and $\boldsymbol{\alpha} > \mathbf{0}$ component-wisely.

This implies that $J^\pi(\mu)$ dominates every point within $\mathbb{J}(\mu)$. However, this contradicts the condition that the multi-objective MDP has more than one deterministic Pareto optimal policy. Hence, our assumption is wrong and there exists $J \in \mathrm{N}(J^\pi(\mu), \mathbb{J}(\mu))$ such that $\boldsymbol{w}_J > \mathbf{0}$.

**We next show that $J$, as well as the edge between $J$ and $J^{\pi_1}(\mu)$, is on the Pareto front.** Since there exists a $\boldsymbol{w}_J > \mathbf{0}$ such that $\boldsymbol{w}_J^\top (J - \boldsymbol{x}) \geq \mathbf{0}$ for any $\boldsymbol{x} \in \mathbb{J}(\mu)$, it follows Lemma 8 that $J$ is also Pareto optimal. Moreover, the convex combination of $J$ and $J^{\pi_1}(\mu)$, written as $\alpha J + (1 - \alpha) J^{\pi_1}(\mu)$ with $0 \leq \alpha \leq 1$, is also a solution to the linear scalarization problem with weight vector $\boldsymbol{w}_J > \mathbf{0}$. Hence, the convex combination of $J$ and $J^{\pi_1}(\mu)$, i.e., the edge connecting $J$ and $J^{\pi_1}(\mu)$, is not strictly dominated by any point in $\mathbb{J}(\mu)$ and thus lies on Pareto front. □

# F    LOCALITY PROPERTY OF BOUNDARY OF $\mathbb{J}(\mu)$ AND PARETO FRONT

This section provides the proofs for locality properties of the boundary of $\mathbb{J}(\mu)$ and the Pareto front. We will ignore the initial state distribution $\mu$ in this section for simplicity of notations.

Before presenting the proofs, we first recall some key notations. We denote the set of long-term returns by policies in $\Pi$ as $\mathbb{J}(\mu)$. By Lemma 1, $\mathbb{J}$ is a convex polytope and vertices are the long-return of deterministic policies. Given a polytope $\mathbb{C}$ and a vertex on the polytope $\boldsymbol{x}$, we denote the set of neighbors on the polytope to $\boldsymbol{x}$ as $\mathrm{N}(\boldsymbol{x}, \mathbb{C})$. Suppose deterministic policy $\pi$ whose long term return $J^\pi$ is a vertex on $\mathbb{J}$, denote the set of deterministic policy whose long-term returns composes $\mathrm{N}(J^\pi, \mathbb{J})$ as $\mathrm{N}_\pi(J^\pi, \mathbb{J})$. Let $\mathrm{Conv}(J^{\Pi_1} \cup \{J^\pi\})$ be the convex hull constructed by $J^{\Pi_1} \cup \{J^\pi\}$. We denote the set of deterministic policies whose long-term returns compose $\mathrm{N}(J^\pi, \mathrm{Conv}(J^{\Pi_1} \cup \{J^\pi\}))$ as $\mathrm{N}_\pi(J^\pi, \mathrm{Conv}(J^{\Pi_1} \cup \{J^\pi\}))$. Let $\Pi_1(\pi)$ be the set of deterministic policies that differ from $\pi$ by only one state-action pair.

**Theorem 6.** *Let the long-term return of deterministic policy $\pi$, i.e., $J^\pi$, be a vertex on $\mathbb{J}$. Then, $\mathrm{N}(J^\pi, \mathrm{Conv}(J^{\Pi_1} \cup \{J^\pi\})) = \mathrm{N}(J^\pi, \mathbb{J})$ and $\mathrm{N}_\pi(J^\pi, \mathrm{Conv}(J^{\Pi_1} \cup \{J^\pi\})) = \mathrm{N}_\pi(J^\pi, \mathbb{J})$.*

*Proof.* It is enough to get $\mathrm{N}(J^\pi, \mathrm{Conv}(J^{\Pi_1} \cup \{J^\pi\})) = \mathrm{N}(J^\pi, \mathbb{J})$ from $\mathrm{N}_\pi(J^\pi, \mathrm{Conv}(J^{\Pi_1} \cup \{J^\pi\})) = \mathrm{N}_\pi(J^\pi, \mathbb{J})$, so we will focus on proving $\mathrm{N}_\pi(J^\pi, \mathrm{Conv}(J^{\Pi_1} \cup \{J^\pi\})) = \mathrm{N}_\pi(J^\pi, \mathbb{J})$.

We will prove $\mathrm{N}_\pi(J^\pi, \mathrm{Conv}(J^{\Pi_1} \cup \{J^\pi\})) = \mathrm{N}_\pi(J^\pi, \mathbb{J})$ in two steps: first prove $\mathrm{N}_\pi(J^\pi, \mathbb{J}) \subseteq \mathrm{N}_\pi(J^\pi, \mathrm{Conv}(J^{\Pi_1} \cup \{J^\pi\}))$, and then prove $\mathrm{N}_\pi(J^\pi, \mathrm{Conv}(J^{\Pi_1} \cup \{J^\pi\})) \subseteq \mathrm{N}_\pi(J^\pi, \mathbb{J})$.

**(1) $\mathrm{N}_\pi(J^\pi, \mathbb{J}) \subseteq \mathrm{N}_\pi(J^\pi, \mathrm{Conv}(J^{\Pi_1} \cup \{J^\pi\}))$.** We prove this by contradiction. Suppose there exists a deterministic policy $u$ such that $u \in \mathrm{N}_\pi(J^\pi, \mathbb{J})$ and $u \notin \mathrm{N}_\pi(J^\pi, \mathrm{Conv}(J^{\Pi_1} \cup \{J^\pi\}))$.

Suppose $u \in \Pi_1(\pi)$. Since $u \in \mathrm{N}_\pi(J^\pi, \mathbb{J})$, $J^u$ lies above all the facets formed by $J^\pi$ and the long-term return of other policies, including the policies in $\Pi_1(\pi)$. Since $u \in \Pi_1(\pi)$, then $J^u$ would also belong to $\mathrm{N}(J^\pi, \mathrm{Conv}(J^{\Pi_1} \cup \{J^\pi\}))$ by the definition of the convex hull, contradicting the assumption that $J^u \notin \mathrm{N}(J^\pi, \mathrm{Conv}(J^{\Pi_1} \cup \{J^\pi\}))$. Hence, $u \notin \Pi_1(\pi)$.

Thus, $u \in \mathrm{N}_\pi(J^\pi, \mathbb{J})$ and $u \notin \Pi_1(\pi)$ simultaneously, which contradicts Theorem 4 that $\mathrm{N}_\pi(J^\pi, \mathbb{J}) \subseteq \Pi_1(\pi)$ almost surely. This causes a contradiction, and therefore, $\mathrm{N}_\pi(J^\pi, \mathbb{J}) \subseteq \mathrm{N}_\pi(J^\pi, \mathrm{Conv}(J^{\Pi_1} \cup \{J^\pi\}))$.

**(2) $\mathrm{N}_\pi(J^\pi, \mathrm{Conv}(J^{\Pi_1} \cup \{J^\pi\})) \subseteq \mathrm{N}_\pi(J^\pi, \mathbb{J})$.** We again proceed by contradiction. Suppose there exists a policy $v$ such that $v \in \mathrm{N}_\pi(J^\pi, \mathrm{Conv}(J^{\Pi_1} \cup \{J^\pi\}))$ and $v \notin \mathrm{N}_\pi(\pi, \mathbb{J})$. We denote the straight line between $J^v$ and $J^\pi$ as $E(J^v, J^\pi)$. Since $v \in \mathrm{N}_\pi(\pi, \mathrm{Conv}(J^{\Pi_1} \cup \{J^\pi\}))$, $E(J^v, J^\pi)$ is an edge of $\mathrm{Conv}(J^{\Pi_1} \cup \{J^\pi\}))$, implying $E(J^{\tilde{\pi}}, J^\pi)$ cannot lie below all facets whose vertices are $J^\pi$ and long-term returns of policies belonging to $\mathrm{N}_\pi(\pi, \mathrm{Conv}(J^{\Pi_1} \cup \{J^\pi\}))$.

Given the condition that $v \notin \mathrm{N}_\pi(\pi, \mathbb{J})$, it follows that $E(J^v, J^\pi)$ is not on the boundary of $\mathbb{J}$, that is, $E(J^v, J^\pi)$ lies below all facets of $\mathbb{J}$, including the facets whose vertices correspond to $\pi$ and policies belonging to $\mathrm{N}_\pi(\pi, \mathbb{J})$. By $\mathrm{N}_\pi(J^\pi, \mathbb{J}) \subseteq \mathrm{N}_\pi(J^\pi, \mathrm{Conv}(J^{\Pi_1} \cup \{J^\pi\}))$ from (1), we have that $E(J^v, J^\pi)$ also lies below all facets whose vertices are $J^\pi$ and the long-term returns of policies belonging to $\mathrm{N}_\pi(\pi, \mathrm{Conv}(J^{\Pi_1} \cup \{J^\pi\}))$. This contradicts the fact that $E(J^{\tilde{\pi}}, J^\pi)$ cannot lie below all facets whose vertices are $J^\pi$ and long-term returns of policies belonging to $\mathrm{N}_\pi(\pi, \mathrm{Conv}(J^{\Pi_1} \cup \{J^\pi\}))$.

The assumption that there exists a policy $v$ such that $v \in \mathrm{N}_\pi(J^\pi, \mathrm{Conv}(J^{\Pi_1} \cup \{J^\pi\}))$ and $v \notin \mathrm{N}_\pi(\pi, \mathbb{J})$ leads a contradiction. Hence, $\mathrm{N}_\pi(\pi, \mathrm{Conv}(J^{\Pi_1} \cup \{J^\pi\})) \subseteq \mathrm{N}_\pi(\pi, \mathbb{J})$.

Combining steps (1) and (2), we have $N_\pi(\pi, \text{Conv}(J^{\Pi_1} \cup \{J^\pi\})) = N_\pi(\pi, \mathbb{J})$ and $N(\pi, \text{Conv}(J^{\Pi_1} \cup \{J^\pi\})) = N(\pi, \mathbb{J})$. This concludes the proof. $\square$

**Lemma 22.** *Let $\mathbb{P}$ be a convex polytope, and let $x$ be a vertex of $\mathbb{P}$. Denote by $\mathbb{V}$ the set of all vertices of $\mathbb{P}$ that are neighbors to $x$. Let $\mathbb{C}$ be the convex hull constructed by $\mathbb{V} \cup \{x\}$, i.e., $\text{Conv}(\mathbb{V} \cup \{x\})$, where $\mathbb{V} \cup \{x\}$ are the vertices of $\mathbb{C}$. Then, the set of faces of $\mathbb{P}$ that intersect at $x$, denoted as $\mathbb{F}(x, \mathbb{P})$, is the same as the set of faces of $\mathbb{C}$ that intersect at $x$, denoted as $\mathbb{F}(x, \mathbb{C})$, i.e., $\mathbb{F}(x, \mathbb{P}) = \mathbb{F}(x, \mathbb{C})$.*

*Proof.* We will prove the two set inclusions: $\mathbb{F}(x, \mathbb{P}) \subseteq \mathbb{F}(x, \mathbb{C})$ and $\mathbb{F}(x, \mathbb{C}) \subseteq \mathbb{F}(x, \mathbb{P})$.

**(1)** $\mathbb{F}(x, \mathbb{P}) \subseteq \mathbb{F}(x, \mathbb{C})$.

Suppose $F$ is a face in $\mathbb{F}(x, \mathbb{P})$. Then, there exists a vector $w$ and a scalar $c$ such that $w^\top x = c$, and the face can be written as $F = \mathbb{P} \cap \{z \in \mathbb{R}^d : w^\top z = c\}$, where the vertices of $F$ are a subset of $\mathbb{V}$. Since $\mathbb{C}$ is the convex hull constructed by $\mathbb{V} \cup \{x\}$, $F$ is a slice contained in $\mathbb{C}$.

For $F$ to be a valid face of $\mathbb{C}$, it must satisfy $w^\top z \leq c$ for all $z \in \mathbb{C}$. We prove this by contradiction: suppose there exists $y \in \mathbb{C}$ such that $w^\top y > c$. Therefore, $y$ lies above all points in $\mathbb{P}$ in the direction of $w$. Since $\mathbb{C}$ is a subset of $\mathbb{P}$ (because $\mathbb{C}$ is the convex hull of a subset of the vertices of $\mathbb{P}$), this would imply that $y$ lies above all points in $\mathbb{C}$ in the direction of $w$, which contradicts the assumption that $y \in \mathbb{C}$.

Therefore, $F$ is also a valid face of $\mathbb{C}$, implying $\mathbb{F}(x, \mathbb{P}) \subseteq \mathbb{F}(x, \mathbb{C})$.

**(2)** $\mathbb{F}(x, \mathbb{C}) \subseteq \mathbb{F}(x, \mathbb{P})$.

The argument is analogous to part (1). Suppose $F$ is a face of $\mathbb{C}$ that intersects at $x$. Then $F$ can be written as $F = \mathbb{C} \cap \{y \in \mathbb{R}^d : w^\top y = c\}$, where $w^\top x = c$, and $w^\top y \leq c$ for any $y \in \mathbb{C}$. With the same analysis as in (1), $F$ is a slice contained in $\mathbb{C}$. Since any point $z \in \mathbb{P}$ can be expressed as a linear combination of $x$ and its neighboring vertices $v_i \in \mathbb{V}$, we have:

$$w^\top z = w^\top \left( x + \sum_i \alpha_i (v_i - x) \right) \leq c,$$

where $\alpha_i > 0$, and $w^\top x = c$ and $w^\top v_i \leq c$. Therefore, $F$ is also a face of $\mathbb{P}$.

Combining (1) and (2), we conclude that $\mathbb{F}(x, \mathbb{P}) = \mathbb{F}(x, \mathbb{C})$. $\square$

**Proposition 3.** *Let the long-term return of deterministic policy $\pi$, i.e., $J^\pi$, be a vertex on $\mathbb{J}$. Let $\mathbb{F}(J^\pi, \text{Conv}(J^{\Pi_1} \cup \{J^\pi\})$ be the set of faces of the convex hull constructed by $J^{\Pi_1} \cup \{J^\pi\}$ that intersect at $J^\pi$. Similarly, let $\mathbb{F}(J^\pi, \mathbb{J})$ be the set of faces of $\mathbb{J}$ that intersect at $J^\pi$. Then, $\mathbb{F}(J^\pi, \text{Conv}(J^{\Pi_1} \cup \{J^\pi\})) = \mathbb{F}(J^\pi, \mathbb{J})$.*

*Proof.* By Theorem 6, we know that $N(J^\pi, \text{Conv}(J^{\Pi_1} \cup \{J^\pi\})) = N(J^\pi, \mathbb{J})$. According to Lemma 22, if two sets of vertices have the same set of neighbor vertices, then the sets of faces formed by those vertices are identical. Hence, we conclude that $\mathbb{F}(J^\pi, \text{Conv}(J^{\Pi_1} \cup \{J^\pi\})) = \mathbb{F}(J^\pi, \mathbb{J})$. $\square$

**Proposition 4.** *Let $\pi$ be a policy on the Pareto front of $\mathbb{J}$. Let $\Pi_{1,ND}(\pi) \subseteq \Pi_1(\pi)$ be the set of policies belonging to $\Pi_1(\pi)$ that are not dominated by any other policies in $\Pi_1(\pi)$, and let $N(J^\pi, \text{Conv}(J^{\Pi_{1,ND}} \cup \{J^\pi\}))$ be the set of neighbors of $J^\pi$ in the convex hull constructed by $J^{\Pi_{1,ND}} \cup \{J^\pi\}$. Similarly, let $N(J^\pi, \mathbb{P}(\mathbb{J}))$ be the set of neighbors of $J^\pi$ on the Pareto front of $\mathbb{J}$. Then we have*

$$N(J^\pi, \mathbb{P}(\mathbb{J})) \subseteq N(J^\pi, \text{Conv}(J^{\Pi_{1,ND}} \cup \{J^\pi\})).$$

*Proof.* By Lemma 7, the Pareto front lies on the boundary of $\mathbb{J}$. Therefore, the neighbors of $\pi$ on the Pareto front, $N(J^\pi, \mathbb{P}(\mathbb{J}))$, are the intersection of the set of vertices on $\mathbb{J}$ neighboring $J^\pi$ and the Pareto front. Formally, we have $N(J^\pi, \mathbb{P}(\mathbb{J})) = N(J^\pi, \mathbb{J}) \cap \mathbb{P}(\mathbb{J})$.

Define $\text{ND}(\Pi_1)$ as the set of all policies (including stochastic and deterministic policies) that are not dominated by any other policies in $\Pi_1$. Since the Pareto front consists of the long-term return of policies that are not dominated by any other policies in $\Pi$, it follows that $\mathbb{P}(\mathbb{J}) \subseteq J^{\text{ND}(\Pi_1)}$. Thus, we have $N(J^\pi, \mathbb{P}(\mathbb{J})) \subseteq N(J^\pi, \mathbb{J}) \cap J^{\text{ND}(\Pi_1)}$.

By Theorem 6, we know that $N(J^\pi, \text{Conv}(J^{\Pi_1} \cup \{J^\pi\})) = N(J^\pi, \mathbb{J})$. Therefore, combining with $N(J^\pi, \mathbb{P}(\mathbb{J})) \subseteq N(J^\pi, \mathbb{J}) \cap J^{\text{ND}(\Pi_1)}$, we can conclude:

$$N(J^\pi, \mathbb{P}(\mathbb{J})) \subseteq N(J^\pi, \text{Conv}(J^{\Pi_1} \cup \{J^\pi\})) \cap J^{\text{ND}(\Pi_1)}.$$

Next, consider any point $J^u \in \mathrm{N}(J^\pi, \mathrm{Conv}(J^{\Pi_1} \cup \{J^\pi\})) \cap J^{\mathrm{ND}(\Pi_1)}$. By $\mathrm{N}(J^\pi, \mathrm{Conv}(J^{\Pi_1} \cup \{J^\pi\})) \subseteq J^{\Pi_1}$, we have $J^u \in J^{\Pi_1} \cap J^{\mathrm{ND}(\Pi_1)}$. This means that $u$ is in $\Pi_1(\pi)$ and is not dominated by any policy in $\Pi_1(\pi)$, thus $u \in \Pi_{1,\mathrm{ND}}(\pi)$. And since $J^u$ is also a vertex of the convex hull constructed by $J^{\Pi_1} \cup \{J^\pi\}$, it does not lie below all facets formed by the long-term return of any other policies belonging to $\Pi_1(\pi)$ and $\pi$. Since $\Pi_{1,\mathrm{ND}}(\pi) \subseteq \Pi_1(\pi)$, it follows that $J^u$ does not lie below all facets formed by the long-term return of any other policies in $\Pi_{1,\mathrm{ND}}(\pi)$ and $\pi$. Therefore, $J^u \in \mathrm{N}(J^\pi, \mathrm{Conv}(J^{\Pi_{1,\mathrm{ND}}} \cup \{J^\pi\}))$. Thus, we have:

$$\mathrm{N}(\pi, \mathrm{Conv}(J^{\Pi_1} \cup \{J^\pi\})) \cap J^{\mathrm{ND}(\Pi_1)} \subseteq \mathrm{N}(J^\pi, \mathrm{Conv}(J^{\Pi_{1,\mathrm{ND}}} \cup \{J^\pi\})).$$

Combining this with the previous result, we conclude:

$$\mathrm{N}(J^\pi, \mathbb{P}(\mathbb{J})) \subseteq \mathrm{N}(J^\pi, \mathrm{Conv}(J^{\Pi_{1,\mathrm{ND}}} \cup \{J^\pi\})).$$

Then we also have $\mathrm{N}_\pi(J^\pi, \mathbb{P}(\mathbb{J})) \subseteq \mathrm{N}_\pi(J^\pi, \mathrm{Conv}(J^{\Pi_{1,\mathrm{ND}}} \cup \{J^\pi\}))$. $\qquad\square$

**Proposition 5.** *Let $\pi$ be a policy on the Pareto front of $\mathbb{J}$. Let $\Pi_{1,ND}(\pi) \subseteq \Pi_1(\pi)$ be the set of policies belonging to $\Pi_1(\pi)$ that are not dominated by any other policies in $\Pi_1(\pi)$, and let $\mathbb{F}(J^\pi, \mathrm{Conv}(J^{\Pi_{1,ND}} \cup \{J^\pi\}))$ be the set of faces of the convex hull constructed by $J^{\Pi_{1,ND}} \cup \{J^\pi\}$ that intersect at $J^\pi$. Similarly, let $\mathbb{F}(J^\pi, P(\mathbb{J}))$ be the set of faces on the Pareto front of $\mathbb{J}$ that intersect at $J^\pi$. Then, we have:*

$$\mathbb{F}(J^\pi, P(\mathbb{J})) \subseteq \mathbb{F}(J^\pi, \mathrm{Conv}(J^{\Pi_{1,ND}} \cup \{J^\pi\})).$$

**Remark 1.** *Note that in this case, the faces of the convex hull $\mathrm{Conv}(\Pi_{1,ND} \cup \{x\})$ that intersect at $x$ may not necessarily lie on $\mathbb{J}$. However, we can use the condition $\mathrm{N}(\pi, \mathbb{P}(\mathbb{J})) \subseteq \mathrm{N}(\pi, \mathrm{Conv}(\Pi_{1,ND}))$ from Proposition 4 and the characteristics of the Pareto front from Theorem 5 to show that faces of the convex hull $\mathrm{Conv}(\Pi_{1,ND} \cup \{x\})$ that intersect at $x$ lie on the Pareto front of $\mathbb{J}$.*

*Proof.* Since $\mathrm{N}_\pi(J^\pi, \mathrm{Conv}(J^{\Pi_{1,\mathrm{ND}}} \cup \{J^\pi\})) \in \Pi_D$, then $\mathrm{N}(J^\pi, \mathrm{Conv}(J^{\Pi_{1,\mathrm{ND}}} \cup \{J^\pi\})) \subseteq \mathbb{J}$. By Proposition 4, we have $\mathrm{N}(J^\pi, \mathbb{P}(\mathbb{J})) \subseteq \mathrm{N}(J^\pi, \mathrm{Conv}(J^{\Pi_{1,\mathrm{ND}}} \cup \{J^\pi\}))$. By Theorem 5, the set of faces in Pareto front constructed by $J^\pi$ and vertices belonging to $\mathrm{N}(J^\pi, \mathbb{P}(\mathbb{J}))$ is a subset of the set of faces of the convex hull constructed by $J^{\Pi_{1,\mathrm{ND}}} \cup \{J^\pi\}$, i.e., $\mathbb{F}(J^\pi, P(\mathbb{J})) \subseteq \mathbb{F}(J^\pi, \mathrm{Conv}(J^{\Pi_{1,\mathrm{ND}}} \cup \{J^\pi\}))$. This concludes our proof.

$\qquad\square$

**Proposition 6.** *(Restatement of Proposition 1)* *Let $\mathbb{F}_P(J^\pi, \mathrm{Conv}(J^{\Pi_{1,ND}} \cup \{J^\pi\}))$ denote the set of faces of $\mathbb{F}(J^\pi, \mathrm{Conv}(J^{\Pi_{1,ND}} \cup \{J^\pi\}))$ that satisfying Lemma 3. Then, we have*

$$\mathbb{F}(J^\pi, P(\mathbb{J})) = \mathbb{F}_P(J^\pi, \mathrm{Conv}(J^{\Pi_{1,ND}} \cup \{J^\pi\}))$$

*Proof.* We prove this by two inclusions: $\mathbb{F}(J^\pi, P(\mathbb{J})) \subseteq \mathbb{F}_P(J^\pi, \mathrm{Conv}(J^{\Pi_{1,\mathrm{ND}}} \cup \{J^\pi\}))$ and $\mathbb{F}_P(J^\pi, \mathrm{Conv}(J^{\Pi_{1,\mathrm{ND}}} \cup \{J^\pi\})) \subseteq \mathbb{F}(J^\pi, P(\mathbb{J}))$.

(1) $\mathbb{F}(J^\pi, P(\mathbb{J})) \subseteq \mathbb{F}_P(J^\pi, \mathrm{Conv}(J^{\Pi_{1,\mathrm{ND}}} \cup \{J^\pi\}))$. Let $F \in \mathbb{F}(J^\pi, P(\mathbb{J}))$. Since Proposition 5 proves that $\mathbb{F}(J^\pi, P(\mathbb{J})) \subseteq \mathbb{F}(J^\pi, \mathrm{Conv}(J^{\Pi_{1,\mathrm{ND}}} \cup \{J^\pi\}))$, we have $F \in \mathbb{F}(J^\pi, \mathrm{Conv}(J^{\Pi_{1,\mathrm{ND}}} \cup \{J^\pi\}))$.

Since $F$ is a face on the Pareto front of $\mathbb{J}$ and $\mathrm{Conv}(J^{\Pi_{1,\mathrm{ND}}} \cup \{J^\pi\})$ is a subset of $\mathbb{J}$, $F$ is also the Pareto front of $\mathrm{Conv}(J^{\Pi_{1,\mathrm{ND}}} \cup \{J^\pi\})$. Therefore, $F$ satisfies Lemma 3 in $\mathrm{Conv}(J^{\Pi_{1,\mathrm{ND}}} \cup \{J^\pi\})$ and $F \in \mathbb{F}_P(J^\pi, \mathrm{Conv}(J^{\Pi_{1,\mathrm{ND}}} \cup \{J^\pi\}))$. Thus, we have $\mathbb{F}(J^\pi, P(\mathbb{J})) \subseteq \mathbb{F}_P(J^\pi, \mathrm{Conv}(J^{\Pi_{1,\mathrm{ND}}} \cup \{J^\pi\}))$.

(2) $\mathbb{F}_P(J^\pi, \mathrm{Conv}(J^{\Pi_{1,\mathrm{ND}}} \cup \{J^\pi\})) \subseteq \mathbb{F}(J^\pi, P(\mathbb{J}))$. Let $F \in \mathbb{F}_P(J^\pi, \mathrm{Conv}(J^{\Pi_{1,\mathrm{ND}}} \cup \{J^\pi\}))$. By Lemma 8, there exists a weight vector $\boldsymbol{w} > 0$ such that $\boldsymbol{w}^\top (\boldsymbol{x} - \boldsymbol{y}) \geq 0$ for any $\boldsymbol{x} \in F$ and $\boldsymbol{y} \in \mathrm{Conv}(J^{\Pi_{1,\mathrm{ND}}} \cup \{J^\pi\})$. Since $J^{\Pi_{1,\mathrm{ND}}} \subseteq \mathrm{Conv}(J^{\Pi_{1,\mathrm{ND}}} \cup \{J^\pi\})$, the inequality also holds for any $\boldsymbol{x} \in F$ and $\boldsymbol{y} \in J^{\Pi_{1,\mathrm{ND}}}$. For any $\boldsymbol{z} \in J^{\Pi_1}$, $\boldsymbol{z}$ either belongs to $J^{\Pi_{1,\mathrm{ND}}}$ or is dominated by $\boldsymbol{y} \in J^{\Pi_{1,\mathrm{ND}}}$. In both cases, $\boldsymbol{w}^\top (\boldsymbol{x} - \boldsymbol{z}) \geq 0$ implying $\boldsymbol{x}$ is not dominated by any $\boldsymbol{z} \in J^{\Pi_1}$. Thus, $F$ is a Pareto optimal face of $\mathrm{Conv}(J^{\Pi_1} \cup \{J^\pi\})$. Suppose that $F$ written as the intersection of $n \geq 1$ facets, i.e., $F = \cap_{i=1}^n F_i$, where $F_i = \mathrm{Conv}(J^{\Pi_1} \cup \{J^\pi\}) \cap \{\boldsymbol{x} \in \mathbb{R}^d : \boldsymbol{w}_i^\top \boldsymbol{x} = c_i\}$, we can express $F$ as:

$$F = \mathrm{Conv}(J^{\Pi_1} \cup \{J^\pi\}) \cap \left\{ \boldsymbol{x} \in \mathbb{R}^d : \sum_i \boldsymbol{\alpha}_i \boldsymbol{w}_i^\top \boldsymbol{x} = \sum_i \boldsymbol{\alpha}_i c_i \right\},$$

where for any $\boldsymbol{\alpha} \geq 0$. As $F = \mathrm{Conv}(J^{\Pi_1} \cup \{J^\pi\}) \cap \{\boldsymbol{x} \in \mathbb{R}^d : \boldsymbol{w}^\top \boldsymbol{x} = c\}$, there exists a non-negative $\boldsymbol{\alpha}$ such that $\boldsymbol{w} = \sum_i \boldsymbol{\alpha}_i \boldsymbol{w}_i > \boldsymbol{0}$. That is, there exists a non-negative linear combination of these facets $\mathrm{Conv}(J^{\Pi_1} \cup \{J^\pi\})$ that intersects on $J^\pi$ such that the linear combination is strictly positive.

As $\mathbb{F}(\mathrm{Conv}(J^{\Pi_1} \cup \{J^\pi\})) = \mathbb{F}(J^\pi, \mathbb{J})$ by Proposition 3, there exists a non-negative linear combination of facets of $\mathbb{F}(J^\pi, \mathbb{J})$ that intersects on $J^\pi$ such that the combination is strictly positive. By Lemma 3 again, $F \in \mathbb{F}(J^\pi, P(\mathbb{J}))$.

Combining (1) and (2) concludes the proof. $\qquad\square$

## G   FIND PARETO FRONT ON CCS

By Lemma 9, each nonempty face $\mathbb{F}$ of a convex polytope can be written as the intersection of facets $\{\mathbb{F}_i\}_i$, i.e., $\mathbb{F} = \cap_i(\mathbb{F}_i)$.

**Lemma 23.** *(Restatement of Lemma 3) Given a convex polytope $\mathbb{C}$, a face $\mathbb{F}$ of $\mathbb{C}$ lies on the Pareto front if and only if the following conditions hold:*

- *When $\mathbb{F}$ is a facet of $\mathbb{C}$, i.e., $dim(\mathbb{F}) = dim(\mathbb{C}) - 1$, there exists $\boldsymbol{w} > \mathbf{0}$ such that $\mathbb{F} = \mathbb{C} \cap \{\boldsymbol{x} \in \mathbb{R}^d : \boldsymbol{w}^\top \boldsymbol{x} = c_0\}$.*

- *When $0 \leq dim(\mathbb{F}) < dim(\mathbb{C}) - 1$, and $\mathbb{F}$ is the intersection of facets of $\mathbb{C}$, i.e., $\mathbb{F} = \cap_i \mathbb{F}_i$, where each facet is written as $\mathbb{F}_i = \mathbb{C} \cap \{\boldsymbol{x} \in \mathbb{R}^d : \boldsymbol{w}_i^\top \boldsymbol{x} = c_i\}$, there exists a linear combination of the normal vectors with weight vector $\boldsymbol{\alpha} \geq 0$, such that $\sum_i \boldsymbol{\alpha}_i \boldsymbol{w}_i > 0$.*

*Proof.* When $\mathbb{F} = \mathbb{C} \cap \{\boldsymbol{x} \in \mathbb{R}^d : \boldsymbol{w}^\top \boldsymbol{x} = c_0\}$ is a facet of $\mathbb{C}$, by the definition of a valid linear inequality, the inequality $\boldsymbol{w}^\top \boldsymbol{y} \leq c_0$ holds for all points $\boldsymbol{y} \in \mathbb{C}$. Since $\boldsymbol{w} > 0$, for any $\boldsymbol{x}$ on $\mathbb{F}$ and any $\boldsymbol{y} \in \mathbb{C}$, we have $\boldsymbol{w}^\top(\boldsymbol{y} - \boldsymbol{x}) \leq 0$. By Lemma 8, this is equivalent to that no point in $\mathbb{C}$ dominates $\boldsymbol{x}$ and $\mathbb{F}$ lies on the Pareto front.

When $\mathbb{F}$ is a lower-dimensional face of $\mathbb{C}$, written as the intersection of $n$ facets, i.e., $\mathbb{F} = \cap_{i=1}^n \mathbb{F}_i$, where $\mathbb{F}_i = \mathbb{C} \cap \{\boldsymbol{x} \in \mathbb{R}^d : \boldsymbol{w}_i^\top \boldsymbol{x} = c_i\}$, we can express $\mathbb{F}$ as:

$$\mathbb{F} = \mathbb{C} \cap \left\{\boldsymbol{x} \in \mathbb{R}^d : \sum_i \boldsymbol{\alpha}_i \boldsymbol{w}_i^\top \boldsymbol{x} = \sum_i \boldsymbol{\alpha}_i c_i\right\},$$

where $\boldsymbol{\alpha} \geq 0$. Since $\sum_i \boldsymbol{\alpha}_i \boldsymbol{w}_i > 0$, for any $\boldsymbol{x}$ on $\mathbb{F}$ and any $\boldsymbol{y} \in \mathbb{C}$, there exists a vector $\boldsymbol{w} > 0$ such that $\boldsymbol{w}^\top(\boldsymbol{y} - \boldsymbol{x}) \leq 0$. Therefore, by Lemma 8, this is equivalent to no point in $\mathbb{C}$ strictly dominates $\boldsymbol{x}$ and $\mathbb{F}$ lies on the Pareto front.

$\square$

**Lemma 24.** *Let $F$ be a face of $\mathbb{J}$, and let $\Pi_F$ be the set of deterministic policies corresponding to the vertices of $F$. Then $F$ can be constructed by the long-term returns of the convex combinations of $\Pi_F$, i.e.,*

$$F = \left\{J^\pi \middle| \pi(a|s) = \sum_{\pi_i \in \Pi_F} \boldsymbol{\alpha}_i \pi_i(a|s), \boldsymbol{\alpha} \geq 0, \sum_i \boldsymbol{\alpha}_i = 1\right\}.$$

*Proof.* By Lemma 1, $\mathbb{J}$ is a convex polytope, and its vertices correspond to deterministic policies. Since $F$ is a face of the convex polytope $\mathbb{J}$, by the definition of a face, there exist $\boldsymbol{w}$ and $c$ such that $F = \{\boldsymbol{x} \in \mathbb{R}^D \mid \boldsymbol{w}^\top \boldsymbol{x} = c\} \cap \mathbb{J}$, and for any $\boldsymbol{y} \in \mathbb{J}, \boldsymbol{w}^\top \boldsymbol{y} \leq c$.

Therefore, $\Pi_F = \arg\max_{\pi \in \Pi_D} \boldsymbol{w}^\top J^\pi$, where $\Pi_D$ denotes the set of all deterministic policies. By Lemma 14, the convex hull of $\Pi_D$, i.e., $\{\pi \mid \pi(a|s) = \sum_{\pi_i \in \Pi_F} \boldsymbol{\alpha}_i \pi_i(a|s), \boldsymbol{\alpha} \geq 0, \sum_i \boldsymbol{\alpha}_i = 1\}$, constructs all policies (including stochastic and deterministic) that maximize $\boldsymbol{w}^\top J^\pi$. Thus, the long-term returns of $\{\pi \mid \pi(a|s) = \sum_{\pi_i \in \Pi_F} \boldsymbol{\alpha}_i \pi_i(a|s), \boldsymbol{\alpha} \geq 0, \sum_i \boldsymbol{\alpha}_i = 1\}$ construct $F$. Formally, $F = \{J^\pi \mid \pi(a|s) = \sum_{\pi_i \in \Pi_F} \boldsymbol{\alpha}_i \pi_i(a|s), \boldsymbol{\alpha} \geq 0, \sum_i \boldsymbol{\alpha}_i = 1\}$. This concludes the proof. $\square$

## H   COMPLEXITY COMPARISON BETWEEN OLS AND ALGORITHM 1

Let $S$, $A$, and $D$ represent the state space size, the action space size, and the reward dimension, respectively. Let $N$ denote the number of vertices on the Pareto front.

For each iteration, OLS solves the single-objective MDP with a preference vector, removes obsolete preference vectors with the new points, and computes new preference vectors. Solving the single-objective planning problem incurs a complexity of $\mathcal{O}(C_{\text{so}} + C_{\text{pe}})$, where $C_{\text{so}}$ and $C_{\text{pe}}$ represent the computational complexity of single-objective MDP solver and policy evaluation, respectively. The complexity of comparing with the previous candidate Pareto optimal policies and removing obsolete ones is $\mathcal{O}(ND)$, and the complexity of calculating a new preference vector is denoted as $C_{\text{we}}$. The computational complexity of OLS is $\mathcal{O}(N^2 D + NC_{\text{we}} + NC_{\text{so}} + NC_{\text{pe}})$.

Algorithm 1 requires only a one-time single-objective planning step during the initialization phase to obtain the initial deterministic Pareto-optimal policy. The overall computational complexity of our algorithm is $\mathcal{O}(C_{\text{so}} + NSAC_{\text{pe}} +$

$NC_{\mathrm{ND}} + NC_{\mathrm{cvx}})$, where $C_{\mathrm{ND}}$ and $C_{\mathrm{cvx}}$ represent the computational complexity to select the non-dominated policies and computing faces of the convex hull, respectively.

Specifically, we dive deeper into the computational complexity of each module.

- $C_{\mathrm{so}}$. The computational complexity of single-objective planning is $C_{\mathrm{so}} = \mathcal{O}\left(S^4 A + S^3 A^2\right)$.

- $C_{\mathrm{pe}}$. The computational complexity of the policy evaluation is $C_{\mathrm{pe}} = \mathcal{O}(S^3 D)$.

- $C_{\mathrm{ND}}$. $C_{\mathrm{ND}}$ refers to the complexity of finding $\Pi_{1,\mathrm{ND}}$, the set of non-dominated policies from $\Pi_1$. Let $N_{\mathrm{ND}} = |\Pi_{1,\mathrm{ND}}|$. The non-dominated policies searching complexity is $C_{\mathrm{ND}} = \mathcal{O}(SAN_{\mathrm{ND}}D)$.

- $C_{\mathrm{cvx}}$. $C_{\mathrm{cvx}}$ is the complexity of computing the convex hull faces formed by $N_{\mathrm{ND}} + 1$ points that intersect at the current policy. If the number of vertices of the output convex hull is $N_{\mathrm{cvx}}$. By [1], the complexity of the calculation of convex hull is $\mathcal{O}(N_{\mathrm{ND}} \log N_{\mathrm{cvx}})$ when $D \leq 3$ and $\mathcal{O}(N_{\mathrm{ND}} f_{N_{\mathrm{cvx}}}/N_{\mathrm{cvx}})$, where $f_L = \mathcal{O}(L^{\lfloor D/2 \rfloor}/(\lfloor D/2 \rfloor!))$.

As Algorithm 1 directly searches the Pareto front, the total number of iterations corresponds to the number of vertices on the Pareto front, avoiding the quadratic complexity introduced by the preference vector updates in OLS. Moreover, note that calculating new weight in OLS requires calculating the vertices of the convex hull constructed by $N + D + 2$ hyperplanes of dimension $D + 1$, making it significantly more expensive than the local convex hull construction of neighboring policies in our proposed algorithm. Hence, $C_{we}$ is much larger than $C_{\mathrm{cvx}}$.

In conclusion, our algorithm induces total computational complexity of $\mathcal{O}(S^4 A + S^3 A^2 + NS^4 A + NSAN_{\mathrm{ND}}D + C_{\mathrm{cvx}})$, while OLS has computation complexity $\mathcal{O}(N(S^4 A + S^3 A^2) + NS^3 + N^2 D + NC_{we})$. Given $C_{\mathrm{cvx}} < C_{we}$, our algorithm is more efficient than OLS in all cases.

# I SUPPORTING ALGORITHMS

For completeness, we provide the PPrune algorithm (Roijers, 2016), which identifies the non-dominated vector value functions $\mathbb{V}^*$ from a given set of vector value functions $\mathbb{V}$. Note that, with slight abuse of notation, $\mathbb{V}$ here refers to the set of vector value functions rather than the set of vertices used in the main text.

---

**Algorithm 3** PPrune($\mathbb{V}$) (Roijers, 2016)

> **Input:** A set of value vectors $\mathbb{V}$
> 1: $\mathbb{V}^* \leftarrow \varnothing$
> 2: **while** $\mathbb{V} \neq \varnothing$ **do**
> 3:     $V \leftarrow$ the first element of $\mathbb{V}$
> 4:     **for all** $V' \in \mathbb{V}$ **do**
> 5:         **if** $V' \succ V$ **then**
> 6:             $V \leftarrow V'$
> 7:         **end if**
> 8:     **end for**
> 9:     Remove $V$ and all vectors dominated by $V$ from $\mathbb{V}$
> 10:    Add $V$ to $\mathbb{V}^*$
> 11: **end while**
> 12: **return** $\mathbb{V}^*$

---

The benchmark algorithm retrieves the Pareto front in two steps: (1) evaluate all deterministic policies and compute the convex hull of all non-dominated deterministic policies and (2) apply Algorithm 2 to identify the Pareto front. The details of the benchmark algorithm are shown in Algorithm 4.

# J EXPERIMENT RESULTS

We adopt the deep-sea-treasure setting from Yang et al. (2019a) and construct a scenario where the agent navigates a grid world, making trade-offs between different objectives. At each step, the agent can move up, down, left, or right, and the reward for each grid is generated randomly. We compare the performance of the OLS algorithm and the proposed Pareto front searching algorithm under different grid sizes (varying numbers of rows and columns). The experiments

---

**Algorithm 4** Benchmark Algorithm

**Input:** MDP settings: $(\mathcal{S}, \mathcal{A}, \mathbf{P}, \boldsymbol{r}, \gamma)$
**Output:** the set of Pareto optimal faces $\mathbb{P}(\Pi)$ and the set of deterministic Pareto optimal policies $\mathbb{V}(\Pi)$
1: Generate all deterministic policies $\Pi_D$.
2: Select non-dominated policies $\Pi_{D,\text{ND}}$ from $\Pi_D$.
3: Calculate the convex hull formed by $\{J^\pi \mid \pi \in \Pi_{D,\text{ND}}\}$.
4: Extract Pareto optimal faces $\mathbb{P}(\Pi)$ and vertices $\mathbb{V}(\Pi)$ from the convex hull with Algorithm 2.

---

were conducted on smaller grid sizes to ensure that the running time of OLS remained manageable. As shown in the Fig. 7, the proposed algorithm consistently outperforms OLS in terms of efficiency across all tested settings.

We evaluate the trend of the number of vertices $N$ in terms of $S$ and $A$ with the state space size ranging from 10 to 50 and the action space sizes ranging from 10 to 20. The reward dimension is 3. As shown in Fig. 8, the result shows that the number of vertices grows approximately as $N \approx kS^3A$, where $k$ is a constant number. Combined with the computational complexity analysis $\mathcal{O}(S^4A + S^3A^2 + NS^4A + NSAN_{\text{ND}}D + C_{\text{cvx}})$, this demonstrates the scalability of our algorithm.

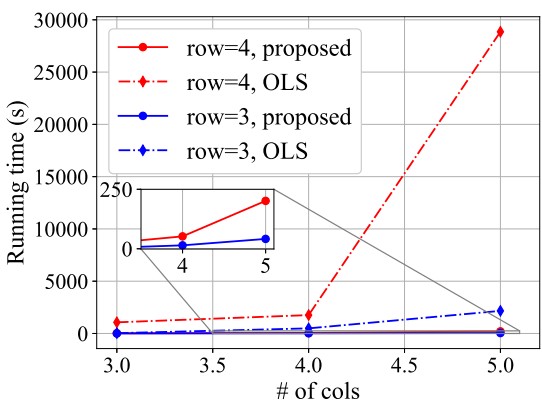
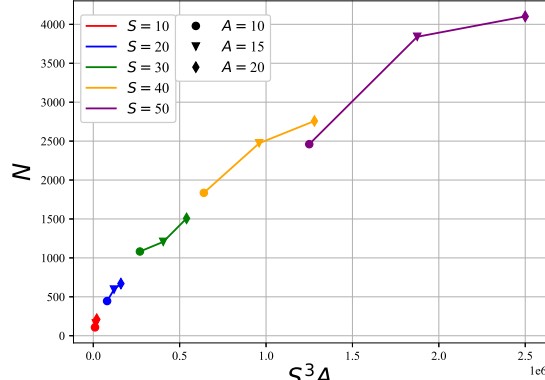

Figure 7: Pareto front of a simple MDP with $S = 4$, $A = 3$, and $D = 3$.

Figure 8: Comparison between the proposed Pareto front searching algorithm and the benchmark algorithm when $D = 3$.

