# OpenReview forum: "How to Find the Exact Pareto Front for Multi-Objective MDPs?"
_ICLR.cc/2025/Conference — ICLR 2025 Spotlight_

### Official Review · Reviewer_4Q5J · 2024-10-22

**Soundness:** 4
**Presentation:** 4
**Contribution:** 3
**Rating:** 8
**Confidence:** 4

**Summary:**

This paper is concerned with characterizing optimal Pareto front and a corresponding algorithm to efficiently build the optimal Pareto front for Multi-Objective MDPs. In contrast with the existing works using preference vector and traversing continuous space, this paper characterizes Pareto front using deterministic policies, and elucidating the relationship between neighboring vertices. The authors present the relevant algorithms to extract Pareto optimal faces. The numerical experiments and detailed proofs were provided.

**Strengths:**

I really like the presentation of this paper. Sections 4.1, 4.2, and 4.3 sequentially validate the algorithm in a very clear way. The figures help understand the characterization of the vertices and faces, and the algorithm description is very clear.
The idea of identifying the vertices as deterministic policies through a linear transformation for Multi-Objective MDPs is impressive.
I believe that the concept of traversing the vertices (deterministic policies) is clearly distinguished from the previous works which typically uses approximation while searching continuous spaces.

**Weaknesses:**

I doubt if the author’s algorithm is really more efficient than the OLS in all cases. Generally, dealing with discrete sets such as the author's approach takes longer than handling continuous sets. I understand that the experiment demonstrates the efficiency of the algorithm, but this was done when S and A are small enough. For example, in Line 496, the author had clarified the complexity of the algorithm which is $O(NS(A-1))$, but if the state/action spaces are large, $S(A-1)$ may be larger than $N$ and it may have a higher complexity than $O(N^2)$. Can you present why the above situation may not be the case? If impossible, I suggest the authors discuss trade-offs between N, S, and A, and preferable scenarios where the algorithm would work well (e.g. propose certain situations where $N$ >> $S(A-1)$ and why).

**Questions:**

1. Line 905 says that $\Pi$ can contain non-stationary policies, while Line 684 defines $\Pi$ as a set of stationary policies. I am confused about whether the authors only consider stationary policies or not. Can you explain any intentional differences if they exist?

2. What is $L \Pi_D(\mu)$ in Line 749? Is it $\Pi_D (\mu)$? Or does it mean that the vertices are the occupancy measures of a set of policies $\Pi_D (\mu)$? (If it is a separate concept, the authors should define what $L\Pi$ is.)

3. To recover “full” exact Pareto front, I believe that the “if” statement in Lemma 3 should be modified to “if and only if” (necessary and sufficient). If you intended to do so, can you modify that? (It seems that Line 409 talks about “only if” part, but it is not accurately stated in the lemma.) If not, please explain why the sufficient conditions are enough to prove the following theorems or propositions.
Also, I cannot locate the proof of Lemma 3.

4. I think the point of Lines 10-11 in Algorithm 2 is to include all normal vectors associated with lower-dimensional faces to apply (1). If that’s the case, shouldn’t Line 4 of Algorithm 2 specify that .pop() be done from high-dimensional faces to lower-dimensional faces (or specify that .push() is done in the order of dimension)? If the order matters, please modify the algorithm accordingly and if it doesn't, please explain why this is the case.

5. It would be better to add relevant existing algorithms in the appendix (e.g. Roijers, PPrune algorithm) for the completeness.

6. It would be better to note the corresponding proof locations below all theorems (e.g. Theorem 2, Proposition 1).

7. In the abstract, it would be better to use MO-MDPs inside the first parenthesis instead of MDPs to understand the 12th line of the abstract (I personally didn't get it at first glance).

I am giving the score of 6 for the current draft, but I am eager to talk with the authors and reconsider the score.

---

> ### Author Response · Authors · 2024-11-20
> **Response to Reviewer 4Q5J [1/2]**
>
> We have uploaded the revised paper, with the major changes highlighted in blue.
>
> > I doubt if the author’s algorithm is really more efficient than the OLS in all cases. Generally, dealing with discrete sets such as the author's approach takes longer than handling continuous sets. I understand that the experiment demonstrates the efficiency of the algorithm, but this was done when S and A are small enough. For example, in Line 496, the author had clarified the complexity of the algorithm which is O(NS(A-1)), but if the state/action spaces are large, S(A-1) may be larger than N and it may have a higher complexity than O(N^2). Can you present why the above situation may not be the case? If impossible, I suggest the authors discuss trade-offs between N, S, and A, and preferable scenarios where the algorithm would work well (e.g. propose certain situations where N >>S(A-1) and why).
>
> Thanks for your question.
> We first want to clarify that our algorithm is designed to find the entire Pareto front, including stochastic policies, whereas OLS focuses solely on finding deterministic Pareto optimal policies. Moreover, even when restricted to finding deterministic Pareto optimal policies, our algorithm consistently outperforms OLS in terms of computational efficiency.
>
> In our earlier complexity analysis, we did not fully specify the computational complexities of the policy evaluation and optimization steps, which are related to the MDP size. Both complexities of OLS and our algorithm are related to the MDP sizes ($S$ and $A$) and the number of vertices ($N$). It is important to note that solving a single-objective MDP (policy optimization) has a significantly higher computational cost than policy evaluation. OLS requires solving a single-objective MDP at every iteration, whereas our algorithm solves it only once during the initialization phase. This difference significantly impacts the overall computational cost.
> In the following, we compare the computation complexity of finding deterministic Pareto optimal policies using OLS and our algorithm.
>
> Let $S$, $A$, and $D$ represent the state space size, the action space size, and the reward dimension, respectively.  Let $N$ denote the number of vertices on the Pareto front. The overall computational complexity of our algorithm is $\mathcal{O}(C_{\text{so}}+NSAC_{\text{pe}}+NC_{\text{ND}}+NC_{\text{cvx}})$, where the terms are defined as follows:
>
> (1) $C_{\text{so}}$: The computational complexity of single-objective planning is $C_{\text{so}} = \mathcal{O}\left(S^4A+S^3A^2\right)$;
>
> (2) $C_{\text{pe}}$: The computational complexity of the policy evaluation is $C_{\text{pe}} = \mathcal{O}(S^3D)$;
>
> (3) $C_{\text{ND}}$: $C_{\text{ND}}$ refers to the complexity of finding $\Pi_{1,\text{ND}}$, the set of non-dominated policies from $\Pi_1$. Let $N_{\text{ND}} = |\Pi_{1,\text{ND}}| = $. The non-dominated policies searching complexity is $C_{\text{ND}} = \mathcal{O}(SAN_{\text{ND}}D)$.
>
> (4) $C_{\text{cvx}}$ is the complexity of computing the convex hull faces formed by $N_{\text{ND}}+1$ points that intersect at the current policy. If the number of vertices of the output convex hull is $N_{\text{cvx}}$. By [1], the complexity of the calculation of convex hull is $\mathcal{O}(N_{\text{ND}}\log N_{\text{cvx}})$ when $D\leq 3$ and $\mathcal{O}(N_{\text{ND}}f_{N_{\text{cvx}}}/N_{\text{cvx}})$ otherwise, where $f_L=\mathcal{O}(L^{\lfloor D/2\rfloor}/(\lfloor D/2\rfloor!))$.
>
>
> For each iteration, OLS solves the single-objective MDP with a preference vector, removes obsolete preference vectors with the new points, and computes new preference vectors. For each iteration, solving the single-objective planning problem incurs a complexity of $\mathcal{O}(C_{\text{so}}+C_{\text{pe}})$. The complexity of comparing with the previous candidate Pareto optimal policies and removing obsolete ones is $\mathcal{O}(ND)$, and the complexity of calculating a new preference vector is denoted as $C_{\text{we}}$.
> The computational complexity of OLS is $\mathcal{O}(N^2D+NC_{\text{we}}+N\left(\left(S^4A+S^3A^2\right)\right)+NS^3D)$.
> Note that calculating new weight requires calculating the vertices of the convex hull constructed by $N+D+2$ hyperplanes of dimension $D+1$, making it significantly more expensive than the local convex hull construction of neighboring policies in our proposed algorithm. Hence, $C_{we}$ is much larger than $C_{\text{cvx}}$.
>
> The response continues in [2/2].

---

> ### Author Response · Authors · 2024-11-20
> **Response to Reviewer 4Q5J [2/2]**
>
> In conclusion, our algorithm induces total computational complexity of $\mathcal{O}(S^4A+S^3A^2+NS^4AD+NSAN_{\text{ND}}D+C_{\text{cvx}})$, while OLS has computation complexity $\mathcal{O}(N(S^4A+S^3A^2)+NS^3D+N^2D+NC_{\text{we}})$. Given $C_{\text{cvx}}$ is much smaller than $C_{\text{we}}$, our algorithm is more efficient than OLS in all cases, especially when $N$ is larger than $SA$.
>
> In the revised paper, we added the following sentences in Section 5.2 to clarify the computational complexity difference between our algorithm and OLS:” *For each iteration, OLS solves the single-objective MDP for a given preference vector and calculates new preference vectors by constructing a convex hull of all candidate preference vectors. In contrast, our algorithm requires only a single single-objective planning step during the initialization phase to obtain the initial deterministic Pareto-optimal policy. Rather than constructing a global convex hull encompassing all potential Pareto-optimal policies, our algorithm builds the convex hull locally, based on the vertex and its neighboring policies. As a result, our algorithm achieves significantly lower computational complexity compared to OLS. A detailed complexity comparison between the two algorithms is provided in Section H.*”
>
>
> > Line 905 says that $\Pi$ can contain non-stationary policies, while Line 684 defines $\Pi$ as a set of stationary policies. I am confused about whether the authors only consider stationary policies or not. Can you explain any intentional differences if they exist?
>
> Thanks for the thoughtful question!
>
> The set of long-term returns of stationary policies is complete, meaning that the long-term return of any non-stationary policy can always be achieved by a stationary policy (Theorem 3.1 of [1]). Therefore, our analysis is based on stationary policies since they are sufficient to represent the entire Pareto front. We clarify this distinction at the start of Section D of the revised manuscript (Line 905 of the previous version) to avoid confusion.
>
> [1] Altman, Eitan. Constrained Markov decision processes. Routledge, 2021.
>
> > What is $L\Pi_D(\mu)$ in Line 749? Is it $\Pi_D(\mu)$? Or does it mean that the vertices are the occupancy measures of a set of policies $\Pi_D(\mu)$? (If it is a separate concept, the authors should define what is.)
>
> Thanks for your question. This was a typo. $L{\Pi_D}(\mu)$ should be $L^{\Pi_D}(\mu)$, which represents the set of the occupancy measures for the set of deterministic policies $\Pi_D$ under the initial state distribution $\mu$. This typo has been corrected in the revised manuscript.
>
> > To recover “full” exact Pareto front, I believe that the “if” statement in Lemma 3 should be modified to “if and only if” (necessary and sufficient). If you intended to do so, can you modify that? (It seems that Line 409 talks about “only if” part, but it is not accurately stated in the lemma.) If not, please explain why the sufficient conditions are enough to prove the following theorems or propositions. Also, I cannot locate the proof of Lemma 3.
>
> Thank you for the suggestion! You are correct that the “if” in Lemma 3 should be “if and only if” to ensure that the extracted faces are Pareto optimal and all discarded faces are not Pareto optimal. We have updated the lemma accordingly. Lemma 3 is now restated as Lemma 22 in Appendix G. Additionally, we have included a sentence after Lemma 3 indicating the location of its proof.
>
> > I think the point of Lines 10-11 in Algorithm 2 is to include all normal vectors associated with lower-dimensional faces to apply (1). If that’s the case, shouldn’t Line 4 of Algorithm 2 specify that .pop() be done from high-dimensional faces to lower-dimensional faces (or specify that .push() is done in the order of dimension)? If the order matters, please modify the algorithm accordingly and if it doesn't, please explain why this is the case.
>
> Thank you for your suggestion. You are correct that the algorithm requires clarification. Specifically, after popping the high-dimensional faces from the queue, we must push the corresponding lower-dimensional faces into the queue. We have updated Algorithm 2 accordingly in the revised paper to reflect this correction.
>
> > It would be better to add relevant existing algorithms in the appendix (e.g. Roijers, PPrune algorithm) for completeness.
>
> Thanks for your suggestion. We added the PPrune algorithm in Appendix I.
>
> > It would be better to note the corresponding proof locations below all theorems (e.g. Theorem 2, Proposition 1).
>
> Thanks for your suggestions, we have added the corresponding proof locations after each theorem in the revised manuscript.
>
> > In the abstract, it would be better to use MO-MDPs inside the first parenthesis instead of MDPs to understand the 12th line of the abstract (I personally didn't get it at first glance).
>
> Thanks for your suggestion. We have changed the abbreviation in the abstract.

---

> > ### Comment · Reviewer_4Q5J · 2024-11-20
> >
> > Thanks for the detailed explanation, especially on the complexity analysis. After carefully reading through the rebuttal and the updated manuscript, I have raised my score to 8.

---

> > > ### Author Response · Authors · 2024-11-21
> > >
> > > Thank you very much. Please feel free to let us know if you have any further questions.

---

### Official Review · Reviewer_hhfj · 2024-11-04

**Soundness:** 3
**Presentation:** 3
**Contribution:** 3
**Rating:** 6
**Confidence:** 3

**Summary:**

The paper proposes an exact method for finding Pareto fronts (in turn Pareto optimal policies) for Multi-Objective Markov Decision Processes (MO-MDPs). Leveraging the geometric properties of the Pareto front, the authors demonstrate that it lies on the boundaries of a convex polytope, with neighboring Pareto front vertices differing by at most one state-action pair. The paper provides proofs showing that every deterministic policy on the Pareto front is connected to at least one deterministic Pareto optimal policy, which can be extracted via the convex hull of a Pareto optimal policy and its neighbors. Building on these findings, the paper introduces a search algorithm for Pareto fronts that requires solving each single objective of the MO-MDP once for initialization, with total iterations bounded by the number of Pareto fronts. The proposed algorithm is evaluated on small MDPs with 5 to 8 states, 5 to 7 actions, and 3 objectives. The benchmark used for comparison is the Optimistic Linear Support (OLS) method, which can find Pareto vertices but not the entire Pareto front. Results indicate that the proposed method can find exact Pareto fronts and is more computationally efficient than the OLS method.

**Strengths:**

* The paper makes a novel theoretical contribution by providing an exact solution method for finding optimal Pareto front in MO-MDPs. The proofs are rigorous, offering valuable insights into the geometric properties of the Pareto front, which could impact other optimization methods for MO-MDPs.
* The method reduces computational complexity compared to OLS, marking a notable advantage in computational efficiency for exact MO-MDP solutions. The iterative nature of the algorithm also suggests it could be more robust and reliable than blind search-based methods.
* The writing is clear and well-structured with good visual presentations.

**Weaknesses:**

* The practical relevance of finding exact solutions versus approximation is not thoroughly addressed. Since approximate solutions are computationally easier to obtain, the paper could benefit from providing scenarios and discussion where exact solutions are preferred.  Comparisons and discussions against approximation methods, such as those in [1,2], are missing, which could help contextualize the result.
* The experimental evaluation is relatively weak, relying on very small, toy MO-MDPs with only a single benchmark for comparison.  Is there any practical relevance or application of exact solutions to such problems on such small problem sizes?
* The scalability of the algorithm is not adequately discussed. From the results, it seems like running times are negligible for small MDPs, it's unclear how the method performs on larger problems, and when scalability could become an issue.

[1] Xingchao Liu, Xin Tong, and Qiang Liu. Profiling Pareto front with multi-objective Stein variational gradient descent. Advances in Neural Information Processing Systems, 34:14721–14733, 2021.

[2] Xiaoyuan Zhang, Xi Lin, Bo Xue, Yifan Chen, and Qingfu Zhang. Hypervolume maximization: A geometric view of Pareto set learning. Advances in Neural Information Processing Systems, 36:38902–38929, 2023.

**Questions:**

Please address the concerns and questions raised in the weaknesses section.

---

> ### Author Response · Authors · 2024-11-20
> **Response to Reviewer hhfj [1/2]**
>
> > The practical relevance of finding exact solutions versus approximation is not thoroughly addressed. Since approximate solutions are computationally easier to obtain, the paper could benefit from providing scenarios and discussions where exact solutions are preferred.
>
> Thank you for your suggestion.
>
> Finding the exact Pareto front is crucial in scenarios where low-dimensional faces of the Pareto front are difficult to approximate through preference sampling. For example, as shown in Fig. 4, even in a simple MDP with three objectives, there can be a line segment on the Pareto front that is not incident to any higher-dimensional Pareto-optimal face. Sampling this 1D line segment within the 2D space has a probability of zero, making it nearly impossible for approximation methods to identify such edge cases and achieve the exact Pareto front.
>
> To clarify this point, we have added the following sentences to the introduction to highlight when the exact Pareto front is preferred over an approximate one:” In cases where the Pareto front contains low-dimensional faces, sampling preference vectors on those faces makes it nearly impossible for approximation methods to identify such edge cases and accurately reconstruct the exact Pareto front.”
>
>
> > Comparisons and discussions against approximation methods, such as those in [1,2], are missing, which could help contextualize the result.
>
> Thanks for your feedback.
>
> While [1] and [2] propose efficient algorithms for approximating the Pareto front in multi-objective optimization problems, these methods are not designed for MDPs and do not leverage the structural properties of MDPs. In our setting, the long-term returns of all policies form a convex polytope, and the Pareto front corresponds to its boundary. These MDP-specific geometric properties significantly reduce the computational complexity of finding Pareto-optimal solutions. Moreover, our algorithm exploits additional geometric features of the Pareto front in the MO-MDP setting.
>
> We added the following sentences to the Related Work section of the revised manuscript to discuss the relationship with approximation methods in multi-objective learning:” Some works in multi-objective optimization [1, 2] can be extended to solve MO-MDPs. However, these algorithms do not take advantage of the specific structure of MO-MDPs, where multiple objectives share the same state and action spaces, as well as identical transition dynamics. Exploiting this structure in MO-MDP settings could potentially lead to significant reductions in computational complexity.”
>
>
> > The experimental evaluation is relatively weak, relying on very small, toy MO-MDPs with only a single benchmark for comparison. Is there any practical relevance or application of exact solutions to such problems on such small problem sizes?
>
> Thank you, while the key contribution of our paper is indeed theoretical, we now add another experiment as follows to address your concern. Inspired by the deep-sea-treasure setting from [1], which is a related paper in MORL, construct a scenario where the agent navigates a grid world, making trade-offs between different objectives. At each step, the agent can move up, down, left, or right, and the reward for each grid is generated randomly. We compare the performance of the OLS algorithm and the proposed Pareto front searching algorithm under different grid sizes (varying numbers of rows and columns). The experiments were conducted on smaller grid sizes to ensure that the running time of OLS remained manageable. The figure shows that the proposed algorithm consistently outperforms OLS in terms of efficiency across all tested settings. We attached the related experiments in Appendix J.
>
>
> [1] Yang, Runzhe, Xingyuan Sun, and Karthik Narasimhan. "A generalized algorithm for multi-objective reinforcement learning and policy adaptation." Advances in neural information processing systems 32 (2019).

---

> ### Author Response · Authors · 2024-11-20
> **Response to Reviewer hhfj [2/2]**
>
> > The scalability of the algorithm is not adequately discussed. From the results, it seems like running times are negligible for small MDPs, it's unclear how the method performs on larger problems, and when scalability could become an issue.
>
> Thanks for your suggestion. We analyzed the complexity of the proposed Pareto searching algorithm in Appendix H of the revised manuscript. Let $S$, $A$, and $D$ represent the state space size, the action space size, and the reward dimension, respectively. We have shown that the overall computational complexity is $\mathcal{O}(C_{\text{so}}+N(SAC_{\text{pe}}+C_{\text{ND}}+C_{\text{cvx}}))$, where the terms are defined as follows:
>
> (1) $C_{\text{so}}$: The computational complexity of single-objective planning is $C_{\text{so}} = \mathcal{O}\left(S^4A+S^3A^2\right)$;
>
> (2) $C_{\text{pe}}$: The computational complexity of the policy evaluation is $C_{\text{pe}} = \mathcal{O}(S^3D)$;
>
> (3) $C_{\text{ND}}$: This  refers to the complexity of finding $\Pi_{1,\text{ND}}$, the set of non-dominated policies from $\Pi_1$. If $|\Pi_{1,\text{ND}}| = N_{\text{ND}}$, the computational complexity of finding the non-dominated policies among all neighboring policies is $\mathcal{O}(SAN_{\text{ND}}D)$.
>
> (4) $C_{\text{cvx}}$ is the complexity of computing the convex hull faces formed by $N_{\text{ND}}+1$ points that intersect at the current policy. If the number of vertices of the output convex hull is $N_{\text{cvx}}$. By [1], the complexity of the calculation of convex hull is $\mathcal{O}(N_{\text{ND}}\log N_{\text{cvx}})$ when $D\leq 3$ and $\mathcal{O}(N_{\text{ND}}f_{N_{\text{cvx}}}/N_{\text{cvx}})$ when $D>3$, where $f_L=\mathcal{O}(L^{\lfloor D/2\rfloor}/(\lfloor D/2\rfloor!))$.
>
> In conclusion, our algorithm induces total computational complexity of $\mathcal{O}(S^4A+S^3A^2+NS^4AD+NSAN_{\text{ND}}D+C_{\text{cvx}})$. Since the number of policies in $\Pi_{1,\text{ND}}$ is limited, $N_{\text{ND}}$ and $N_{\text{cvx}}$ are limited, therefore the complexities $C_{\text{ND}}$ and $C_{\text{cvx}}$ are similarly constrained.
>
> We also want to emphasize that obtaining the exact Pareto front requires traversing all deterministic policies, making the proportionality to $N$ in the complexity unavoidable. Additionally, since planning in a single-objective MDP has a complexity of $\mathcal{O}(S^4A + S^3A^2)$, the $\mathcal{O}(S^4A)$ term in our algorithm’s complexity is reasonable and acceptable.
>
> It may be possible to address scalability issues by borrowing ideas from previous RL research that apply abstractions over MDPs. We leave the exploration of scalability improvements to future work.

---

> > ### Comment · Reviewer_hhfj · 2024-11-26
> >
> > We thank the author for the detailed explanation. The paper makes good theoretical contribution, but the experiments/ablation still do not demonstrate empirically the scalability of the algorithm. Therefore I maintain my final score.

---

> ### Author Response · Authors · 2024-11-26
>
> We thank the reviewer for the appreciation on our theoretical contribution. Regarding the experiments on scalability, in Appendix J of the revised version, we add additional simulation figures with larger size ($S^3A$ can be as large as $2.5\times 10^6$) and figures in a realistic setting deep-sea-treasure setting inspired from [1].
>
> [1] Yang, Runzhe, Xingyuan Sun, and Karthik Narasimhan. "A generalized algorithm for multi-objective reinforcement learning and policy adaptation." Advances in neural information processing systems 32 (2019).

---

### Official Review · Reviewer_WFhh · 2024-11-06

**Soundness:** 3
**Presentation:** 2
**Contribution:** 3
**Rating:** 8
**Confidence:** 3

**Summary:**

The paper studies how to find the Pareto front of multi-objective Markov decision processes efficiently.  The authors first show that the Pareto front lies on the boundary of a convex polytope, with vertices representing deterministic policies, and neighboring vertices of the Pareto front differ by only a state-action pair in the deterministic policy. Based on this observation, the authors propose a Pareto front searching algorithm with three key steps: (1) neighbor search; (2) incident faces identification; (3) Pareto face extraction. A major advantage of this method is that the localized search among deterministic policies that differ only one state-action pair significantly reduces the computational burden. The authors also provide some empirical results to show the effectiveness of this proposed method.

**Strengths:**

- The key idea is clearly presented. Compared with the state of the art for multi-objective Markov decision processes, the authors highlight that the proposed method can find the entire Pareto front.

- The authors derive a few interesting geometric properties of the vector value function: (1) distance one property; (2) sufficiency of traversing over edges; (3) locality property of the Pareto front. I believe this is the main theoretical contribution to the literature. The authors also show how these properties are used to find the entire Pareto front, which is new to me.

- Based on the geometric properties of the Pareto front, the authors develop an efficient Pareto search algorithm. This algorithm initiates with a neighbor search of a deterministic policy, computes incident faces for a convex hull, and then extracts the Pareto front.

- The authors evaluate the performance of the proposed method in an experiment, showing better runtime compared to benchmark.

**Weaknesses:**

- A key contribution is to find the full exact Pareto front for multi-objective MDPs. Although it is of theoretical interest, it is less discussed the practical use of the entire Pareto front, such as multi-objective learning. It would be helpful if the authors could provide some examples that warrant the entire Pareto front.

- The authors first introduce the algorithm and then the key properties. Without reading Section 4, it is difficult to understand why Algorithm 1 works. It would be helpful to reverse the order of Section 3 and Section 4.

- The convergence study of Algorithm 1 is not provided. It seems that it is not justified that Algorithm 1 can find the entire Pareto front. Also, the stop criterion and computational complexity are not analyzed.

- The scalability of Algorithm 1 is not analyzed regarding the size of MDP and dimension of reward.

- Regarding the properties of the vector value function, it would be helpful if the authors could provide some examples or intuitions behind them.

- The current analysis seems to be limited to finite MDPs. It would be useful to extending it to MDPs with continuous spaces.

- The example in experiments is artificial. It would be useful if the authors could provide some practical use cases.

**Questions:**

- The authors have focused on identifying deterministic policies in the Pareto front. What are specific reasons not considering the whole policy space including stochastic policies?

- What are assumptions about MDPs made in analysis (e.g. finite state/action spaces, ergodicity, etc.)? Can the authors state them in theorems?

- How sensitive of Algorithm 1 to the accuracy (e.g., errors or approximations in these subroutines) of calling subroutines like line 11, 12, and 13?

- Can the authors provide some use examples of the entire Pareto front?

---

> ### Author Response · Authors · 2024-11-20
> **Response to Reivewer WFhh [1/4]**
>
> We have uploaded the revised paper, with the major changes highlighted in blue.
>
> > A key contribution is to find the full exact Pareto front for multi-objective MDPs. Although it is of theoretical interest, it is less discussed the practical use of the entire Pareto front, such as multi-objective learning. It would be helpful if the authors could provide some examples that warrant the entire Pareto front. Can the authors provide some use examples of the entire Pareto front?
>
> We provide three examples where having the full Pareto front is more advantageous than finding a single Pareto optimal solution for a given preference [1,2,4,5].
>
> **Understanding trade-offs in dynamic decision settings**: Users may start with a specific Pareto optimal policy, but if they are unsatisfied with the outcome, they might need to adjust their policies to explore other trade-offs. The full Pareto front provides crucial information on how the trade-offs change across different Pareto optimal policies, allowing users to make informed adjustments. In contrast, solving for a single Pareto optimal policy does not provide this broader context, forcing users to blindly search for nearby policies.
>
> **Different objective scales**: In scenarios where the objectives have different scales [3, 4],  assigning a preference vector may fail to accurately capture the true preferences. For example, when one objective dominates due to its larger scale, even assigning a small weight to it might still result in it influencing the weighted sum significantly. As a result, finding a desirable Pareto optimal solution through a preference vector abstracted from the user’s preference can require multiple iterations of trials of preferences. However, with the full Pareto front at hand, users can explore the front directly to identify solutions that best reflect their true preferences without the need for repeated adjustments.
>
> **Frequently changing preferences**: In scenarios where the users’ preferences change frequently [5], recomputing a new Pareto optimal policy for each updated preference via online learning can be inefficient and computationally expensive. With access to the full Pareto front, users can instantly retrieve the Pareto optimal policy that aligns with their updated preferences, eliminating the need to solve the MDP from scratch for every iteration.
>
>
> [1] Liu, Xingchao, Xin Tong, and Qiang Liu. "Profiling Pareto front with multi-objective stein variational gradient descent." Advances in Neural Information Processing Systems 34 (2021): 14721-14733.
>
> [2] Zhang, Xiaoyuan, et al. "Hypervolume maximization: A geometric view of Pareto set learning." Advances in Neural Information Processing Systems 36 (2023): 38902-38929.
>
> [3] Kim, Dohyeong, et al. "Scale-Invariant Gradient Aggregation for Constrained Multi-Objective Reinforcement Learning." arXiv preprint arXiv:2403.00282 (2024).
>
> [4] Abdolmaleki, Abbas, et al. "A distributional view on multi-objective policy optimization." International conference on machine learning. PMLR, 2020.
>
> [5] Yang, Runzhe, Xingyuan Sun, and Karthik Narasimhan. "A generalized algorithm for multi-objective reinforcement learning and policy adaptation." Advances in neural information processing systems 32 (2019).
>
>
> > The authors first introduce the algorithm and then the key properties. Without reading Section 4, it is difficult to understand why Algorithm 1 works. It would be helpful to reverse the order of Section 3 and Section 4.
>
> Thank you for your thoughtful suggestion.
>
> To address your concern regarding the clarity of why Algorithm 1 works, we have included the following high-level explanation of the theoretical results in Section 3.2: “*The intuition behind the proposed algorithm is that any deterministic Pareto optimal policy can identify all neighboring policies on the Pareto front by considering deterministic policies that differ by only one state-action pair. This ensures that all Pareto optimal policies are discovered during the search process. Additionally, the Pareto optimality justification excludes any policies that are not Pareto optimal, guaranteeing that the search trajectory remains on the Pareto front. The detailed theoretical results will be shown in Section 4.*”
>
> Additionally, we have added the following sentence at the start of Section 3 of the revised paper: “*A proof sketch of the proposed Pareto front searching algorithm will be provided in Section 4.*” This addition aims to provide readers with a clearer understanding of the algorithm’s rationale. We hope this change improves the flow and overall readability of the paper.
>
> However, we would prefer to not exchange Sections 3 and 4 because presenting the Pareto optimal searching algorithm in Section 3 allows us to first provide the readers with a high-level overview of how the algorithm operates,  before diving into the detailed theoretical justification and proof sketches in Section 4.

---

> ### Author Response · Authors · 2024-11-20
> **Response to Reivewer WFhh [2/4]**
>
> > The convergence study of Algorithm 1 is not provided. It seems that it is not justified that Algorithm 1 can find the entire Pareto front.
>
> Please see Section 4, where we address the convergence of Algorithm 1 and its ability to find the entire Pareto front.
>
> Theorem 1 establishes that neighboring deterministic policies on the Pareto front differ by only one state-action pair. Lemma 2 further shows that any deterministic Pareto optimal policy has at least one neighbor on the Pareto front. Together, these results ensure that we can traverse all deterministic Pareto optimal policies by investigating policies that differ by just one state-action pair. This justifies Step 1, corresponding to line 10 in Algorithm 1.
>
> Proposition 1 guarantees that the faces of the Pareto front intersecting at a vertex can be extracted from the convex hull constructed by the vertex and the return of all non-dominated deterministic policies differ by only one state-action pair. This supports lines 11-13 of Algorithm 1.
>
> These results collectively justify the correctness and convergence of Algorithm 1, ensuring that it can find the exact Pareto front.
>
> > Also, the stop criterion and computational complexity are not analyzed. The scalability of Algorithm 1 is not analyzed regarding the size of MDP and the dimension of reward.
> Thank you for the question. The algorithm terminates once all Pareto-optimal deterministic policies have been explored. Algorithm 1 continuously adds neighboring unvisited deterministic policies to the queue and verifies whether the policies in the queue are Pareto optimal. The algorithm stops when the queue is empty, indicating that all neighbors of deterministic Pareto-optimal policies have been visited.
>
>
> We have provided the complexity analysis in terms of the size of the MDP and the reward dimension in Appendix H of the revised paper. Let $S$, $A$, and $D$ represent the state space size, the action space size, and the reward dimension, respectively. The overall computational complexity is $\mathcal{O}(C_{\text{so}}+N(SAC_{\text{pe}}+C_{\text{ND}}+C_{\text{cvx}}))$, where the terms are defined as follows:
>
> (1) $C_{\text{so}}$: The computational complexity of single-objective planning is $C_{\text{so}} = \mathcal{O}\left(S^4A+S^3A^2\right)$;
>
> (2) $C_{\text{pe}}$: The computational complexity of the policy evaluation is $C_{\text{pe}} = \mathcal{O}(S^3D)$;
>
> (3) $C_{\text{ND}}$: This refers to the complexity of finding $\Pi_{1,\text{ND}}$, the set of non-dominated policies from $\Pi_1$. If $|\Pi_{1,\text{ND}}| = N_{\text{ND}}$, the computational complexity of finding the non-dominated policies among all neighboring policies is $\mathcal{O}(SAN_{\text{ND}}D)$.
>
> (4) $C_{\text{cvx}}$ is the complexity of computing the convex hull faces formed by $N_{\text{ND}}+1$ points that intersect at the current policy. If the number of vertices of the output convex hull is $N_{\text{cvx}}$. By [1], the complexity of the calculation of convex hull in the balanced setting is $\mathcal{O}(N_{\text{ND}}\log N_{\text{cvx}})$ when $D\leq 3$ and $\mathcal{O}(N_{\text{ND}}f_{N_{\text{cvx}}}/N_{\text{cvx}})$ when $D>3$, where $f_L=\mathcal{O}(L^{\lfloor D/2\rfloor}/(\lfloor D/2\rfloor!))$.
>
> Since the number of policies in $\Pi_{1,\text{ND}}$ is limited, $N_{\text{ND}}$ and $N_{\text{cvx}}$ are limited, therefore the complexities $C_{\text{ND}}$ and $C_{\text{cvx}}$ are similarly constrained. In conclusion, our algorithm induces total computational complexity of $\mathcal{O}(S^4A+S^3A^2+NS^4AD+NSAN_{\text{ND}}D+C_{\text{cvx}})$.
>
> [1] Barber, C. Bradford, David P. Dobkin, and Hannu Huhdanpaa. "The quickhull algorithm for convex hulls." ACM Transactions on Mathematical Software (TOMS) 22.4 (1996): 469-483.
>
> > Regarding the properties of the vector value function, it would be helpful if the authors could provide some examples or intuitions behind them.
>
> Thank you for your suggestion.
> Theorem 1 demonstrates that all neighboring deterministic policies on the Pareto front differ by only one state-action pair. Lemma 2 further establishes that any vertex lies on the edge of the Pareto front. For example, consider a toy scenario with 2 states, 2 actions, and 2 objectives, where the number of Pareto-optimal policies exceeds 1. We represent a policy as $(i, j)$, where $i$ denotes the action chosen in state 1 and $j$ denotes the action chosen in state 2, with $i, j \in $ {$1,2$}. If $(1, 1)$ is a Pareto-optimal policy, then for almost all MDPs (with probability 1), there exists another Pareto-optimal policy in the set {${(1, 2), (2, 1)\}$}. These results imply that we can traverse all Pareto-optimal deterministic policies by exploring policies that differ by only a single state-action pair.
> We have also added corresponding explanations in Section 4 to enhance understanding.

---

> ### Author Response · Authors · 2024-11-20
> **Response to Reivewer WFhh [3/4]**
>
> > The current analysis seems to be limited to finite MDPs. It would be useful to extend it to MDPs with continuous spaces.
>
> Please note that this is the first work that characterizes the exact Pareto front for even finite MDPs. Extending the analysis to MDPs with continuous spaces presents additional challenges. Specifically, properties of the Pareto front demonstrated in finite state and action spaces, such as the Pareto front lying on the boundary of a convex polytope, are not guaranteed in continuous spaces. Additionally, the key geometric characteristic that neighboring policies differ by only one state-action pair no longer applies in continuous MDPs.
>
> One potential approach is to discretize the continuous action space into a discrete action space using various discretization methods [1, 2, 3]. Once the continuous action space is appropriately discretized, our current Pareto front searching algorithm can be applied to the resulting approximated finite MDP to find the approximate Pareto front of the original continuous MDP. The general question is beyond the scope of the current paper, but is an interesting question for future work.
>
> [1] Metz, Luke, et al. "Discrete sequential prediction of continuous actions for deep RL." arXiv preprint arXiv:1705.05035 (2017).
> [2] Andrychowicz, OpenAI: Marcin, et al. "Learning dexterous in-hand manipulation." The International Journal of Robotics Research 39.1 (2020): 3-20.
> [3] Tang, Yunhao, and Shipra Agrawal. "Discretizing continuous action space for on-policy optimization." Proceedings of the AAAI conference on artificial intelligence. Vol. 34. No. 04. 2020.
>
> > The example in experiments is artificial. It would be useful if the authors could provide some practical use cases.
>
> We add another experiment as follows to address your concern.
> We adopt the deep-sea-treasure setting from [1], which is a related paper in MORL, and construct a scenario where the agent navigates a grid world, making trade-offs between different objectives. At each step, the agent can move up, down, left, or right, and the reward for each grid is generated randomly. We compare the performance of the OLS algorithm and the proposed Pareto front searching algorithm under different grid sizes (varying numbers of rows and columns). The experiments were conducted on smaller grid sizes to ensure that the running time of OLS remained manageable. The result shows that the proposed algorithm consistently outperforms OLS in terms of efficiency across all tested settings. We append the related experiments in Appendix J of the revised paper.
>
> > The authors have focused on identifying deterministic policies in the Pareto front. What are specific reasons not considering the whole policy space including stochastic policies?
>
> It is not true that our results are focused only on identifying deterministic policies on the Pareto front.  Please note that our algorithm characterizes the entire Pareto front, including both deterministic and stochastic policies. In our approach, while we focus on traversing the edges of the Pareto front to find neighboring Pareto optimal deterministic policies, we also examine the incident faces of these deterministic Pareto optimal policies, which includes the set of stochastic Pareto optimal policies. Thus, our method captures the full Pareto front, including both deterministic and stochastic policies. We have added the following sentence in the abstract to explicitly indicate that our algorithm characterizes both deterministic and stochastic policies: “In this work, we address the challenge of efficiently discovering the Pareto front, involving both deterministic and stochastic Pareto optimal policies.”
>
> > What are assumptions about MDPs made in analysis (e.g. finite state/action spaces, ergodicity, etc.)? Can the authors state them in theorems?
>
> The two assumptions we made in the paper can be found in Section 3.1. However, after revising the manuscript based on suggestions from Reviewer JJrB, we realized that the ergodicity assumption is not necessary.
> The only remaining assumption is that the MDP has finite state and action spaces and the initial state distribution $\mu$ has the full state space coverage. We now explicitly state this assumption in the relevant theorems to clarify the scope of our analysis.

---

> ### Author Response · Authors · 2024-11-20
> **Response to Reivewer WFhh [4/4]**
>
> > How sensitive of Algorithm 1 to the accuracy (e.g., errors or approximations in these subroutines) of calling subroutines like lines 11, 12, and 13?
>
> The sensitivity of Algorithm 1 to errors can be analyzed by examining its three key steps corresponding to Steps 10, 12, and 13 of Algorithm 1.
>
> **Step 10**: This step selects non-dominated policies from all neighboring policies. As long as the policy evaluation errors do not affect the dominance relationships among the policies, the correct set of non-dominated policies will be preserved.
>
> **Step 12**: This step constructs the convex hull from the set of non-dominated policies. The convex hull remains valid as long as the relative geometric relationships of the points are preserved. Specifically, this requires that the points belonging to the ground-truth convex hull are correctly identified and retained.
>
> **Step 13**: This step identifies the Pareto optimal faces from the boundary of the convex hull. If the directionality of the faces is preserved despite minor inaccuracies, the algorithm will still correctly detect the Pareto optimal faces.
>
> In summary, Algorithm 1 is robust to minor errors in subroutines, provided these errors do not disrupt key geometric and dominance relationships critical to each step.
>
>
> Thank you for your detailed review. Please let us know if you have further questions that need clarification.

---

> ### Comment · Reviewer_WFhh · 2024-11-26
>
> Thank you for the response. Since the response and the revised version address my concerns, I am inclined to recommend acceptance.

---

### Official Review · Reviewer_QTi5 · 2024-11-10

**Soundness:** 3
**Presentation:** 3
**Contribution:** 3
**Rating:** 8
**Confidence:** 3

**Summary:**

The paper investigates the geometric structure of the Pareto front in multi-objective MDPs and reveals key properties, including that the vertices on the Pareto front correspond to deterministic policies and that neighboring vertices differ by only one state-action pair, almost surely.
The paper also presents an efficient algorithm that leverages these geometric properties to find the Pareto front, which could reduces the computational complexity compared to existing methods.
The empirical studies demonstrate the proposed algorithm maintains a much better run time than Optimistic Linear Support (OLS).

**Strengths:**

The paper introduces an original contribution to the field by uncovering key geometric properties of the Pareto front in multi-objective Markov Decision Processes.
The paper has good quality, and is supported by theoretical proofs and empirical evidence.
The paper provides a clear explanation of the geometric properties of the Pareto front and a clear demonstration of the proposed algorithm. The paper should ideally explain the empirical setup in more detail.
The paper makes a significant contribution to solving multi-objective MDP problems by offering insights into the geometry of the Pareto front and developing an efficient algorithm for finding the exact Pareto front.

**Weaknesses:**

In the evaluation of the algorithm, more details could be provided about the experiment setting to give context to the results. For instance, specifying the states, actions, transition kernel, reward function, and the number of time steps considered. Also, including a few more benchmark algorithms would help demonstrate the efficiency of the proposed algorithm more effectively.

**Questions:**

The evaluation of the algorithm demonstrates its efficiency with state space sizes of 5 and 8, action spaces ranging from 5 to 7, and a reward dimension of 3. If the state and action space cardinalities were significantly larger, would this algorithm remain more efficient compared to others? Specifically, could sampling the preference space be more efficient in certain cases?

Can the algorithm be adapted for use in continuous action spaces?

---

> ### Author Response · Authors · 2024-11-20
> **Response to Reviewer QTi5 [1/2]**
>
> We have uploaded the revised paper, with the major changes highlighted in blue.
>
> > In the evaluation of the algorithm, more details could be provided about the experiment setting to give context to the results. For instance, specifying the states, actions, transition kernel, reward function, and the number of time steps considered.
>
> Thanks for your suggestion. We have added the experiment settings in Section 5.
>
> > Also, including a few more benchmark algorithms would help demonstrate the efficiency of the proposed algorithm more effectively.
>
> Thank you for your suggestion. There are very few algorithms for finding the exact whole Pareto front or even the entire set of deterministic Pareto optimal policies.
>
> To the best of our knowledge, our algorithm is the first to find the exact Pareto front in multi-objective MDPs. Consequently, we constructed a benchmark algorithm that evaluates all deterministic policies, selects the non-dominated policies among them, and identifies the Pareto optimal faces from the convex hull constructed by these non-dominated policies by analyzing their normal vectors. We compare our algorithm with this benchmark to demonstrate its effectiveness in finding the exact Pareto front.
>
> Additionally, regarding the byproduct of our algorithm in identifying deterministic Pareto optimal policies, the OLS algorithm is the only other method we are aware of that can obtain the complete set of deterministic Pareto optimal policies. Most other approaches approximate the Pareto front by sampling over the preference space. Therefore, we provide a comparison with the OLS algorithm to evaluate the performance in finding deterministic Pareto optimal policies.
>
>
> > The evaluation of the algorithm demonstrates its efficiency with state space sizes of 5 and 8, action spaces ranging from 5 to 7, and a reward dimension of 3. If the state and action space cardinalities were significantly larger, would this algorithm remain more efficient compared to others? Specifically, could sampling the preference space be more efficient in certain cases?
>
> Thank you for your insightful questions.
>
> Note that our algorithm is the first to find the exact Pareto front in MO-MDPs. Finding the exact Pareto front is essential for capturing low-dimensional faces that are difficult to approximate through preference sampling. Our algorithm achieves the exact Pareto front by traversing the state-action space along the edges without needing to explore the infinite preference space. Hence, if the state and action space cardinalities were significantly larger, this algorithm would continue to remain efficient.
>
> While there may be cases where sampling the preference space could be more efficient, finding the exact Pareto front is essential for capturing low-dimensional faces that are difficult to approximate through preference sampling. For instance, as shown in Fig. 4, even in a simple MDP with three objectives, four states, and three actions, there exists a line segment on the Pareto front that is not incident to any Pareto-optimal face. Sampling this 1D line segment in a 2D space has a probability of zero, making it nearly impossible for preference-based sampling methods to detect such edge cases and fully reconstruct the exact Pareto front. However, we acknowledge that sampling the preference space can provide a rough approximation of the Pareto front, which can be useful for understanding the general trade-offs between objectives. Exploring how to efficiently combine an initial rough approximation with our exact Pareto front search is an interesting direction for future work.

---

> > ### Comment · Reviewer_QTi5 · 2024-11-27
> >
> > Thank you for providing additional details on the experiment settings and including Appendix J to address scalability.The discussion on the trade-offs between the accuracy of the Pareto front and efficiency is a helpful to add, especially in highlighting the the 1D edge case in the provided example. I maintain my score of 8.

---

> ### Author Response · Authors · 2024-11-20
> **Response to Reviewer QTi5 [2/2]**
>
> > Can the algorithm be adapted for use in continuous action spaces?
>
> Extending the analysis to MDPs with continuous spaces presents additional challenges. Specifically, properties of the Pareto front demonstrated in finite state and action spaces, such as the Pareto front lying on the boundary of a convex polytope, are not guaranteed in continuous spaces. Additionally, the key geometric characteristic that neighboring policies differ by only one state-action pair no longer applies in continuous MDPs.
>
> One potential approach is to discretize the continuous action space into a discrete action space using various discretization methods [1, 2, 3]. Once the continuous action space is appropriately discretized, our current Pareto front searching algorithm can be applied to the resulting approximated finite MDP to find the approximate Pareto front of the original continuous MDP. However, note that this is the first work that characterizes the exact Pareto front for even finite MDPs. The general question is beyond the scope of the current paper.
>
> [1] Metz, Luke, et al. "Discrete sequential prediction of continuous actions for deep RL." arXiv preprint arXiv:1705.05035 (2017).
>
> [2] Andrychowicz, OpenAI: Marcin, et al. "Learning dexterous in-hand manipulation." The International Journal of Robotics Research 39.1 (2020): 3-20.
>
> [3] Tang, Yunhao, and Shipra Agrawal. "Discretizing continuous action space for on-policy optimization." Proceedings of the AAAI conference on artificial intelligence. Vol. 34. No. 04. 2020.

---

> > ### Comment · Reviewer_QTi5 · 2024-11-27
> >
> > Thank you for addressing the question regarding the potential extension to continuous action spaces. The discrete action space constraint is similar to the Pareto Q-learning approach discussed in Section 2. While not directly within the scope of this paper, exploring the potential combination with approximation methods or extensions to continuous spaces could broaden the applications of this algorithm.

---

### Official Review · Reviewer_JJrB · 2024-11-10

**Soundness:** 3
**Presentation:** 4
**Contribution:** 3
**Rating:** 8
**Confidence:** 3

**Summary:**

The authors provide an algorithm for finding the entire Pareto-optimal front for multi-objective MDPs. Their algorithm is based upon several structural/theoretical observations about the geometry of the Pareto front, relating neighboring vertices of the Pareto front to policies which differ in a single state-action pair. This leads to their local-search-based algorithm, which outperforms baselines and prior approaches.

**Strengths:**

I think the paper is written very clearly. In particular the proof sketches are structured in a helpful manner.
The paper makes good use of examples to explain technical points of the algorithm (ex. paragraph around 407).

The paper also seems to make a original and significant contribution to the study of multi-objective MDPs, not just outperforming prior work but also developing a fundamentally different algorithmic approach and presenting useful theoretical/structural results.

**Weaknesses:**

In other settings the assumption of ergodicity is a significant one. I think this should be highlighted a bit more.

This is subjective but I think the organization of section 3 would be improved if the problem setup (Section 3.2) was provided before the algorithm overview (Section 3.1). (I still think the algorithm overview is very helpful.)

I think the wording of the definition of ergodicity should be clarified. The provided definition sounds similar to the definition of a communicating MDP.

The plots in Figures 5 and 6 are hard to understand due to overlapping curves. I think changing the y-axis scale might be a better solution than the current approach. Also I think the caption of Figure 6 is incorrect? (Should be OLS not benchmark)

**Questions:**

Could you give an overview of the difficulties in generalizing your results beyond the ergodicity assumption and describe how it is used in the proofs?

Regarding the benchmark algorithm in Section 5.1, are there situations where the benchmark algorithm has comparable performance to the main algorithm, and what would they look like?

---

> ### Author Response · Authors · 2024-11-20
> **Response to Reviewer JJrB**
>
> We have uploaded the revised paper, with the major changes highlighted in blue.
>
> > In other settings the assumption of ergodicity is a significant one. I think this should be highlighted a bit more. I think the wording of the definition of ergodicity should be clarified. The provided definition sounds similar to the definition of a communicating MDP. Could you give an overview of the difficulties in generalizing your results beyond the ergodicity assumption and describe how it is used in the proofs?
>
> Thank you for your excellent suggestion. After reviewing our proofs carefully, we found that the ergodicity assumption is not necessary, and we have revised the paper accordingly.
>
> In our original analysis, we assumed that for any two deterministic policies $\pi_1$ and $\pi_2$ that differ only on a subset of states $\mathcal{M}$ (i.e., they behave identically in $\mathcal{S} \backslash \mathcal{M}$), any state in $\mathcal{S} \backslash \mathcal{M}$ can reach $\mathcal{M}$ within a finite number of steps. This assumption was meant to ensure that the difference between the long-term returns of $\pi_1$ and $\pi_2$ arises solely from their behavior on $\mathcal{M}$.
>
> However, upon closer inspection, we realized that for states that can never reach $\mathcal{M}$ in finite steps, $\pi_1$ and $\pi_2$ will always yield the same value function for those states. As a result, these states do not affect the difference in long-term returns between the two policies. Thus, it suffices to focus only on the states that can reach $\mathcal{M}$ in finite steps, without requiring ergodicity.
>
>
> > This is subjective but I think the organization of section 3 would be improved if the problem setup (Section 3.2) was provided before the algorithm overview (Section 3.1). (I still think the algorithm overview is very helpful.)
>
> Thank you for your suggestion. We have revised the paper to present the problem setup (previous Section 3.2) before the algorithm overview (previous Section 3.1).
>
>
> > The plots in Figures 5 and 6 are hard to understand due to overlapping curves. I think changing the y-axis scale might be a better solution than the current approach. Also I think the caption of Figure 6 is incorrect? (Should be OLS not benchmark)
>
> Thanks for your suggestion and pointing out the typo. We changed the y-axis scale of Figs. 5 and 6 to make the comparison more clear. And I also corrected the caption of Fig. 6.
>
>
> > Regarding the benchmark algorithm in Section 5.1, are there situations where the benchmark algorithm has comparable performance to the main algorithm, and what would they look like?
>
> The proposed algorithm has better computational efficiency than the benchmark algorithm (whose pseudo-code is provided in Appendix I) in all cases. The benchmark algorithm is designed to generate the ground-truth Pareto front and validate the effectiveness of our proposed algorithm. It identifies all non-dominated deterministic policies through pairwise comparisons and constructs a convex hull over these policies. The Pareto-optimal faces are then selected using the same criteria as our proposed Pareto front searching algorithm.
>
> However, since the benchmark algorithm evaluates all possible deterministic policies, its computational cost is significantly higher. In contrast, the proposed algorithm focuses only on evaluating neighboring policies, which drastically reduces the number of candidate policies to evaluate.  The exhaustive evaluation process of the benchmark algorithm inherently makes it less efficient, especially as the size of the state and action spaces grows. As a result, the proposed algorithm consistently outperforms the benchmark algorithm in terms of efficiency.

---

> > ### Comment · Reviewer_JJrB · 2024-11-26
> >
> > Thank you for your response. I will maintain my score and continue to recommend acceptance.

---

### Official Review · Reviewer_n21h · 2024-11-12

**Soundness:** 3
**Presentation:** 3
**Contribution:** 3
**Rating:** 6
**Confidence:** 3

**Summary:**

This paper studies the multi-objective MDPs and explores the properties of their Pareto front. By using the insight that the Pareto front is on the boundary of a convex polytope where each vertex is a deterministic policy, a fast algorithm is designed to compute the Pareto front efficiently.

**Strengths:**

- This paper is well written.

- The idea is interesting and novel, and the results are significant.

**Weaknesses:**

- The complexity analysis lacks clarity, particularly when compared to other algorithms. A more explicit comparison would significantly improve the discussion.

- The numerical evaluation is restricted to small instances, which are insufficient to demonstrate the scalability of the proposed algorithm.

**Questions:**

see weaknesses.

---

> ### Author Response · Authors · 2024-11-20
> **Response to Reviewer n21h**
>
> We have uploaded the revised paper, with the major changes highlighted in blue.
>
> > The complexity analysis lacks clarity, particularly when compared to other algorithms. A more explicit comparison would significantly improve the discussion.
>
> Thanks for your suggestion. We modified the complexity comparison part and moved the detailed comparison to Appendix H.
>
> Let $S$, $A$, and $D$ represent the state space size, the action space size, and the reward dimension, respectively.  Let $N$ denote the number of vertices on the Pareto front. The overall computational complexity of our algorithm is $\mathcal{O}(C_{\text{so}}+NSAC_{\text{pe}}+NC_{\text{ND}}+NC_{\text{cvx}})$, where the terms are defined as follows:
>
> (1) $C_{\text{so}}$: The computational complexity of single-objective planning is $C_{\text{so}} = \mathcal{O}\left(S^4A+S^3A^2\right)$;
>
> (2) $C_{\text{pe}}$: The computational complexity of the policy evaluation is $C_{\text{pe}} = \mathcal{O}(S^3D)$;
>
> (3) $C_{\text{ND}}$: $C_{\text{ND}}$ refers to the complexity of finding $\Pi_{1,\text{ND}}$, the set of non-dominated policies from $\Pi_1$. Let $N_{\text{ND}} = |\Pi_{1,\text{ND}}| = $. The non-dominated policies searching complexity is $C_{\text{ND}} = \mathcal{O}(SAN_{\text{ND}}D)$.
>
> (4) $C_{\text{cvx}}$ is the complexity of computing the convex hull faces formed by $N_{\text{ND}}+1$ points that intersect at the current policy. If the number of vertices of the output convex hull is $N_{\text{cvx}}$. By [1], the complexity of the calculation of convex hull is $\mathcal{O}(N_{\text{ND}}\log N_{\text{cvx}})$ when $D\leq 3$ and $\mathcal{O}(N_{\text{ND}}f_{N_{\text{cvx}}}/N_{\text{cvx}})$ when $D>3$, where $f_L=\mathcal{O}(L^{\lfloor D/2\rfloor}/(\lfloor D/2\rfloor!))$.
>
>
> For each iteration, OLS solves the single-objective MDP with a preference vector, removes obsolete preference vectors with the new points, and computes new preference vectors. For each iteration, solving the single-objective planning problem incurs a complexity of $\mathcal{O}(C_{\text{so}}+C_{\text{pe}})$. The complexity of comparing with the previous candidate Pareto optimal policies and removing obsolete ones is $\mathcal{O}(ND)$, and the complexity of calculating a new preference vector is denoted as $C_{\text{we}}$.
> The computational complexity of OLS is $\mathcal{O}(N^2D+NC_{\text{we}}+N\left(S^4A+S^3A^2\right)+NS^3D)$.
> Note that calculating new weight requires calculating the vertices of the convex hull constructed by $N+D+2$ hyperplanes of dimension $D+1$, making it significantly more expensive than the local convex hull construction of neighboring policies in our proposed algorithm. Hence, $C_{we}$ is much larger than $C_{\text{cvx}}$.
>
> In conclusion, our algorithm induces total computational complexity of $\mathcal{O}(S^4A+S^3A^2+NS^4AD+NSAN_{\text{ND}}D+C_{\text{cvx}})$, while OLS has computation complexity $\mathcal{O}(N(S^4A+S^3A^2)+NS^3D+N^2D+NC_{\text{we}})$. Given $C_{\text{cvx}}<C_{\text{we}}$, our algorithm is more efficient than OLS in all cases.
>
>
>
>
>
> > The numerical evaluation is restricted to small instances, which are insufficient to demonstrate the scalability of the proposed algorithm.
>
>
>
> We evaluate the proposed Pareto front searching algorithm with the state space size ranging from 10 to 50 and the action space sizes ranging from 10 to 20. The reward dimension is 3. With the computational complexity of the benchmark algorithms becoming extremely high in this setting, we focus solely on the proposed algorithm.
>
> Fig. 8 on Page 31 shows the relationship between the number of vertices and the state/action space size. As shown in the figure, the number of vertices grows approximately as $N\approx kS^3A$, where k is a constant number. Combined with the computational complexity analysis $\mathcal{O}(\left(S^4A+S^3A^2\right)+NS^4AD+NSAN_{\text{ND}}D+C_{\text{cvx}}))$, this demonstrates the scalability of our algorithm.

---

> > ### Comment · Reviewer_n21h · 2024-11-26
> >
> > Thank you for your response. After reviewing your feedback and considering the comments from other reviewers, I have decided to maintain my score.

---

### Official Review · Reviewer_5te3 · 2024-11-13

**Soundness:** 3
**Presentation:** 4
**Contribution:** 3
**Rating:** 8
**Confidence:** 3

**Summary:**

This paper presents an efficient method to identify the exact Pareto front in multi-objective Markov Decision Processes (MO-MDPs). The authors use the geometric insight that the Pareto front forms on a polytope boundary, allowing for a simplified, localized search by adjusting one state-action pair at a time. This approach keeps computations efficient and complete.

**Strengths:**

1. The paper takes an original approach to MO-MDPs, revealing geometric properties of the Pareto front that support an efficient search algorithm, standing apart from previous methods that rely on scalarizing preferences or using approximations.
2. The algorithm provides a strong computational edge, needing only one single-objective solution to start, then systematically traversing edges between Pareto-optimal policies—ideal for applications requiring efficient navigation of complex Pareto fronts.

**Weaknesses:**

1. The complexity analysis could benefit from a more detailed comparison with existing algorithms, especially in larger, higher-dimensional state-action spaces. Testing on bigger benchmarks would clarify the algorithm's scalability and relevance for complex MO-MDPs.
2. The experiments use small, artificial setups. Adding tests in larger, more realistic MO-MDP scenarios would better show the algorithm’s practical applicability and effectiveness.

**Questions:**

Could the authors elaborate on potential modifications to improve the algorithm's scalability with larger action and state spaces?

---

> ### Author Response · Authors · 2024-11-20
> **Response to Reviewer 5te3 [1/2]**
>
> We have uploaded the revised paper, with the major changes highlighted in blue.
>
> > The complexity analysis could benefit from a more detailed comparison with existing algorithms, especially in larger, higher-dimensional state-action spaces. Testing on bigger benchmarks would clarify the algorithm's scalability and relevance for complex MO-MDPs.
>
> Thanks for your suggestion. We modified the complexity comparison part and moved the detailed comparison to Appendix H.
>
> Let $S$, $A$, and $D$ represent the state space size, the action space size, and the reward dimension, respectively.  Let $N$ denote the number of vertices on the Pareto front. The overall computational complexity of our algorithm is $\mathcal{O}(C_{\text{so}}+NSAC_{\text{pe}}+NC_{\text{ND}}+NC_{\text{cvx}})$, where the terms are defined as follows:
>
> (1) $C_{\text{so}}$: The computational complexity of single-objective planning is $C_{\text{so}} = \mathcal{O}\left(S^4A+S^3A^2\right)$;
>
> (2) $C_{\text{pe}}$: The computational complexity of the policy evaluation is $C_{\text{pe}} = \mathcal{O}(S^3D)$;
>
> (3) $C_{\text{ND}}$: $C_{\text{ND}}$ refers to the complexity of finding $\Pi_{1,\text{ND}}$, the set of non-dominated policies from $\Pi_1$. Let $N_{\text{ND}} = |\Pi_{1,\text{ND}}| = $. The non-dominated policies searching complexity is $C_{\text{ND}} = \mathcal{O}(SAN_{\text{ND}}D)$.
>
> (4) $C_{\text{cvx}}$ is the complexity of computing the convex hull faces formed by $N_{\text{ND}}+1$ points that intersect at the current policy. If the number of vertices of the output convex hull is $N_{\text{cvx}}$. By [1], the complexity of the calculation of convex hull is $\mathcal{O}(N_{\text{ND}}\log N_{\text{cvx}})$ when $D\leq 3$ and $\mathcal{O}(N_{\text{ND}}f_{N_{\text{cvx}}}/N_{\text{cvx}})$ when $D>3$, where $f_L=\mathcal{O}(L^{\lfloor D/2\rfloor}/(\lfloor D/2\rfloor!))$.
>
>
> For each iteration, OLS solves the single-objective MDP with a preference vector, removes obsolete preference vectors with the new points, and computes new preference vectors. For each iteration, solving the single-objective planning problem incurs a complexity of $\mathcal{O}(C_{\text{so}}+C_{\text{pe}})$. The complexity of comparing with the previous candidate Pareto optimal policies and removing obsolete ones is $\mathcal{O}(ND)$, and the complexity of calculating a new preference vector is denoted as $C_{\text{we}}$.
> The computational complexity of OLS is $\mathcal{O}(N^2D+NC_{\text{we}}+N\left(S^4A+S^3A^2\right)+NS^3D)$.
> Note that calculating new weight requires calculating the vertices of the convex hull constructed by $N+D+2$ hyperplanes of dimension $D+1$, making it significantly more expensive than the local convex hull construction of neighboring policies in our proposed algorithm. Hence, $C_{we}$ is much larger than $C_{\text{cvx}}$.
>
> In conclusion, our algorithm induces total computational complexity of $\mathcal{O}(S^4A+S^3A^2+NS^4AD+NSAN_{\text{ND}}D+C_{\text{cvx}})$, while OLS has computation complexity $\mathcal{O}(N(S^4A+S^3A^2)+NS^3D+N^2D+NC_{\text{we}})$. Given $C_{\text{cvx}}$ is far smaller than $C_{\text{we}}$, our algorithm is more efficient than OLS in all cases.
>
> We evaluate the trend of the number of vertices $N$ in terms of $S$ and $A$ with the state space size ranging from 10 to 50 and the action space sizes ranging from 10 to 20. The reward dimension is 3. The result shows that the number of vertices grows approximately as $N\approx kS^3A$, where k is a constant number. Combined with the computational complexity analysis $\mathcal{O}(\left(S^4A+S^3A^2\right)+NS^4A+NSAN_{\text{ND}}D+C_{\text{cvx}}))$, this demonstrates the scalability of our algorithm. We also added the experiment result in Appendix J to reflect the result.
>
> > The experiments use small, artificial setups. Adding tests in larger, more realistic MO-MDP scenarios would better show the algorithm’s practical applicability and effectiveness.
>
>
> We adopt the deep-sea-treasure setting from [1] and construct a scenario where the agent navigates a grid world, making trade-offs between different objectives. At each step, the agent can move up, down, left, or right, and the reward for each grid is generated randomly. We compare the performance of the OLS algorithm and the proposed Pareto front searching algorithm under different grid sizes (varying numbers of rows and columns). The experiments were conducted on smaller grid sizes to ensure that the running time of OLS remained manageable. The figure shows that the proposed algorithm consistently outperforms OLS in terms of efficiency across all tested settings. We attached the related experiments in Appendix J.
>
> [1] Yang, Runzhe, Xingyuan Sun, and Karthik Narasimhan. "A generalized algorithm for multi-objective reinforcement learning and policy adaptation." Advances in neural information processing systems 32 (2019).

---

> ### Author Response · Authors · 2024-11-20
> **Response to Reviewer 5te3 [2/2]**
>
> > Could the authors elaborate on potential modifications to improve the algorithm's scalability with larger action and state spaces?
>
> Scalability is a critical consideration for real-world applications involving large state and action spaces. Below, we outline several potential modifications to improve the scalability of the proposed algorithm.
>
> **State and Action Space Abstraction**: Abstraction techniques can be applied to reduce the effective size of the problem while preserving the key characteristics of the MDP. By aggregating similar states or actions, the algorithm operates on a simplified model, which improves the efficiency of finding the Pareto front.
>
> **Integration with Approximate Pareto Front Estimation**: Another promising direction is to integrate the proposed algorithm with an approximate Pareto Front estimation. This method would use the approximate Pareto front as a warm start, providing a good initial estimate. The algorithm could then refine this estimate to explore the exact Pareto front more efficiently.

---

> > ### Comment · Reviewer_5te3 · 2024-11-27
> >
> > Thank you for your response. I will maintain my score.

---

### Meta-Review · Area_Chair_gERs · 2024-12-21

**Metareview:**

This paper studies methods for identifying the exact Pareto front in Multi-Objective Markov Decision Processes (MO-MDPs), leveraging novel geometric insights that the Pareto front lies on the boundary of a convex polytope and can be efficiently explored through localized searches among deterministic policies. It presents a new algorithm that is theoretically grounded and computationally efficient, outperforming prior methods like Optimistic Linear Support (OLS) in empirical studies. The reviewers found the paper to be well-written, theoretically novel, and a significant contribution to MO-MDP research, although they suggested clarifying complexity analysis, scalability, and adding practical evaluations. During the rebuttal phase, the reviewers were satisfied with the authors’ detailed responses and updates, leading to a consensus on acceptance.

**Additional Comments On Reviewer Discussion:**

During the rebuttal period, reviewers raised several points, including the need for clearer complexity analysis, enhanced scalability discussions, practical evaluations beyond small, artificial setups, and better integration of prior works, such as comparisons with approximation methods. Reviewer 4Q5J questioned computational efficiency in larger state-action spaces and suggested clarifications in the algorithm and proofs. Reviewer WFhh sought practical examples for using the full Pareto front and addressed robustness and sensitivity to errors. Reviewer n21h and others requested improved explanations of assumptions and adjustments for clarity in presentation. The authors addressed these concerns comprehensively by adding detailed complexity analyses, expanded experimental results with larger setups, practical examples of Pareto front usage, and clarifications in proofs and assumptions. They also updated figures, algorithms, and text for better clarity and organization. These updates adequately resolved reviewer concerns, demonstrating the paper's thoroughness and robustness. This responsiveness, combined with the consensus on the paper's theoretical contributions and novelty, strongly informed my decision to recommend acceptance.

---

### Decision · Program_Chairs · 2025-01-22

Accept (Spotlight)